# Advances in altimetric snow depth estimates using bi-frequency SARAL/CryoSat-2 Ka/Ku measurements.

Florent Garnier[1], Sara Fleury[1], Gilles Garric[2], Jérôme Bouffard[3], Michel Tsamados[4], Antoine Laforge[5], Marion Bocquet[1], Rénée Mie Fredensborg Hansen[3], and Frédérique Remy[1]

[1]Laboratoire d'Etudes en Géophysique et Océanographie Spatiales (LEGOS), CNRS/UMR5566, Université Paul Sabbatier, 31400 Toulouse, France
[2]Mercator Ocean, Ramonville Saint Agne, 31520, France
[3]ESA (European Space Agency), Earth Observation Directorate, Via Galileo Galilei, 2-00044 Frascati, Italy
[4]Centre for Polar Observation and Modelling, Department of Earth Sciences, University College London, London, WC1E 6BT, UK
[5]Serco c/o ESA, Earth Observation Directorate, Via Galileo Galilei, 2-00044 Frascati, Italy

**Correspondence:** F. Garnier (florent.garnier@legos.obs-mip.fr)

**Abstract.** Although snow depth on sea ice is a key parameter for sea-ice thickness (SIT) retrieval, there currently does not exist reliable estimations. In the Arctic, nearly all SIT products use a snow depth climatology (the modified Warren-99 climatology, W99m) constructed from in-situ data obtained prior to the first significant impacts of climate change. In the Antarctic, the lack of information on snow depth remains a major obstacle in the development of reliable SIT products.

In this study, we present the latest version of the Altimetric Snow Depth (ASD) product computed over both hemispheres from the difference of the radar penetration into the snow pack between the Ka-band frequency SARAL/Altika and the Ku-band frequency CryoSat-2. The ASD solution is compared against a wide range of snow depth products including model data (Pan-Arctic Ice-Ocean Modelling and Assimilation System (PIOMAS) or its equivalent in the Antarctic the Global Ice-Ocean Modeling and Assimilation System (GIOMAS), the MERCATOR model and NASA's Eulerian Snow On Sea Ice Model

(NESOSIM, only in the Arctic)), the Advanced Microwave Scanning Radiometer-2 (AMSR2) passive radiometer data, and the Dual-altimeter Snow Thickness (DuST) Ka/Ku product (only in the Arctic). The ASD product is further validated in the Arctic against the Ice Mass Balance (IMB) buoys, the CRYOsat Validation EXperiment (CryoVEx) and Operation Ice Bridge's (OIB) airborne measurements. These comparisons demonstrate that ASD is a relevant snow depth solution, with spatiotemporal patterns consistent with those of the alternative Ka/Ku DuST product, but with a mean bias of about 6.5 cm. We

also demonstrate that ASD is consistent with the validation data: comparisons with Operation Ice Bridge's (OIB) airborne snow radar in the Arctic during the period of 2014-2018 show a correlation of 0.66 and a RMSE of about 6 cm. Furthermore, a first-guess monthly climatology has been constructed in the Arctic from the ASD product, which shows a good agreement with OIB during 2009-2012. This climatology is shown to provide a better solution than the W99m climatology when compared with validation data. Finally, we have characterised the SIT uncertainty due to the snow depth from an ensemble of SIT solutions

computed for the Arctic by using the different snow depth products previously used in the comparison with the ASD product. During the period of 2013-2019, we found a spatially averaged SIT mean standard deviation of 20 cm. Deviations between SIT

estimations due to snow depths can reach up to 77 cm. Using the ASD data instead of W99m to estimate SIT over this time period leads to a reduction of the average SIT of about 30 cm.

## 1 Introduction

Since the launch of CryoSat-2 (CS-2) in 2010 (Wingham et al., 2006; Parrinello et al., 2018), sea-ice thickness (SIT) observations are routinely derived from altimetric measurements. The principle is to measure the fraction of the sea ice above the sea level, called the sea ice freeboard, from differences between the heights in leads (cracks in the ice referring to the local sea level) and the heights of the ice floes (Laxon et al., 2003). By integrating such sea ice freeboard estimations in the hydrostatic equilibrium equation, several SIT products have been computed (e.g., Laxon et al., 2013; Kwok and Cunningham, 2015; Guerreiro et al., 2017; Paul et al., 2018; Landy et al., 2019; Laforge et al., 2020).

Among the parameters involved in the SIT calculation, snow depth (sd) over sea ice is one of the most significant contributors adding to the overall SIT uncertainty (e.g., Giles et al., 2007; Zygmuntowska et al., 2014; Guerreiro et al., 2016). For example, it is necessary to account for the snow loading (Laxon et al., 2013) and for the decrease in altimetric radar speed as it penetrates into the snow pack (Kwok and Cunningham, 2015; Mallett et al., 2020). In fact, variabilities of snow properties affect radar signals and then the snow depth measurements. more generally, snow cover has strong impacts on the sea ice (e.g., Massom et al., 2001; Powell et al., 2005; Bin et al., 2008; Sturm and Massom, 2009; Ricker et al., 2014) that affects the entire climate system (e.g., Ingram et al., 1989; Ledley, 1991; Eicken et al., 1995; Singarayer et al., 2006). Because of its high albedo and a low thermal conductivity, the snow regulates the transfer of solar heat energy penetration across the ice-ocean interface (e.g., Grenfell and Maykut, 1977; Sturm et al., 1997). It acts as an insulator, slowing down sea ice melt in summer and slowing down sea ice growth in winter (e.g., Perovich et al., 2003; Sturm and Massom, 2016). Such processes of sea ice formation and melting govern sea ice physical and chemical properties that impact the biological processes in sea ice (Van Leeuwe et al., 2018). The vertical distribution of light under sea ice that control biological processes and biogeochemical fluxes is also strongly linked with the snow depth (e.g., Perovich, 2007; Arndt and Nicolaus, 2014; Arndt et al., 2017). In addition snow accumulated over sea ice from precipitations represent a tank of freshwater likely to be carried into the ocean. Recent increasing of seasonal ice has promoted the amount of snow water discharge in the ocean which impacts the freshwater budget (Andersen et al., 2019; Overland et al., 2019). Snow cover also modifies surface roughness that impacts the air/ice drag coefficient and transfer coefficients of latent and sensible heat fluxes (Andreas et al., 2005).

Despite evidence of its major importance, snow cover over sea ice is still not sufficiently known. Until now, nearly all SIT products in the Arctic are computed using the snow depth of the modified Warren-99 climatology (W99m, Warren et al. (1999)). W99m is mainly constructed using snow depth observations obtained from the Soviet North Pole drifting stations based in the central Arctic, collected during the last century (1950-1980s). Considering the ongoing rapid modifications of the Arctic due to climate change, several studies have shown these data to be outdated, even with the modification of snow depth reduction (by 50%) applied over the seasonal ice zone (Kurtz and Farrell, 2011; Kern et al., 2015; Kwok et al., 2011). The climatology of Forsström et al. (2011) provides snow depths in areas outside the central Arctic where the W99m climatology

is not working properly. More recently, Shalina and Sandven (2018) have created an improved snow depth climatology taking into account snow depth observations obtained during campaigns over seasonal ice. However, it is still based on data collected mainly in the 1960s, 1970s and 1980s. Apart from the data used to construct these climatologies, several campaigns have provided snow depth measurements in the Arctic. Between 1997 and 1998, the SHEBA project has highlighted the complex temporal and spatial snow depth variations (Perovich et al., 1999; Sturm et al., 2002). The Ice Mass Balance buoys (IMB), originally deployed by the U.S. Army CRREL-Dartmouth Mass Balance Program during the SHEBA project have measured snow depths since the 2000s. It allows for monitoring changes of the sea ice volume in key areas of the Arctic (Richter-Menge et al., 2006). Since 2003, the CryoSat Validation Experiment (CryoVEx) campaigns (e.g Haas et al. (2006); Helm et al. (2006); Skourup et al. (2013); Hvidegaard et al. (2006)) have provided data with the main goal of investigating radar penetrations into ice and snow cover. Their measurements include bi-frequency altimetry snow depth estimations as of 2017 (H. Skourup and Wilkinson, 2019) (see Sect. 3.3). In 2003, the AMSR-Ice03 campaign (Sturm et al., 2006) was carried out in the Beaufort sea region to validate the Advanced Microwave Scanning Radiometer for the Earth Observing System (AMSR-E, from 2002 to 2011, Kelly (2009)) passive microwave snow depth data. The same year, an equivalent mission (ARISE) took place in East Antarctic (Massom et al., 2006). Since 2009, and until their final campaign in 2020, the Operation Ice Bridge (OIB, see Sect. 3.3) campaigns have each year between March and April provided airborne snow depth measurements in the Arctic (Kurtz et al., 2013; Koenig et al., 2010) using a FMCW snow radar (Kurtz and Farrell, 2011). Note also, that Webster et al. (2014) assesses spring snow depth distribution on the Arctic sea ice from 2009-2013 airborne OIB radar observations. In parallel, in-situ snow thickness data were collected in 2009 at the Danish GreenArc 2009 ice camp in the North of Greenland (Farrell et al., 2011). Between February and March 2015, snow depth was measured in the Nansen Basin during the N-ICE expedition (Granskog et al., 2016), demonstrating among others the impact of heavy precipitations on thin ice growth (Merkouriadi et al., 2017). From September 2019 to October 2020, the international MOSAIC expedition monitored the central Arctic (Shupe et al., 2020) providing, among others, data on snow precipitation, snow water equivalent (SWE) and magnaprobe snow depth measurements (e.g., Munoz-Martin et al., 2020). Hence, several campaigns have been conducted during the last two decades in the Arctic, all of which have provided crucial information and proved invaluable as reference data.

Compared to the Arctic, snow depth measurements, and more generally sea ice in-situ data, are very limited in the Antarctic. Actually, several CryoVEx and OIB missions have been conducted in the Antarctic, but the data are still only available in raw L1b level and difficult to interpret. To the best of our knowledge, only the meereis data portal (https://seaiceportal.de/en/) provides snow buoys data of snow accumulation (Grosfeld et al., 2016) (for both hemispheres), but the comparison with snow depth is not direct. It is, for instance, limited by the flooding of ice floes due to the heavy snow loading occurring in the Antarctic. Currently, the main expedition providing snow depth data in the Southern hemisphere, is the ASPeCt program (Worby et al., 2008a), established in 1996 to model the role of the Antarctic sea ice in the coupled atmosphere-ice-ocean system. ASPeCt recent standardized data (Kern, 2020) cover the period 2002-2019 with a too sparse amount of information for a reliable assessment. Future validations in the Antarctic will also benefit from the recent AWI IceBird campaigns carrying a snow radar since 2019 (https://www.awi.de/en/science/climate-sciences/sea-ice-physics/projects/ice-bird.html).

At a basic scale, several sea ice models produce snow accumulation estimations from atmospheric reanalyses. These are converted into snow depth over sea ice by considering sea-ice drifting and thermodynamic transfers of the coupled sea ice and snow system (Blanchard-Wrigglesworth et al., 2015; Lecomte et al., 2011). Likely due to biases in precipitation inputs and omission of snow processes, these global sea ice models hardly reproduce consistent on-ice snow depths (e.g., Holland et al., 1993; Serreze et al., 2000; Déry and Tremblay, 2004; Leonard and Maksym, 2011; Blazey et al., 2013). Notz (2012) states that an improved representation of the snow on sea ice in model is the most urgent task, and recently, Kaminski et al. (2018) argued that assimilating snow depth products could considerably increase sea ice model performances.

Therefore, a solution allowing to provide large scale information on the spatial and temporal variation of snow depth and to improve the performance of seasonal sea-ice forecasts, is to use satellite data. For now, only few datasets exist. The first snow depth estimates come from passive radiometric data. Among the various sensors that have been deployed (e.g., the SMMR, Chang et al. (1987) or the SSM/I Grody (1991) ), the most relevant estimations of snow depth over sea ice have been calculated from spectral vertical gradients of brightness temperatures (Comiso et al., 2003; Worby et al., 2008b) of AMSR-E and its successor AMSR2 (Lee et al., 2015)). In spite of the relevant long term availability of data, the commonly used snow depth retrieval algorithm is not adapted for Multi Year Ice (MYI) zones (Brucker and Markus, 2013). Consequently, AMSR's official data in the Arctic are only calculated in First Year Ice (FYI) regions. As far as we know, the AMSR2 product version of the University of Bremen (AMSR2B, Rostosky et al. (2018)) and very recently the SPICES project snow depth retrieval method (Mäkynen et al., 2020) are the only attempts to compute snow depths in the Arctic MYI zones. The dataset developed from the work of Rostosky et al. (2018) is available, but has the inconvenience of being re-calibrated from OIB data which limits their availability to the months of March and April. With the same disadvantage of depending on OIB, Maaß et al. (2013) and Zhou et al. (2018) have proposed algorithms to retrieve snow depth from SMOS, with mitigated results. Recently, Braakmann-Folgmann and Donlon (2019) proposed to compute snow depth by combining AMSR2 and SMOS data with a neural network approach.

Studies have also estimated snow depth on sea ice in the Antarctic (e.g., Kacimi and Kwok, 2020; Maksym and Markus, 2008). AMSR-E and AMSR2 also provide snow depths in the Antarctic. Compare to the Arctic it covers the whole domain since all sea ice is considered as seasonal ice. The recent work of Kacimi and Kwok (2020) estimated the Antarctic snow depth from the difference between the total freeboard obtained from ICESat-2 lidar measurements and sea ice freeboards of CS-2. They show a very low variability of CS-2 freeboards compared to ICESat-2 and that ICESat-2 freeboards explained >90% of the variance in snow depth. They also highlight that the validation sea ice parameters in the Antarctic remains a challenge, since no seasonally or regionally diverse datasets from field records can be used to assess the large-scale satellite retrievals. Based on this, they urge that more sustained and extensive field measurements must be conducted. Note that such lidar minus ku-band snow depths can also been computed in the Arctic. Finally, we note that a recent study of Fons et al. (2021), an update from the former approach experimented by Fons and Kurtz (2019), directly retrieves snow freeboard from CS-2 using a two-layer scattering model, and shows promise in estimations of SIT in the Antarctic.

An alternative to passive microwave radiometry is to derive snow depth estimations from bi-frequency altimetric measurements. Recently, Armitage and Ridout (2015) have demonstrated this possibility by considering the difference of penetration

between the Ku-band frequency CS-2 (13.5 GHz), assuming it is reflected near the snow/ice interface, and the Ka-band frequency SARAL/AltiKa (35.7 GHz), assuming it is reflected near the top of the snow pack, i.e the air/snow interface. Thereafter, a preliminary Altimetric Snow Depth (ASD) version covering the 2013-2016 winter period has been developed at the LEGOS during the ESA CryoSeaNICE project (Guerreiro et al., 2016). Meanwhile, Lawrence et al. (2018) also developed a bi-frequency altimeter snow depth product (DuST) available for the Arctic. Similar to AMSR2B, the main difference between ASD and DuST is that DuST relies on a re-calibration of the Ka and Ku freeboards using OIB data, in this case to account for the difference of operating mode between CS-2 and SARAL (Synthetic Aperture Radar (SAR) versus Low Resolution Mode (LRM) modes). These two products are the only existing publicly available snow depth products from altimetry.

A recent study of Zhou et al. (2020b) presented an inter-comparison of available snow depth products from re-analyses, passive radiometry and altimetry (DuST). Similar, this paper reviews the state-of-the-art by comparing current main snow depth estimations. Yet, the main objective is to present and assess the upgraded version of the ASD product (see Sect. 2), covering the 2013-2019 period in both hemispheres. The article fits in with the upcoming HPCM CRISTAL mission (Kern et al., 2020) and aims to demonstrate the potential of such snow depth data to further specific studies on, for instance, improved sea ice volume estimations, freshwater budgets, snow properties or data assimilation. Except from an analyse of the impact of the snow depth uncertainty on SIT retrieval, it doesn't explicitly address these questions.

The paper is organized as follows:

 – First, we detail the methodology to process the ASD product and present all the datasets used in this study.

 – Section 4 compare ASD with the main other existing snow depth satellite and model data in both hemispheres.

 – The datasets are then assessed against OIB, CryoVEx and IMB validation data in the Arctic in section 5.

 – To circumvent the temporal limitation induced by SARAL, we propose in section 6 a preliminary snow depth climatology based on ASD.

 – The last section 7 aims to quantify the SIT level of uncertainty due to the snow depth from an ensemble of SIT estimations calculated from the satellite and model snow depth datasets presented in the previous sections.

 – We finally discuss and conclude, emphasizing current needs for snow depth data in sea ice studies.

## 2  Data processing of ASD

The Altimetric Snow Depth product (ASD) presented here, is an upgraded version of the data presented in Guerreiro et al. (2016). It has been developed at the LEGOS laboratory in the framework of the ESA CSAO+ (http://cryosat.mssl.ucl.ac.uk/csao/index.html) and Polar+ Snow on Ice projects. It is freely available on the AVISO+/ODATIS national data centres since mid-2020 (http://ctoh.legos.obs-mip.fr/data/sea-ice-products.). The ASD product includes data from March 2013 to October 2019 provided on a monthly basis for the six winter months (from November to April in the Arctic and from May to October in

the Antarctic). It is projected onto a $500 \times 500$ EASE2 grid with a 12.5 km pixel size resolution. Snow depth calculation is based on the difference of penetration between AltiKa, the Ka-band range altimeter of SARAL (which is assumed to be reflected at the air/snow interface) and SIRAL, the Ku-band range altimeter of CS-2 (which is assumed to be reflected at the snow/ice interface). The main assumption is that the difference between these two surface elevations is only due to the penetration of the Ku radar in the snow pack, and that the Ku radar penetrates fully to the snow/ice interface. Note that the validity of

this hypothesis is strongly linked with the space and time variability of snow properties such as snow density, grain sizes, the liquid water content or the surface roughness. We use the CS-2 Ku-band waveforms in Pseudo LRM (PLRM) extracted from GOP (https://earth.esa.int/documents/10174/125272/CryoSat-Baseline-C-Ocean-Product-Handbook) generated by the ESA CryoSat Ocean Processor (Bouffard et al., 2018a). The Ka-band SARAL/AltiKa waveforms are extracted from the CNES sgdr T official products. Note that SARAL orbit is limited to $81.5°$ in latitude, which reduces the coverage of snow depth data

in the Arctic. The use of a degraded version of the CS-2 SAR waveforms (the PLRM mode), allows to have a footprint in accordance with the LRM mode of SARAL/AltiKa waveforms, which avoid to have different impact of the surface roughness. Since the latest Baseline-C PLRM GOP product was only available from 2017 at the time we have computed the ASD data, we have used the Baseline B for the period 2013-2016. It does not impact ASD since we use only the L1b product levels which have identical waveforms on both baselines. However, the baseline B does not include the SARin data. The next version of the

ASD product will be produced with only the Baseline-C PLRM GOP product to include all SARin mode zones.

The first step to compute ASD data consists in extracting heights, $H$, from both Ku-band and Ka-band satellite waveforms following Eq. (1):

$$H = alt - (range - (tropo_{dry} + tropo_{wet} + iono + MSS + tide_{ocean} + tide_{bar})), \tag{1}$$

where $alt$ is the satellite altitude and $range$ is the altimetric range retracked from the waveforms. $tropo_{dry}$ is the dry tropo-

175 spheric correction, $tropo_{wet}$ is the wet tropospheric correction, $iono$ is the ionospheric correction, $MSS$ is the DTU15 Mean Sea Surface correction (Andersen and Knudsen, 2015), $tide_{ocean}$ is the oceanic tide corrections and $tide_{bar}$ is the barometer tide correction.

The surface classification between heights corresponding to ocean surfaces (leads) and heights of sea ice surfaces (ice floes) is performed using the waveform pulse-peakiness (PP) criteria calculated from the waveforms (Eq. (2)):

$$PP = \frac{max(WF)}{\sum_{i=1}^{N_{WF}} WF_i}, \tag{2}$$

where $WF$ is the discrete waveform echoes and $N_{WF}$ is the number of along-track measurements. Following Guerreiro et al. (2017), surfaces with a PP larger than 0.3 are considered as leads and surfaces with a PP smaller than 0.1 are considered as sea ice floes. Observations with a PP between 0.1 and 0.3 are considered as mixed echoes and are discarded. This criteria is the same for SARAL and CS-2.

Altimetric ranges ($range$ parameter in Eq. (1)) of leads and floes are thereafter calculated using the TFMRA (Helm et al., 2014) with a 50% threshold. A 25 km median filter is then applied to the 20 Hz along-track surface height of sea ice floes and sea ice leads. For each observation, the SLA under floes are calculated as the median value of all leads in a 25 km radius around

the considered along-track point. Freeboard heights are computed by $fb = H_{floe} - H_{lead}$, where $H_{floe}$ is the heights of the ice floes and $H_{leads}$ the heights of the leads. Finally, a 25 km radius median smoothing is applied to the retrieved freeboards. The reason is that we assume that the ASD product should not be able to provide relevant information at smaller scales. However, additional analyses and comparisons with validation data would be necessary to properly characterize this point. Note that the ability to consistently observe small scales would certainly be significantly improved in the future dual-frequency snow depth products from the CRISTAL mission. In the first ASD data version (Guerreiro et al., 2016), snow depths were only calculated at CS-2 and SARAL crossing track points. For this version, snow depths are calculated from the difference between monthly EASE2 gridded freeboards as: $sd_r = fb_{Ka} - fb_{Ku}$, where $sd_r$ is a radar snow depth. Furthermore, to compute snow depth ($sd$), we need to take into account the decrease of the Ku radar echo velocity as it penetrates into the snow pack (Ulaby et al., 1986; Kwok and Cunningham, 2015):

$$sd = sd_r \times (1 + 0.51 \times \rho_s)^{(-1.5)}, \tag{3}$$

where $\rho_s$ is the snow density, set to $300 \ kg/m^3$. Note this formulation agrees with the conventional interpretation to correct for the slower wave propagation speed through snow, as recently shown in Mallett et al. (2020).

To estimate snow depth uncertainties, we assume that errors are unbiased, uncorrelated and follow the Gaussian propagation law. Snow depth uncertainty is then given by:

$$\delta_{sd} = \sqrt{(\delta_{sd_r} \times C)^2 + (sd_r \times B \times \delta_{\rho_s})^2}, \tag{4}$$

with

$$C = (1 + 0.51 \times \rho_s)^{-1.5}, \quad B = -1.5 \times 0.51 \times (1 + 0.51 \times \rho_s)^{-2.5}, \quad \delta_{sd_r} = \sqrt{\delta_{fb_{Ka}}^2 + \delta_{fb_{Kur}}^2} \quad and \quad \delta_{\rho_s} = 3.2.$$

The freeboard uncertainties $\delta_{fb} = \sqrt{\delta_{H_{leads}}^2 + \delta_{H_{floes}}^2}$ are calculated along-track from the statistical dispersion of heights in a 12.5 km radius. Since sea ice topography can significantly vary within this area, we assume that the statistical dispersion of floes is the same as that of the leads.

$$\delta_{H_{floes}}^2 = \frac{\sigma_{H_{leads}}}{N_{floes}} \quad and \quad \delta_{H_{leads}}^2 = \frac{\sigma_{H_{leads}}}{N_{leads}}. \tag{5}$$

Using this methodology, Fig. 1 presents an example of ASD snow depth estimations and its associated uncertainties in the two hemispheres. In the Antarctic, thicker snow is located in the Weddell sea and, in a less extend, in the coastal zones of the bellingshausen and Amundsen seas. It is also relevant to identify the very low snow values associated with the Ross ice shelf. In the Arctic, the snow distribution follows well the dynamics of sea ice, the most characteristic of which is the export of MYI in the Beaufort gyre. This figure also shows that the ASD data are very different from the W99m climatology, where the W99m climatology tend to exhibit thicker snow layers over sea ice. The ASD mean level of uncertainties (of about 4 cm on average with standard deviation of 1 cm) are smaller than the deviations between these two products (on average about 6 cm with standard deviation of 7 cm).

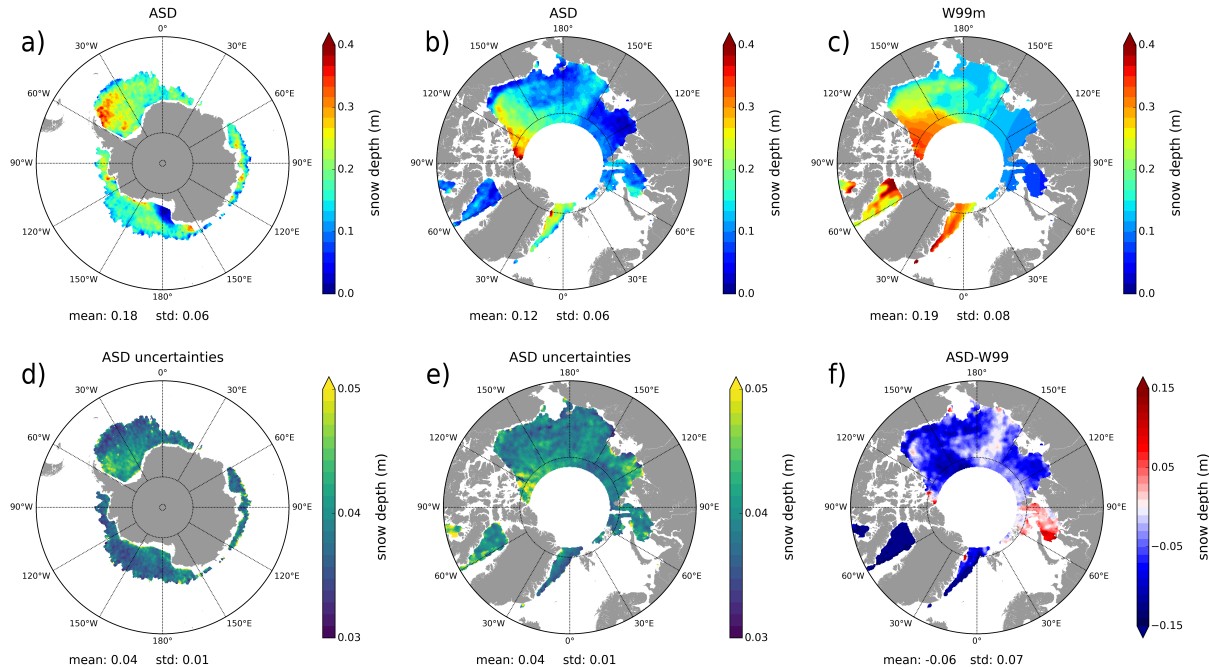

**Figure 1.** Maps of the Altimetric Snow Depth (ASD) annual mean snow depth (a) and its uncertainties (d) in the Antarctic and the Arctic (b and e) for 2015. the third column presents the annual mean snow depth map of the modified Warren-99 climatology (c) and its difference with the ASD product (ASD-W99m) in the Arctic (f) in 2015.

.

## 3 External datasets

This section presents the various snow depth datasets used in this study. For both hemispheres, the time period of model and
satellite products ranges from 2014–2019 for the Arctic, and 2013–2019 for the Antarctic (with limitations for some data products for both hemispheres), as explained in Fig. 2. Table A4 specifies the time period of the different satellite and model data.

### 3.1 Satellite data

#### 3.1.1 DuST

The DuST (Lawrence et al., 2018) data are provided in the Arctic on a 1.5° longitude x 0.5° latitude grid for by the CPOM/UCL (www.cpom.ucl.ac.uk/DuST). They are available for the 6 winter month (November to April) until 2018. Similar to ASD data, DuST relies on the difference of penetration between the Ka-band SARAL/Altika and the Ku-band CS-2 radar altimeters. However, unlike the ASD product (which uses the CS-2 PLRM data, see Sect. 2), the DuST product uses SAR mode freeboard estimates of CS-2. The CS-2 and SARAL (LRM data) data are then calibrated against data from National Aeronautics and

Space Administration's (NASA's) OIB airborne measurements to align the freeboard observations from SARAL/AltiKa with the snow/air interface, and the freeboard observations from CS-2 with the snow/ice interface. Although this method provides a generic approach adaptable to different footprints (e.g., ICESat-1 and 2), it has the disadvantage of being dependent on the OIB data which are mainly located near the Canadian Archipelago and the Beaufort Sea, and only obtained during March and April.

### 3.1.2 AMSR2

The Advanced Microwave Scanning Radiometer-2 (AMSR2) is a passive radiometer on-board the JAXA/GCOM-W satellite launched in 2012. Snow depth on sea ice retrieval is based on the gradient ratio of vertically polarized brightness temperature at 19 and 37 GHz (Markus and Cavalieri, 1998), and two snow depth products are available. The first product (AMSR2-NSIDC), described in Meier et al. (2018), is accessible on the NSIDC (https://nsidc.org/data/AU_SI12/versions/1). It provides daily L3

snow depth data in 12.5 km x 12.5 km stereopolar grids constructed as five day running averages. Data are only calculated over FYI (with MYI concentration of less than 20%). This is due to the fact that MYI has a spectral signature similar to snow cover over FYI. Because of this limitation, we mainly focus on the Bremen AMSR2B product (Rostosky et al., 2018) developed at the Institute of Environmental Physics of the University of Bremen. This product is available on a daily basis from November 2012 to April 2018 on a polar stereographic grid with a 25 km x 25 km resolution. Unlike in AMSR2-NSIDC, March and April

snow depth data are also calculated over MYI for AMSR2B. Other months are only available over FYI. Note that the algorithm used to calculate snow depth over MYI is also calibrated with observations from OIB.

## 3.2 Model data

### 3.2.1 PIOMAS

The pan-Arctic Ice-Ocean Modelling and Assimilation System (PIOMAS, GIOMAS in the Antarctic) is a pan-Arctic coupled

ocean and sea ice model developed for climate applications (Zhang and Rothrock, 2003). Snow depth data are provided on a daily basis in a 360 x 120 generalized curvilinear coordinate system from the PIOMAS model version 2.1 (http://psc.apl. uw.edu/research/projects/arctic-sea-ice-volume-anomaly/data/model_grid). The atmospheric surface forcing fields are issued from NCEP–NCAR reanalysis. OSI SAF sea ice concentration from EUMETSAT is assimilated in near real time. A complete description can be found in Schweiger et al. (2011).

### 3.2.2 MERCATOR

In the framework of the CMEMS, Mercator Ocean implements a real time global analysis and forecasting system at 1/12° resolution. This system is based on the NEMO ocean platform (Madec et al., 2015) and driven by atmospheric conditions from the ECMWF IFS analysis and forecasting system. The sea ice model component is the Louvain-la-Neuve sea Ice Model (LIM2) (Fichefet and Maqueda, 1997; Vancoppenolle et al., 2012). The unique sea ice quantity assimilated within the system

is the near real time sea ice concentration from OSI SAF provided by CMEMS. The accumulation of the snow depth onto sea

ice originates from ECMWF snowfall forcing fields. A complete description can be found in Lellouche et al. (2018). Snow depth data used in this study has been provided in a monthly basis on the ORCA tripolar grid. This LIM-2 model configuration will be referred as the MERCATOR model.

### 3.2.3 NESOSIM

The NASA Eulerian Snow On Sea Ice Model (NESOSIM) is a 3D snow budget model configured to produce snow depth and density over sea ice in the Arctic Ocean. Data used in this paper are snow depth monthly mean maps provided on a 100 km × 100 km stereographic polar grid issued from the NESOSIM 1.0 configuration (Petty et al., 2018). It is freely available at https://earth.gsfc.nasa.gov/cryo/data/nasa-eulerian-snow-sea-ice-model-nesosim. Snowfall data forcing field are the ECMWF ERA-Interim global reanalysis and a median of three reanalysis: the 55 years reanalysis of the JRA, the NASA's MERRA and 270 the ASR, version 1.

### 3.3 Validation data

The validation data presented in this section are only compared with ASD in the Arctic. As mentioned in the introduction CryoVEx and OIB data in the Antarctic are still only L1b. IMB are only in the Arctic. The geographical location of the validation data are shown in section 4.3.

### 3.3.1 CryoSat Validation EXperiment (CryoVEx)

These airborne data are a joint effort of the DTU National Space Institute and ESA (in cooperation with AWI). The main goal of the project is to quantify and validate CS-2's ability to measure SIT. Since 2003, the CryoVEx missions provides both laser (ALS) and Ku-band radar (ASIRAS) altimetric measurements. In addition, in 2017, the data include for the first time along-track total freeboards from the Ka-band KAREN altimeter (Haas et al., 2017; H. Skourup and Wilkinson, 2019), allowing to 280 compute Ka/Ku snow depth estimations. These data have been developed by AWI in the context of the ESA CryoSeaNICE project. For further informations, please see the technical report at https://earth.esa.int/eogateway/documents/20142/1526226/CryoVEx2017-final-report.pdf.

### 3.3.2 Operation Ice Bridge (OIB)

OIB is one of the largest airborne missions in polar regions aiming to determine sea ice properties. Among others, it carries a 285 snow radar measuring the snow-air and snow-ice interfaces of the signal scattered by the area illuminated beneath the aircraft. All information concerning the different campaigns and instruments can be found in various literature (e.g., Newman et al., 2014; Kurtz et al., 2013; Armitage and Ridout, 2015). OIB snow depth data presented in this paper are the NSIDC OIB Quicklook version (Kurtz et al., 2012; King et al., 2015) validated from in-situ data of the BROMEX campaign (Nghiem et al., 2013). These data are available at https://daacdata.apps.nsidc.org/pub/DATASETS/ICEBRIDGE/Evaluation_Products/.

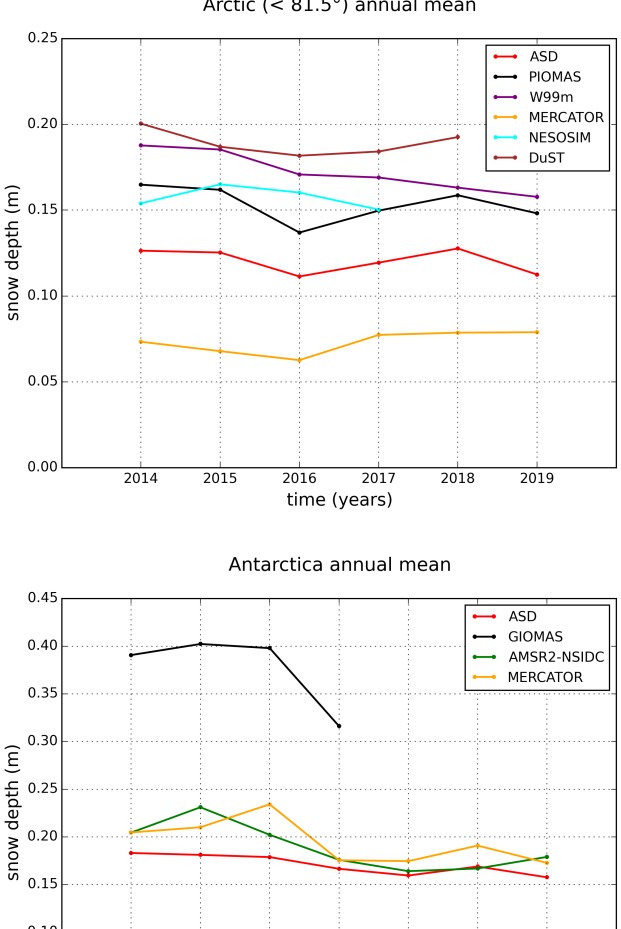

**Figure 2.** Times series of annual mean snow depth of the different products in both hemispheres. Annual means are calculated as the average of the spatial means of all monthly grid maps. Only the 6 winter months are considered in the calculation (November-April in the Arctic and May-October in the Antarctic). Note that the AMSR2 data are not represented in the Arctic since the AMSR2-NSIDC product is only available over FYI and the AMSR2B product is only available with its full spatial coverage (FYI and MYI) in March and April.

For comparison with the ASD dataset, we compare with all spring campaigns during the period of 2014–2018. For comparison with the ASD climatology, we compare with all OIB spring campaigns during 2009–2012.

### 3.3.3 Ice Mass Balance (IMB) buoys

IMB data (Perovich et al., 2021) are issued from the U.S. Army CRREL-Dartmouth Mass Balance Buoy Program. They provide point measurement of the ice mass balance characterised by a combination of snow and ice conditions (Richter-Menge et al., 2006; Perovich and Richter-Menge, 2015). Snow depth data are collected at 4 hours intervals from acoustic sounders. In this study we analyse the data of three winters (2013-2014, 2014-2015 and 2015-2016). Note that we do not use the data of 2017 and 2018 because they are still provisional and subject to revision.

## 4  Comparison between snow depth data

### 4.1  Methodology

Model and satellite snow depth estimations are projected onto a 12.5km pixel size EASE2 grid (similar of ASD) using a linear two-dimensional multivariate interpolation. For each product we compute monthly mean maps for the six winter months (November-April in the Arctic and May-October in the Antarctic) for both hemispheres from March 2013 to October 2019. These maps are provided for 2015 in the supplementary material (Figs. S1 to S13). Note that to improve the consistency of the analyses we always use the larger common time period for all the compared products. Thus, the covered time periods may vary depending on the availability of the snow depth products. Three statistical diagnosis are used to characterise the differences between snow depth datasets:

1) The climatic annual mean, which is simply the average of all monthly snow depth maps from all years.

2) The Mean Annual Variability (MAV), is the average, on the years $y$, of the annual snow depth standard deviation $(std_y(sd))$ maps computed from the 6 monthly mean maps. The MAV is calculated at each grid point following Eq. (6):

$$MAV = \frac{1}{N_y} \sum_y (std_y(sd)) \quad \text{with} \quad std_y(sd) = \sqrt{\frac{1}{N_m} \sum_m (sd_{y,m} - \overline{sd_y})^2}, \tag{6}$$

where $sd_{y,m}$ is the monthly snow depth map of the month $m$ ($m = 11, 12, 01, 02, 03, 04$ in the Arctic and $m = 05, 06, 07, 08, 09, 10$ in the Antarctic) for the year $y$ and $\overline{sd_y}$ is the snow depth annual mean map of the year $y$. $N_m$ and $N_y$ are the number of months (six in our case) and the number of year in the considered time period, respectively.

3) The Mean Inter-annual Variability (MIV), which is the average of the six winter months $m$ inter-annual standard deviation $std_m(sd)$ maps over the 2013-2019 time period. The MIV is calculated at each grid point following Eq. (7):

$$MIV = \frac{1}{N_m} \sum_y (std_m(sd)) \quad \text{with} \quad std_m(sd) = \sqrt{\frac{1}{N_y} \sum_y (sd_{y,m} - \overline{sd_m})^2}, \tag{7}$$

where $\overline{sd_m}$ is average snow depth map of the month $m$.

For now, the lack of available relevant snow depth data in the Antarctic limits these comparisons to the Northern hemisphere. Comparisons with all in-situ and validation data are performed with a comparable methodology: snow depth model and satellite

gridded maps are projected along the aircraft trajectories using a bilinear interpolation. Airborne and in-situ are generally daily data, the comparisons are performed with the mean maps of the month to which that day belongs. In order to achieve similar spatial scales in the comparisons, we have applied a 25 km window rolling mean to smoooth the external data. Due to the non-gaussianities within 25 km sections, we have noticed that the smoothing has a mean effect which slightly tends to reduce the mean value of data. It is important to consider that this allows to correct potential bias that would be induced by a difference

in resolution. Note that the space and time consistency in resolution between all model and satellite data is also ensured by projection onto similar grids. Since snow depth products and validation data are available over different time periods (see table A4), the comparisons are also performed on different time periods. Our approach is to take, for each comparison, the time period common to all the compared data. The aim is to provide reliable statistics. For each comparison, a label specify the considered time period.

## 4.2  Comparison between satellite and model estimations

### 4.2.1  Results in the Arctic

Here, we compare the ASD, DuST and AMSR2B snow depth data in the Arctic by considering the months of March and April over the 2014-2018 period (Fig. 3). These climatic « seasonal » means and inter-annual variability presented in Fig. 3 are the mean and the standard deviation of all monthly maps of March and April over this time period. Statistical diagnosis

are summarised in Table A1. On average, the ASD data are about 3 cm lower than the AMSR2B data and 6 cm lower than the DuST data. ASD data present the stronger snow depth gradients (associated with a higher spatial standard deviation) with clear distinctions between low snow depths over thin sea ice regions (e.g: Laptev sea region) and higher snow depths over regions of thicker sea ice (e.g: Queen Elizabeth islands region). Except in the MYI zones, AMSR2B data are quite smooth and characterised by a weak variability compared to that of the two altimetric products. In MYI zones (towards Queen Elizabeth

islands) AMSR2B has the highest snow depths. We assume that it might be related to the snow depth retrieval algorithm, inadequate over thick ice. Indeed, the statistical results which are presented in the Appendix (Table A1) show that the two Ka/Ku products (ASD and DuST) are in good agreement in terms of spatial variability and the magnitude of their annual cycle considering the entire set of data over the six winter months. Note, that DuST presents the highest climatic « seasonal » mean solution, even higher than W99m (see also Fig. 2). In addition, DuST and ASD spatial distributions are comparable, showcasing

that the difference between the two Ka/Ku altimetric products is likely a bias resulting from the re-calibration of the DusT data with OIB (this point will also be discussed in the next sections). This underlines the consistency of this methodology used to compute bi-frequency snow depth estimations in the absence of external data (ASD), since DuST calibration is limited by the spatial and temporal availability of OIB.

Fig. 4 compare the ASD snow depth estimations with the outputs of three sea ice models in the Arctic (NESOSIM, MER-

CATOR and PIOMAS). The statistical diagnosis presented in Sect. 4.1 are used. They are summarised in Table A1. Amongst the three models, the MERCATOR model provides lower snow depth estimations throughout with a climatic mean of 7.4 cm and lower mean inter-annual and annual variability of about 3cm. On the contrary, climatic means of PIOMAS and NESOSIM

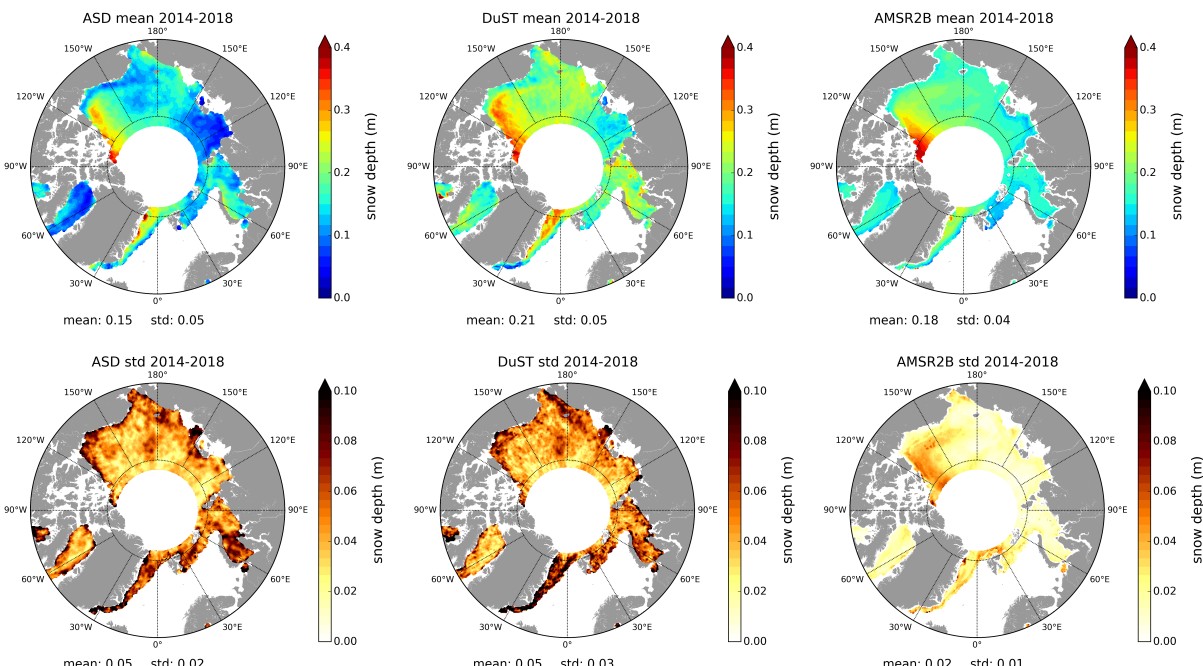

**Figure 3.** Comparisons of satellite snow depth estimations in the Arctic. (Top) Bi-monthly March and April "climatological" snow depth mean maps calculated over the period 2014-2018 for the ASD data (left), the DuST data (center) and the AMSR2B data (right). (Bottom) the standard deviation between all the data used to calculate the mean snow depth maps (equivalent to the Mean Inter-annual Variability computed with March and April over the same 2014-2018 time period).

are similar both in terms of spatial distribution and pan-Arctic mean (with values of respectively 15.5 cm and 16.2 cm). Considering annual and inter-annual variability, PIOMAS presents high values that are more evenly spread. In contrast, NESOSIM

is characterised by lower values everywhere except in the Greenland and the Kara Seas.

In spite of some comparable large scale spatial patterns, these results highlight the large deviations between the ASD data and the model solutions. One striking feature is the transport of MYI along the Canadian coast by the Beaufort Gyre (see also Fig. 1), which is nearly non-existent in model solutions. The patterns of deep snow over MYI are well represented in the models, but do not extend along the Canadian coastline towards the Beaufort Sea, as is the case for ASD. Overall, the models

tend to overestimate snow depths nearly everywhere compared to the ASD data. The ASD data climatic mean is lower (12.3 cm) and zones of thin snow layers (e.g., around 120°E) are less pronounced in models (except by MERCATOR, where snow depth is low everywhere). Models also present higher variability. However, in all datasets the maximum of variability occurs between 0° and 90°E which is relevant because of the proximity to the open ocean and the existence of strong meteorological events (e.g Semenov et al. (2019); Dong et al. (2019)). Although investigating what causes these discrepancies is beyond the

scope of this article, the strong sensitivity of the models to the reanalysis snowfall forcing data (e.g Boisvert et al. (2018); Petty et al. (2018)) could play a predominant role here.

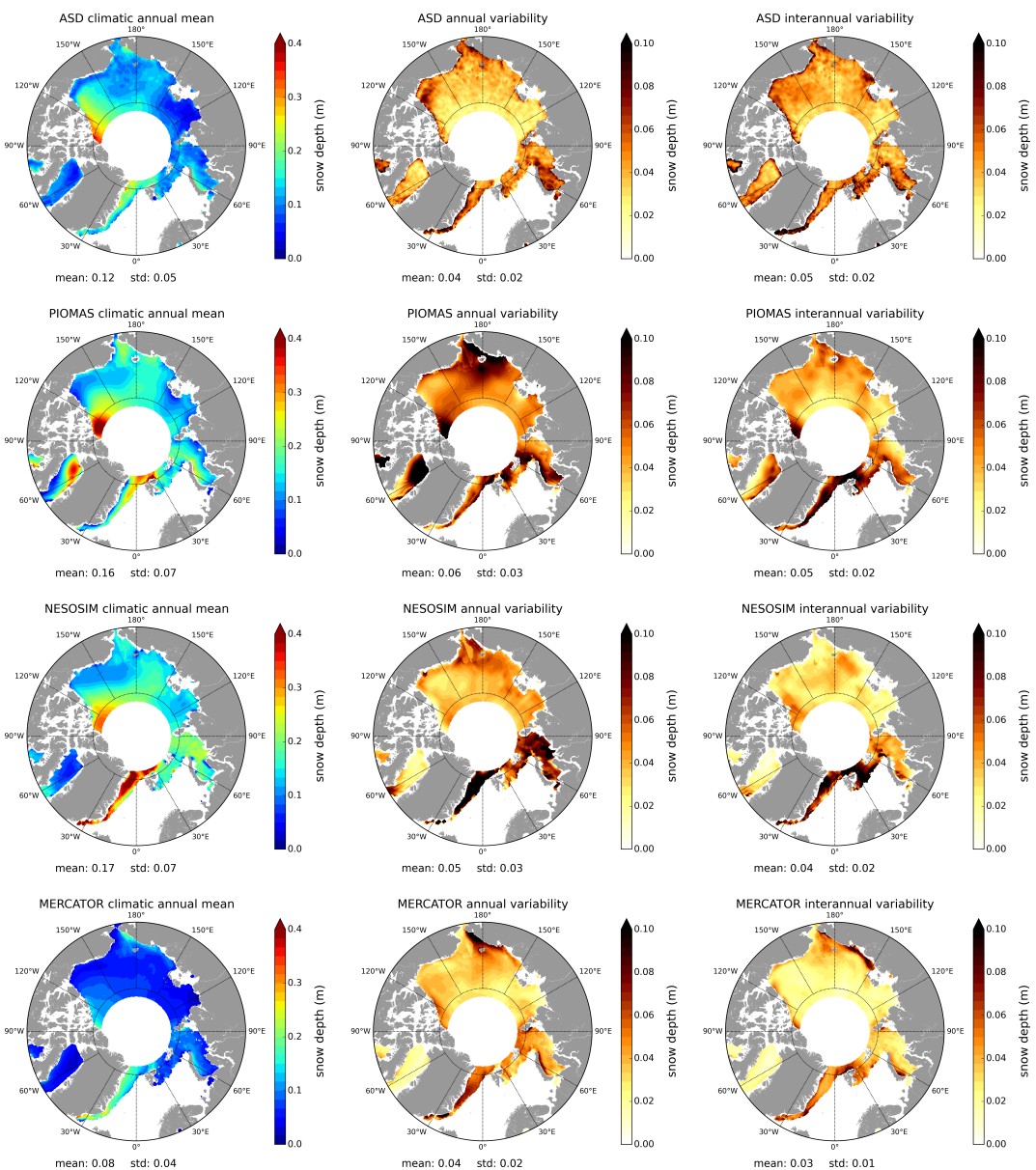

**Figure 4.** Comparisons between the ASD data and model snow depth products (PIOMAS, NESOSIM, MERCATOR) in the Arctic. The first column presents the "climatic mean" snow depth maps, the second column the "annual variability" and the third column the "interannual variability" as presented in Sect. 4.1. These diagnosis are calculated from March 2013 to April 2019 for ASD (first row), PIOMAS (second row) and MERCATOR (fourth row) and from March 2013 to April 2017 for NESOSIM (third row).

### 4.2.2 Results in the Antarctic

ASD is currently the only publicly available altimetric snow depth product in the Antarctic. One major advantage compared to the Arctic, is that the SARAL orbit does not affect the data coverage since all the Antarctic sea ice is below 81.5°S. Similarly to what has been done in the previous section for the Arctic, Fig. 5 compare ASD with the AMSR2-NSIDC data and the GIOMAS and the MERCATOR sea ice model. The statistics are summarised in Table A2.

In spite of a comparable climatic mean and some coherent spatial distributions such as the transport of snow in the Weddell gyre, ASD and AMSR2-NSIDC data show significant discrepancies. Unlike what we have observed in the Arctic, ASD data are relatively smooth with a weak spatial variability. In comparison, AMSR2-NSIDC data are characterised by strong contrasts between large snow depth patterns in the Weddell sea and to a less extent in the western part (from 150° E to 90° W) and areas with nearly no snow (eastern part, between 0° E and 150° E). As in the Arctic, AMSR2-NSIDC annual and inter-annual variability are lower and much more localised than in the ASD data. ASD and AMSR2-NSIDC well reproduce the thin snow pattern of the Ross Ice shelf. One other relevant difference between these two datasets is the systematic decrease of snow depth in October in the AMSR2-NSIDC data (see Table A2).

The presence of thicker snow in East Antarctic for ASD is consistent with Worby et al. (2008b) which comparisons with ARISE in-situ data have shown radiometric measurements snow depth underestimations. Wet-snow conditions due to flooding might be responsible for radiometric brightness temperature contrasts. The low variabilities in the Weddell sea for ASD were not expected since winter snow properties are extremely variable (Massom et al., 1997). In addition the large bias with AMSR2-NSIDC raises questions. Since Ku-band is supposed to better penetrate in cold and dry snow (Willatt et al., 2011), one hypothesis is that the presence of saline and warm moist basal snow layers, even in the absence of flooding, (Massom et al., 2001; Perovich and Richter-Menge, 1994) lead to ASD underestimations. For AMSR2-NSIDC, the snow depth retrieval algorithm is probably not well adapted for rougher snow that can be compared to MYI in the Arctic.

With a climatic spatial mean of about 34.6 cm, GIOMAS simulates everywhere very high snow depths compare to the other solutions. Considering what is provided in the literature (e.g., Worby et al., 2011), these values clearly seem to be overestimated. In the exception of the Weddell Sea, the Mercator model and ASD present comparable snow patterns with substantial coastal values not seen in the AMSR2-NSIDC data. However, the MERCATOR model displays thicker snow everywhere but in the Weddell sea while the presence of deep snow is a well observed feature, which is supported by ASD (e.g., Massom et al., 1997; Eicken et al., 1994). Because of highly dynamics weather conditions, with persistent strong winds in the Antarctic, snow thickness distribution is not directly related to snowfalls (Massom et al., 2001). Conversion from precipitation to snow depth is then very different to the Arctic and Antarctic snow pack is not an uniform slab. These features should partly explain model difficulties and differences between the two hemispheres.

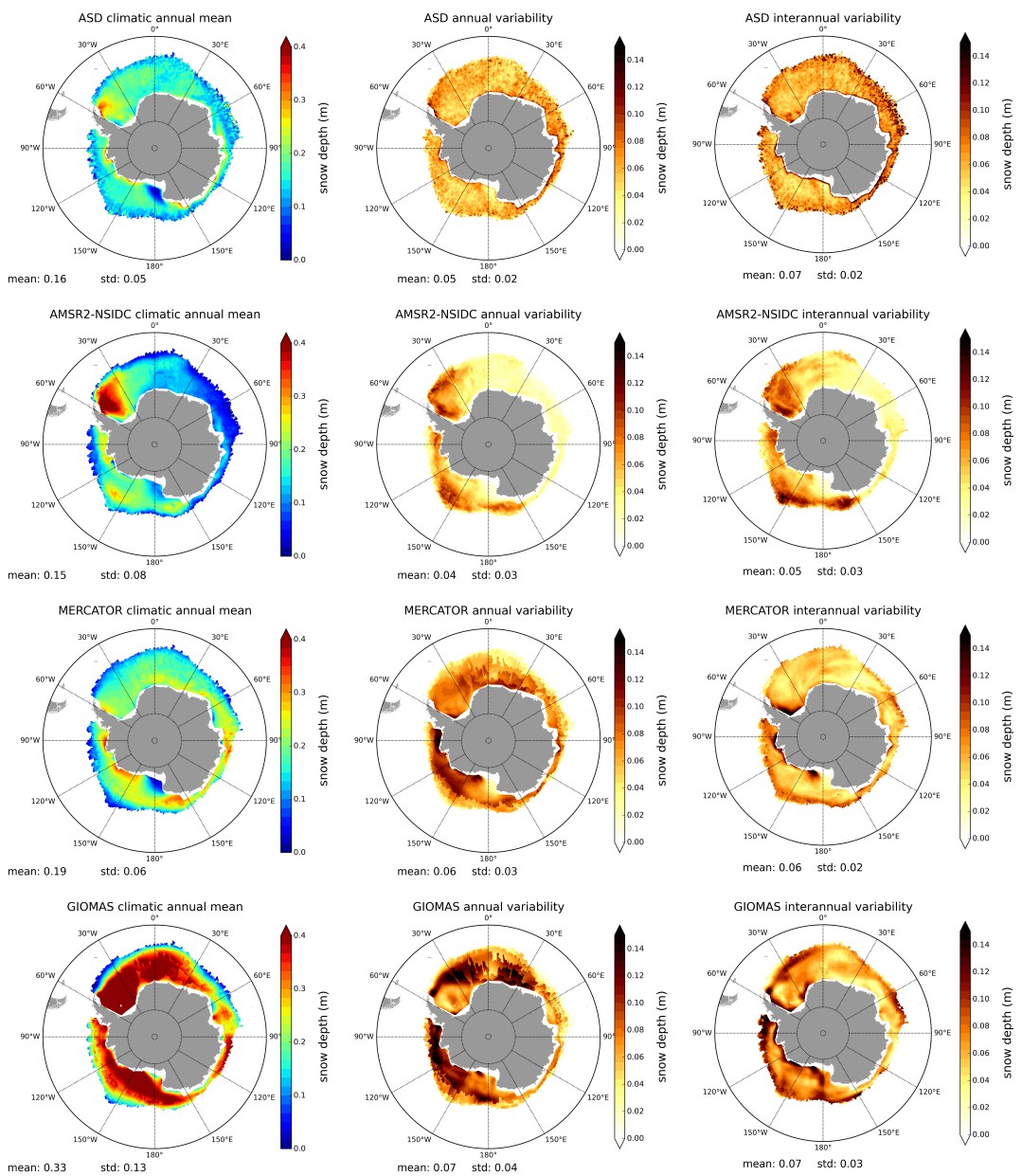

**Figure 5.** Comparisons between ASD data and other snow depth estimations in the Antarctic (AMSR2-NSIDC, MERCATOR, GIOMAS). The first column presents the "climatic mean" snow depth maps, the second column the "annual variability" and the third column the "interannual variability" as presented in Sect. 4.1. These diagnosis are calculated from May 2013 to October 2019 for the ASD data (first row), the AMSR2 NSIDC (second row) and the MERCATOR model (third row) and from May 2013 to October 2016 for the GIOMAS model data (fourth row).

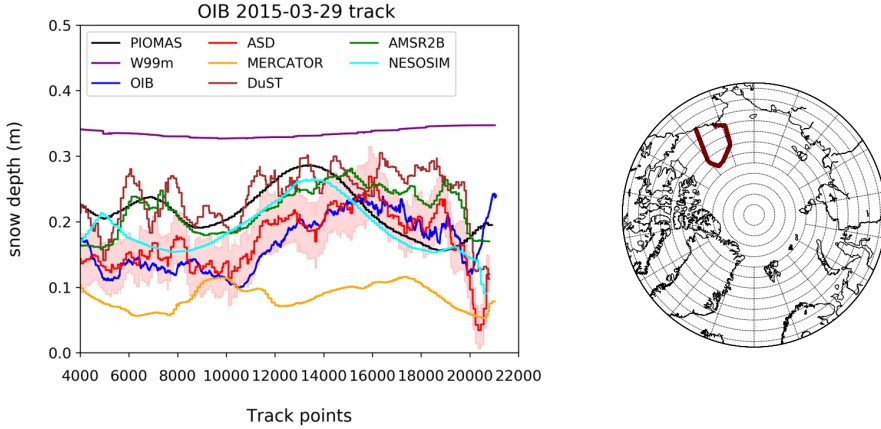

**Figure 6.** Along-track comparison between the snow depth products and the OIB snow radar data acquired on 29[th] March 2015 in the Arctic. The red envelope refers to the ASD uncertainties. The geographical map specifies the geographical location of this particular one-day OIB track.

## 4.3 Comparison with in situ and airborne data

### 4.3.1 OIB

OIB data are compared to all the datasets presented in the previous sections. We only consider OIB observations below 81.5°
N from the 2014-2018 campaigns. Fig. 6 shows an along track example from the OIB flight on 29[th] March 2015 track and Fig. 7 presents scatter plots comparing the various snow depth products with the OIB data over the entire 2014-2018 time period (including all spring OIB campaigns within this period). Statistical results are summarised in Table 1.

One striking feature is the good consistency between ASD and OIB in terms of magnitude and spatial variability. The OIB data are almost within the ASD envelope of uncertainties (in shaded red) at all times. These consistencies between OIB and ASD data are also demonstrated in several other OIB tracks provided in the supplementary material (Figs. S14 to S17). The DuST Ka/Ku data present an along-track variability similar to ASD, but tend to overestimate snow depth compared to OIB. This reinforces the hypothesis that the main difference between ASD and DuST is a bias. The AMSR2B data also reproduce the large scale snow depth variability quite well but overestimate at a level comparable with DuST, when compared to OIB. The DuST and AMSR2B overestimations are quite consistent with Kwok et al. (2017), who found that this OIB product tend to have thinner snow than the NOAA Wavelet Airborne Snow Radar dataset, which has been used for re-calibrations. In fact, it is important to be aware that various OIB datasets have been produced, leading to a variety of snow estimations that can reach 7 cm on FYI and 12 cm on MYI (Kwok et al., 2017). The variabilities of snow properties from OIB daily basis data to the monthly means of the other datasets certainly explain some of the deviations.

Models hardly represent the along-track variability and do not reproduce small scale patterns. A too coarse resolution could partly explain this feature. As expected, the MERCATOR model snow thickness are far below the others, whereas he PI-

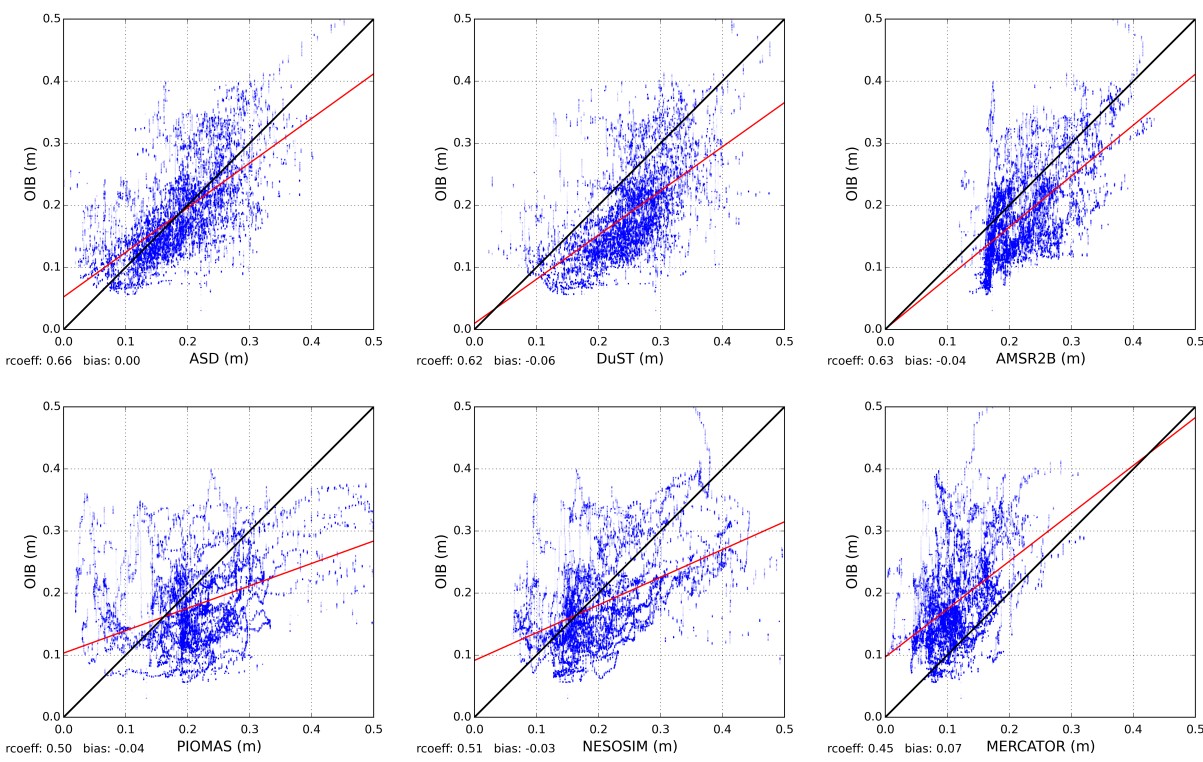

**Figure 7.** Scatter plots between the snow depth products and OIB snow radar data considering the 2014 to 2018 missions. The red line is the linear fitting line. For NESOSIM, 2018 is not included.

| | OIB | ASD | DuST | PIOMAS | MERCATOR | NESOSIM | W99m | AMSR2B |
|---|---|---|---|---|---|---|---|---|
| mean (cm) | 18.6 | 18.5 | 24.8 | 22.9 | 11.5 | 21.1 | 33.9 | 22.6 |
| STD (cm) | 7.5 | 6.9 | 6.5 | 10.4 | 4.4 | 8.5 | 1.5 | 5.8 |
| mean bias (cm) | – | ≈0.0 | -6.2 | -4.3 | 7.1 | -2.6 | -15.03 | -4.0 |
| RMSE (cm) | – | 5.9 | 8.7 | 10.2 | 9.8 | 8.4 | 17.0 | 7.1 |
| R | – | 0.66 | 0.62 | 0.50 | 0.45 | 0.51 | 0.16 | 0.63 |

**Table 1.** Statistics of the comparison between OIB snow radar data and the various snow depth products considering the OIB spring campaigns of 2014–2018.

OMAS and NESOSIM solutions provide comparable results (see also Table 1). As expected, the W99m climatology strongly overestimates snow depths and is clearly a less optimal solution in this case.

| | IMB | ASD | DuST | PIOMAS | MERCATOR | NSIM | W99m | AMSR2B |
|---|---|---|---|---|---|---|---|---|
| Mean (cm) | 21.6 | 19.14 | 25.51 | 19.37 | 7.61 | 18.32 | 28.03 | 24.88 |
| STD (cm) | 2.2 | 3.17 | 2.90 | 4.68 | 2.17 | 3.68 | 5.16 | 1.05 |
| RMSE (cm) | – | 4.73 | 5.50 | 5.11 | 14.32 | 4.96 | 8.2 | 3.71 |

**Table 2.** Statistics of the comparison with IMB data and the various snow depth products considering data acquired during 2013–2016..

### 4.3.2 IMB

The comparison between snow depth estimations and the IMB data for the winters 2013-2014, 2014-2015 and 2015-2016 is
presented in Fig. 8. Statistical results are summarised in Table 2. Except for AMSR2B, which is only available for the months of March and April, all comparisons are performed using the exact same spatial coverage considering only data below 81.5°N. Note that only very limited IMB data are available south of 81.5°N for the winter 2015-2016.

Fig. 8 and Table 2 show how the satellite derived mean values are closer to the in-situ observations than the model. However, the seasonal changes captured in the IMBs are not well reproduced by the satellite products nor the models. This could reflect
sampling differences as the IMB observations consist of hourly point data, that are not representative of the wider averages of the gridded satellite or model products. DuST and ASD along-track variability are once again very similar with DuST being overall higher, highlighting the very likely bias between these two products.

### 4.3.3 CryoVEx

Fig. 9 presents the snow depth products projected along the CryoVEx flight track measured on the 31 March 2017. This figure
gathers the CryoVEx snow depth estimations calculated from (a) the difference between the Ku-band ASIRAS radar freeboard and the ALS laser (ALS/ASR) total freeboard, and (b) from the Ku-band ASIRAS radar freeboard and the Karen (Ka/ASR) total freeboard. Note that because of SARAL spatial coverage limitation (latitude <81.5 °) and the absence of the SARIN mode in the CS-2 ESA Baseline B GOP product (used to construct the ASD data, see Sect. 2), only the track of the 31 March 2017 CryoVEx survey of the Baffin Bay can be compared with the ASD data.

Ka/ASR estimation exhibits very thin snow thickness (< 10cm). Although it might be expected in this area (Landy et al., 2017), this solution still contains unrealistic negative values due to the fact that ASIRAS and KAREN freeboards are nearly equivalent over FYI (without negative values). The main difference between snow depths calculated from the laser and the Ka-band radar snow depth estimations seems to be a bias. Except around 71.3°N, where a specific event may have occurred, the ASD snow product tracks the magnitude of the CryoVEx airborne data with ALS/ASR and KAREN/ASR acting respectively
as an upper and lower bound and within ASD range of uncertainty (in red shading). However, the along-track variability is not well reproduced by any dataset. The MERCATOR and NESOSIM models also show similar magnitudes as the CryoVEx snow depth estimations (ALS/ASR and Ka/ASR), while PIOMAS and the W99m climatology clearly overestimate. AMSR2B and DuST overestimate snow depth, while the DuST along-track variability is comparable to ASD.

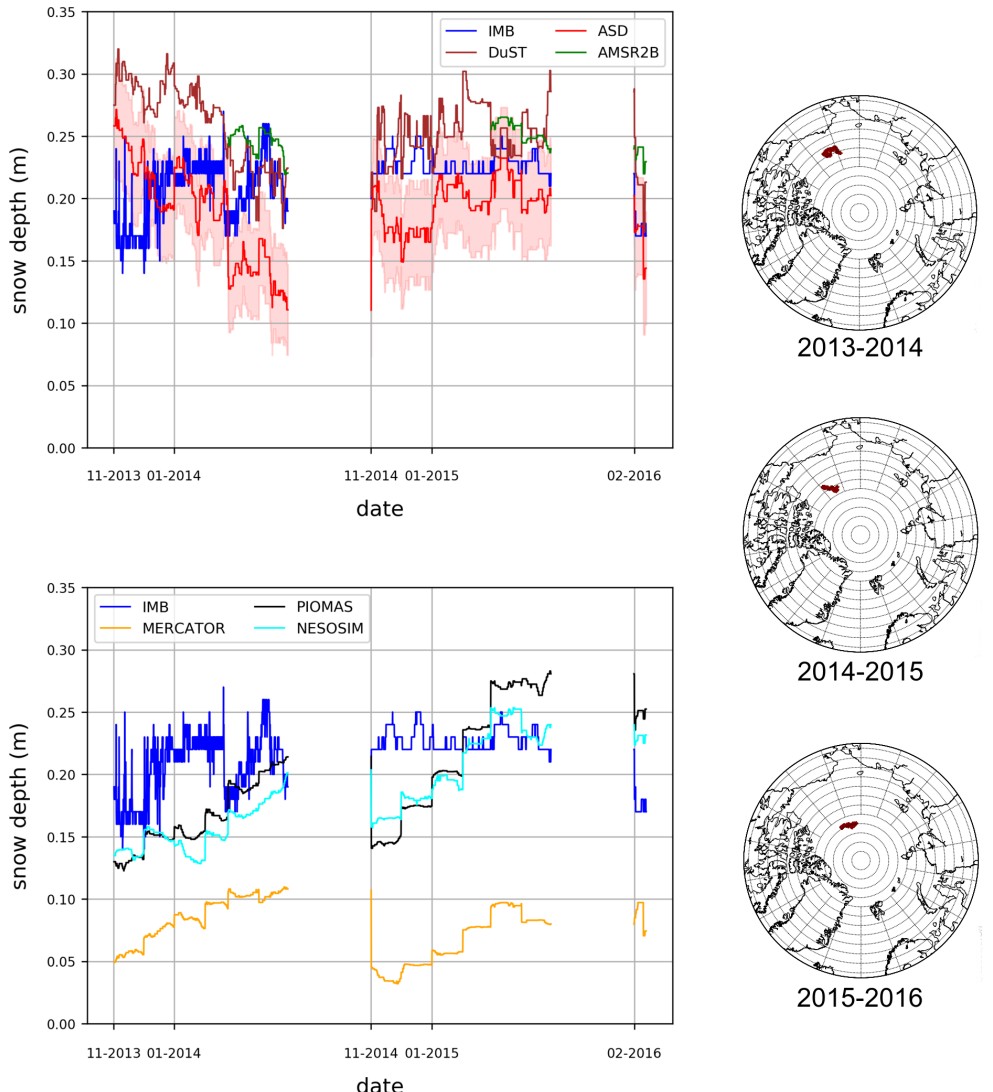

**Figure 8.** Along-track comparison between the snow depth products and the IMB in-situ data. (Top left) IMB buoys compared with satellite data; both passive and active, (bottom left) IMB buoys compared with model data. Right column shows geographical maps from data for all three periods (2013–2014, 2014–2015, 2015–2016). The maps presents the geographical location of the three IMB buoys.

## 5 Towards an ASD snow depth climatology

In spite of the consistency of the ASD data highlighted in the previous sections from comparisons with airborne and in-situ data, the the limited temporal coverage (only available post-2013)) remains an important limitation to the use of these data. Considering the low inter-annual snow precipitation variability compared to that of the spatial variability (e.g Figs. 2 and 4), one solution would be to develop an ASD climatology. For that purpose, we have constructed a preliminary altimetric snow depth

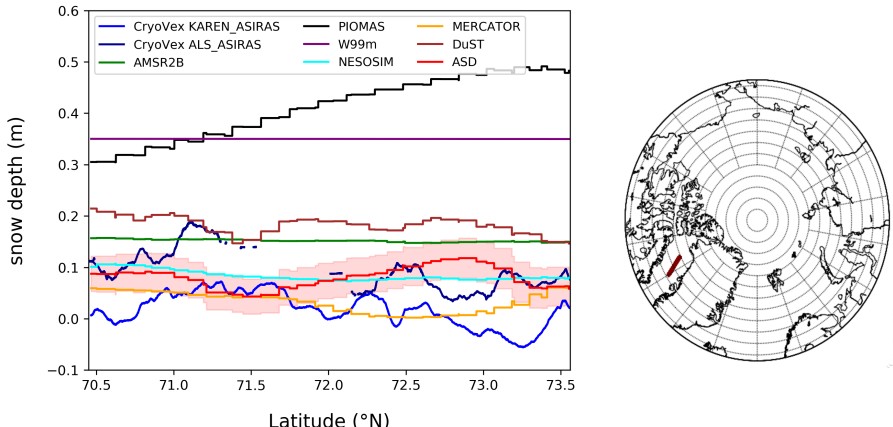

**Figure 9.** Along-track comparison, in the Arctic, between the snow depth products (models and satellites) and the CryoVEx snow depth data calculated from the Ka-band KAREN radar (blue) and the ALS laser (dark blue) with the Ku-band ASIRAS radar during the 31[st] March 2017 campaign. The map specifies the geographical location of the CryoVEx track.

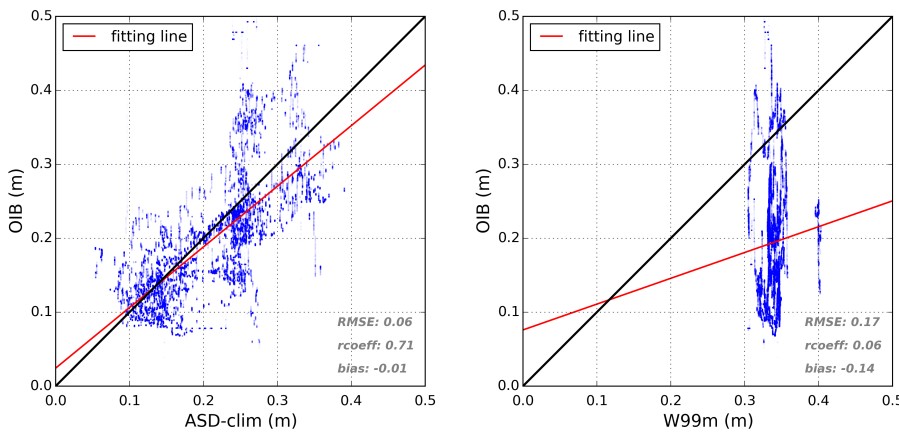

**Figure 10.** Scatterplots comparing the ASD and the W99m climatologies with OIB by considering all the tracks of 2009, 2010, 2011 and 2012 OIB missions.

climatology in the Arctic by averaging all the ASD snow depth maps of each month during the 2013-2019 period (designated
as the ASD-clim). An equivalent climatology could also be constructed for the Antarctic but the lack of validation prevents its validity to be demonstrated at present. To demonstrate the relevance of this climatology in the Arctic for the years prior to 2013, the ASD-clim data are projected on all the tracks of the four OIB missions occurring between 2009 and 2012, and presented in Fig. 10.

    Fig. 10 shows that the ASD climatology would be a better solution than W99m, at least for the CS-2 time period. This is
highlighted with the lower bias (-0.1 m compared with -0.14), the higher correlation coefficients (0.71 compared with 0.06),

and the lower RMSE (0.06 m compared with 0.17 m) . Furthermore, even as a climatology, we obtained a correlation between the ASD-clim and OIB of the same order of magnitude as obtained by direct correlation between ADS and OIB measurements (Fig. 7). However, as future work, a more refined computation should be investigated. In particular, an important point would be to provide a relevant snow product representing the Envisat-era. Since the W99m can be considered as relatively consistent until the year of 2000, a mixed ASD-W99m product, including for instance a time dependency could be considered. Also, it could be relevant to take into account the sea ice type inter-annual variability as it is done for W99m.

Another critical point of the ASD data (for instance for sea ice volume estimations) is the reduced spatial coverage due to the SARAL orbit. While during the time period of Envisat, this does not become an issue since its orbit is equivalent to that of SARAL, the absence of data north of 81.5°N is a major limitation to SIT estimations. A combination of ASD with W99m and/or model data could be done to extrapolate the data. Potentially, one could also investigate a combination of ASD with W99m and/or models to extrapolate the data. In the Antarctic, we do not encounter the same issue since the orbit does not reduce the spatial coverage and FYI is generally considered to be the only sea ice type present in the Southern Ocean. Then, such a climatology would be a solution to compute SIT estimations over the Envisat/CS-2 time period provided that the temporal variability of precipitations are investigated. Note, that experimental SIT estimations (Hendricks et al., 2018) have already been done in the Antarctic in the framework of the Sea Ice Climate Change Initiative (SI-CCI). These data used a snow climatology elaborated from AMSR-E and AMSR2 data in a nearly equivalent manner to what has been presented here.

## 6  Impacts of snow depth on SIT estimation

Since there is still no validated freeboard product in the Antarctic, this analysis has only been performed in the Arctic. From the CS-2 radar freeboard product presented in Guerreiro et al. (2017), extended to 2019, we have computed SIT estimates using the various snow depth datasets presented in this study. It is calculated from the CS-2 L1b ESA Baseline C SAR waveforms (Bouffard et al., 2018b) with the methodology presented in Sect. 2. Freeboard heights are converted into SIT assuming hydrostatic equilibrium between snow covered sea ice and the ocean (Eq. (8))

$$SIT = \frac{\rho_w fb_{ice} + \rho_s sd}{\rho_w - \rho_i} \tag{8}$$

As in Ricker et al. (2014), we assume the sea water density ($\rho_w$) is set to 1024 $kg/m^3$. Consistently with the approach of Laxon et al. (2013), sea ice density ($\rho_i$) is set to 882 $kg/m^3$ for MYI and 917 $kg/m^3$ for FYI (Alexandrov et al., 2010). Furthermore, the snow density ($\rho_s$) is set to 300 $kg/m^3$. Distinction between FYI and MYI follows the EUMETSAT OSI SAF sea ice type classification. The sea ice freeboard $fb_{ice}$ is calculated from the radar freeboards, taking into account the reduction of radar velocity within the snow layer, which depends on snow depth data $sd$ following Eq. (3). SIT monthly maps are calculated for each snow depth product presented in previous sections. These monthly maps are provided for the winter of 2015 in the supplementary material (Figs. S18 to S24). Fig. 11 presents the annual mean time series of the spatial mean of all these SIT solutions (except for AMSR2B which is only calculated in March and April).

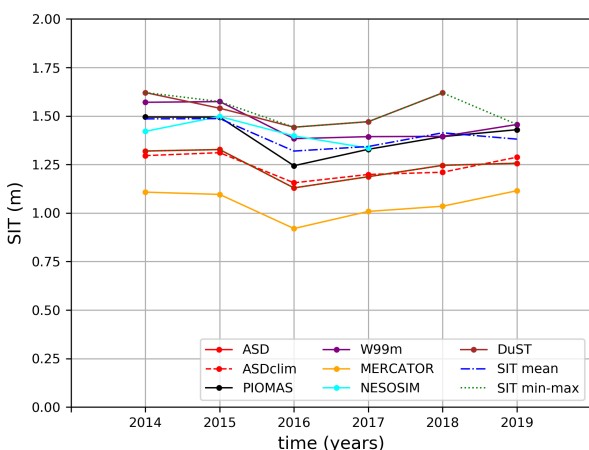

**Figure 11.** Arctic annual mean time series (2014-2019) of the averaged SIT solutions computed using various snow depth products (models, and active/passive satellite-based). Spatial coverage is defined by the smallest sea ice extent. The dotted blue lines define the annual mean, minimum and maximum values between all solutions (except for MERCATOR). The AMSR2 data are not represented since they are not available over the period.

Except for the MERCATOR model (which has shown to be overall significantly low), using the ASD data provide the lowest pan-Arctic mean SIT estimations. Highest SIT estimations are obtained with the DuST product (and W99m), highlighting the impact of the bias previously identified between these two solutions (ASD and DuST). Considering the various snow products together (except for the MERCATOR model), the spatial mean SIT annual mean values are distributed between 1.2 m and 1.6 m. This leads to a pan-Arctic SIT ensemble mean of 1.4 m (mean of the dotted blue line in Fig. 11) associated with a mean maximum deviation of 29 cm (mean of differences between the maximum and minimum blue dotted lines). This mean maximum deviation, which is approximately the mean difference between ASD and DuST, represents about 20% of the mean SIT. Considering the good consistency between ASD and the validation snow depth data (demonstrated in Sect. 4.3), actual SIT products computed using W99m should tend to significantly overestimate sea ice volume in the Arctic. For instance, taking a deviation of about 0.3 m ($\approx$ the mean deviation between ASD and W99m in 2015) and a mean Arctic sea ice extent of 13 millions of km$^2$ (approximately the mean sea ice extent of 2015), the resulting SIT overestimation would lead to an sea ice volume excess of about $4 \times 10^3$km$^3$, which represents about $3.5 \times 10^{15}$ litres of freshwater. Such an uncertainty is not negligible compared to the 8000$\pm$2000 km$^3$ change in the Western Arctic freshwater content between 1995-1996 and 2009-2010 relayed in Giles et al. (2012). To give an illustrating example, this uncertainty of $3.5 \times 10^{15}$ litres of freshwater would be roughly equivalent to six months of the Amazon water discharge (considering a debit of about $2 \times 10^8$ litres/seconds). Deviations of only a few centimetres of snow depth ($\approx$ 6 cm of mean deviation between ASD and W99m snow depths in 2015, see Fig. 1) have strong impacts on the freshwater budget.

Although this provides a relevant overview, such large scale pan-Arctic means do not show the spatial and temporal variabil-
ity of the various SIT solutions. For instance, while DuST and W99m provide comparable SIT pan-Arctic mean estimations,
the spatial and temporal information included in these two datasets are very different (e.g. Fig. 2 and 3). To better charac-
terise the impacts of snow depth on SIT, we propose to consider the deviations between these SIT estimates for each month
of each year and at each grid point. Such an ensemble of SIT solutions will allow to characterise the level of uncertainty due
to discrepancies between the different snow depths. For the month $m$ of year $y$, the standard deviation ($std_{y,m}(SIT)$) and the
maximum deviation ($maxdev_{y,m}(SIT)$) maps are calculated with Eqs. (9) and (10). $SIT_{y,m,p}$ is the SIT solution provided by
the snow product $p$ and $\overline{SIT_{y,m}}$ is the mean map between all these SIT estimations. $N_p$ is the number of snow depth products.
$\max_p(SIT_{y,m,p})$ and $\min_p(SIT_{y,m,p})$ are respectively the maximum and minimum monthly maps between the SIT solution
of the snow depth products.

$$std_{y,m}(SIT) = \sqrt{\frac{1}{N_p} \sum_p (SIT_{y,m,p} - \overline{SIT_{y,m}})^2}, \tag{9}$$

$$maxdev_{y,m}(SIT) = \max_p(SIT_{y,m,p}) - \min_p(SIT_{y,m,p}). \tag{10}$$

For each winter month, inter-annual mean, minimum and maximum maps of these two statistics (standard deviation $std_{y,m}(SIT)$
and the maximum deviation $maxdev_{y,m}(SIT)$) can then be computed from Eqs. (11) to (16) where $N_y$ is the number of years.

$$\overline{std_m} = \frac{1}{N_y} \sum_y (std_{y,m}(SIT)) \tag{11}$$

$$\overline{maxdev_m} = \frac{1}{N_y} \sum_y (maxdev_{y,m}(SIT)) \tag{12}$$

$$max(std_m) = \max_y(std_{y,m}(SIT)) \tag{13}$$

$$max(maxdev_m) = \max_y(maxdev_{y,m}(SIT)) \tag{14}$$

$$min(std_m) = \min_y(std_{y,m}(SIT)) \tag{15}$$

$$min(maxdev_m) = \min_y(maxdev_{y,m}(SIT)) \tag{16}$$

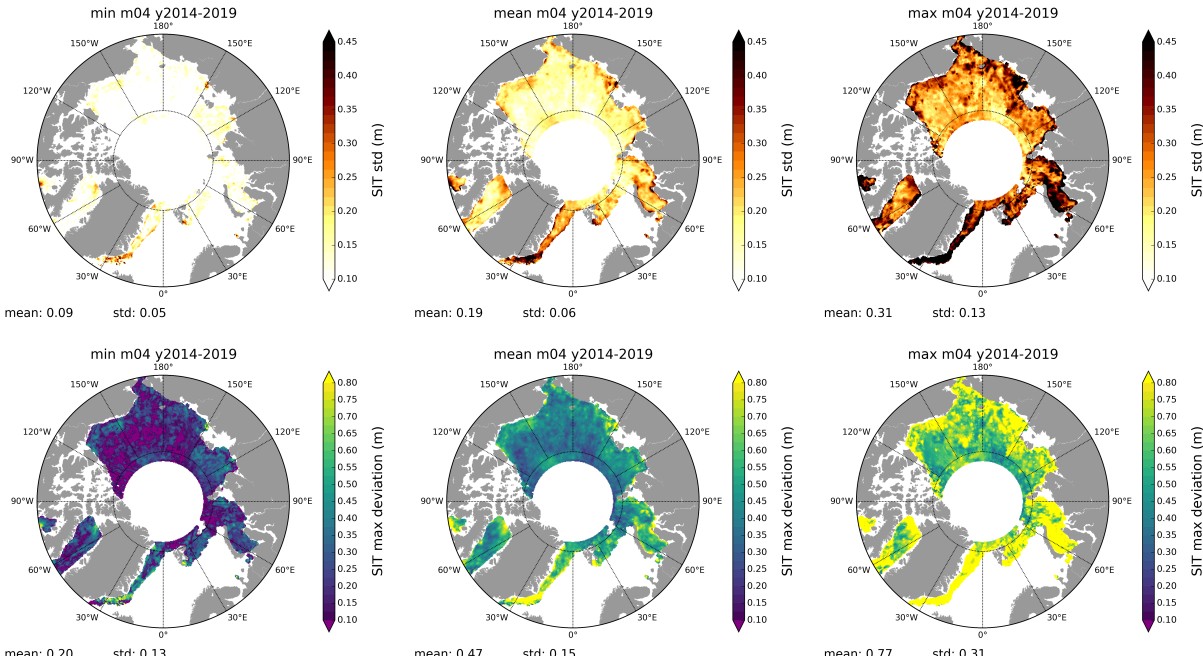

**Figure 12.** Inter-annual minimum ($min(std_m)$, first column), mean ($\overline{std_m}$, second column) and maximum ($max(std_m)$, third column) maps of the standard deviation (first row) and the maximum deviation (second row) for the month of April ($m = 04$). The snow products used to compute the maps are ASD, DuST, AMSR2B and W99m. These maps correspond to the case « obs snow products ».

These statistics quantifies the SIT uncertainty, caused only to snow depth, assuming that deviations between snow depth
solutions refer to the snow depth level of uncertainty. We define two cases: 1) « obs snow products », which considers ASD, DuST, W99m and AMSR2B and 2) « all snow products », which includes in addition the PIOMAS, MERCATOR and NE-SOSIM model solutions. Note, that ASD-clim is not used in calculations in order to avoid the redundancy with ASD data. Considering « obs snow products », Fig. 12 presents maps of the metrics presented in Eqs. (11) to (16) for April. In addition, the monthly minimum, mean and maximum maps over the period 2014-2019 are presented for April in the supplementary
material (Fig. S25). All the results are summarised in Table A3 which presents, for each winter month, the spatial mean value of these maps. The spatial mean of the monthly mean SIT maps ($\overline{SIT_{y,m,p}}$) are also included.

Highest impacts on SIT are generally located in near coastal zones and marginal ice zones. In these areas, maximum standard deviation can reach up to 50 cm and maximum deviation to 80 cm when considering « obs snow products ». Due to larger differences between model and observations, in particular MERCATOR, these values are higher with « all snow products »
(see Table A3). Except for these specific locations, that mostly account for maxima values, the standard deviation maps are rather uniform. This suggests that spatial mean values (presented in Table A3) are a relevant indicator.

The large deviations between minimum and maximum (standard deviation and/or maximum deviation) maps indicate significant variations of the SIT level uncertainty due to snow depth variability from one year to another. On the contrary, spatial mean

standard deviations do not vary significantly in between months. Considering « obs snow products » and all winter months, we obtained a SIT mean standard deviation of $\approx 20$ cm (14%) $\pm 11$ cm ($31 - 9$cm interval, see Table A3) and mean maximum deviation of $\approx 49$ cm (35%) $\pm 25$ cm. These values increase to $\approx 27$ cm (20%) $\pm 9$ cm for the standard deviation and $\approx 76$ cm (58%) $\pm 25$ cm for the maximum deviation when we take into account « all snow products ». Therefore, compared to the deviation (roughly 30 cm) that can be extract from the difference between ASD (red line) and the SIT max line (dotted blue) in Fig. 11, we obtain higher maximum deviations using this methodology. It suggests, that considering large scale global means could tend to underestimate impacts of snow depth on SIT, and therefore the SIT uncertainty. Note, that in addition to a more refined methodology, taking into account AMSR2B, which can reach high values, might have a small impact here as well.

Besides providing a relevant quantification of the impacts of snow depth on SIT, such « multi-observation » approach also provide a space and time variability of uncertainties. It also avoids the necessity of relying on the strong, and usual assumption, of Gaussian propagation of de-correlated errors. These uncertainty estimates are very important for data assimilation, in which the characterisation of observation error is of main importance to constrain models (e.g., Stewart et al., 2008; Bunzel et al., 2016; Janjić et al., 2018). Note, that there are a growing number of research programs and international initiatives which aims at coordinating efforts for better characterising uncertainties and provide traceable quality indicators to sea ice EO products (e.g., World Meteorological Organisation (WMO) Workshop at AWI, European Union/European Commission (EU/EC) Polar Science Week).

## 7 Discussions and conclusions

In this study, we have presented a Ka/Ku snow depth product computed, in both hemispheres, from the data obtained by altimeters on board the satellites SARAL (Ka-band) and CS-2 (Ku-band). This product has been compared over the period 2013-2019 with: the other Ka/Ku snow depth product (DuST, only in the Arctic), AMSR2 passive radiometer snow depth estimations, W99m (only in the Arctic) and snow depth solutions of the models PIOMAS (GIOMAS in the Antarctic), MER-CATOR and NESOSIM (only in the Arctic). Thereafter, all these products have been evaluated, in the Arctic, against OIB airborne data, IMB snow buoys and CryoVEx airborne campaign data, which includes Ka/Ku and ALS/Ku snow depths. In the Antarctic, the lack of validation data remains a major obstacle to the assessment of sea ice products. It is currently a primary focus of the ESA CSAO+ project.

In the Arctic, we observed a good agreement between the two Ka/Ku products (ASD and DuST) in terms of spatial distributions and annual and inter-annual variability but DuST is always biased high of about $6.5 \pm 0.5$ cm (see Table A1). Although this is likely due to the re-calibration of the DuST product with OIB data mainly measuring thick snow, this bias probably also indicate variabilities of Ku-band penetration related to snow properties. Only the surface roughness is considered in the ASD calculation but impact of other snow properties should be analysed and considered.

The AMSR2B (and also AMSR2-NSIDC over FYI) data also tend to overestimate snow depths compared to ASD in the Arctic. This overestimation is lower than DuST, in average of about 3 cm, and does not seem to only be a bias since AMSR2B

spatial patterns present noticeable differences. We reiterate that comparisons with AMSR2B have only been conducted in March and April because the data are not available over MYI for the other months. This absence of variability except in the MYI zone where AMSR2B have been re-calibrated is a surprising feature that need to be considered when using these data.

Comparisons with models in the Arctic have highlighted that the MERCATOR model is always biased low when compared to the other products and simulates limited snow depth variability. This is mainly due to the use of the ERA-interim forcing fields that underestimate precipitations. The impacts of other parameters such as the albedo or the amount of precipitation falling into the ocean should be investigated. In contrast, NESOSIM and PIOMAS spatial patterns consistent with ASD and a bias high on average of a few centimeters. We found comparable representations in PIOMAS and NESOSIM although NESOSIM presents

higher snow depths over MYI and higher variability. In spite of consistencies, these two models hardly represent the MYI sea ice drift in the Beaufort Gyre that can be observed in satellite observations. Note that we have not considered in this study the recent SnowModel-LG model, that should consistently capture the Arctic snow depth over sea ice spatial variability (Stroeve et al., 2020; Liston et al., 2020). We refer to (Zhou et al., 2020a) for a comparison between the SnowModel-LG and various snow depth observation products.

In evaluation and for comparison of the snow depth products, we have compared with OIB, CryoVEx and in-situ IMB snow depths over the Arctic. Comparisons with OIB have shown satisfying consistencies with the ASD estimations. We found a RMSE of about 6 cm with a correlation coefficient of 0.66. By comparison, the DuST product has a RMSE of 8.7 cm and AMSR2B of 7 cm. Note that this higher RMSE for DuST, in spite of a re-calibration, probably comes from the OIB version used for re-calibrating. Indeed, DuST is re-calibrated against the OIB NOAA Wavelet Airborne Snow Radar dataset which

is expected to present higher snow depth values. In addition, further analysis could be done in order to link the space and time variability of these bias with snow properties The various snow depth products are less consistent when comparing with IMB, although ASD mean values present consistent order of magnitude. Due to the data coverage, validation with CryoVEx has been limited to one single track (where both Ku-band radar (ASIRAS), Ka-band radar (KAREN) and lidar scanner (ALS) were available). We observed, that KAREN-ASIRAS snow depth estimations are low, including a non-negligible amount of

negative values. ASD data always remains in between the laser-Ku (ALS-ASR) and Ka/Ku (Ka/ASR) CryoVEx snow depth estimations.

    Since the results of such comparisons with validation data can vary from one methodology to another (e.g., grid sizes, smoothing kernel, dataset versions), they do not aim to assess the best snow depth product. However, they demonstrate how ASD provides a relevant snow depth solution, in good agreement with several validation data, and that ASD allows for char-

acterisation the deviations between the different snow depth products. A more refined comparison with the CryoVEx data (including various tracks) is mandatory to understand the relative impacts of roughness and penetration and their link with snow properties. For instance, Willatt et al. (2011) show that the Ku-band dominant scattering surface can significantly vary with snow temperature, with a reduced penetration when temperatures increase. This feature could enhance an underestimation of ASD data with the warming of temperatures due to climate change. It also points out the difficulty to retrieve snow depth

from altimetry beyond the 6 months of winter as variations in snow properties are most important in the summer period.

In the Antarctic, for satellite-based observations, spatial mean snow depth values of AMSR2-NSIDC and ASD are quite close. However, we found that AMSR2-NSIDC estimations are very low in zones of thin snow while it tends to strongly overestimate snow depths in areas of thick ice. Such patterns of very thin snow haven't been observed in the Arctic. Also patterns of thick snow are very likely due to the retrieval algorithm, that is not adapted for thick ice. Regarding models, over the Antarctic, the MERCATOR model simulates mean values in better agreement with ASD. However, thicker snow patterns and the clockwise drifts in the Weddell sea are not captured by the model. The GIOMAS model simulates very high snow depths nearly everywhere, likely leading to unrealistic snow depth representations. Sea ice models are generally very dependent on atmospheric forcings and differences in snow accumulation in models can lead to very different snow depths. Model biases are largely due to the lack of consistent observations which has, so far, limited the tuning of snow parameters such as albedo or snow diffusion coefficients (Chevallier et al., 2017; Uotila et al., 2019). These comparisons highlight snow properties in the Antarctic very different than in the Arctic. Such opposite model behaviours in the two hemispheres argue that a single model tuning of parameters may not adapted for a global configuration. In the context of data assimilation, the positive impacts of the integration of SIT (Fiedler et al., 2021; Massonnet et al., 2015) could also be enhanced using ASD either by improving the SIT observations or by enabling the simultaneous use of freeboard and snow depth data to constrain the models.

An important limitation of the ASD snow depth data is its temporal coverage, imposed by SARAL which is only available since 2013. To circumvent this limitation, we have shown that a simple monthly climatology, constructed as the average of ASD snow depth maps over the 2013-2019 period, could be a relevant option. More specifically, we have demonstrated that such a climatology could provide a more reliable solution than the W99m climatology, which strongly overestimates snow depths and is useless outside the central Arctic due to extrapolation. Indeed, comparison between the ASD climatology and the 2009-2012 OIB missions shows good consistency (comparable results of similar magnitude as when comparing the actual ASD product with OIB campaigns from 2014-2018), while W99m is on average biased about 14 cm high with an associated RMSE of 17 cm (when comparing with OIB). Such a climatology should be refined in order to provide relevant snow depth representations for at least the Envisat/CS-2 time period. For instance, the possibility of constructing a climatology using both ASD, AMSR and W99m information should be investigated.

A better snow depth representation significantly improve SIT estimates. In this context, this study has investigated the impacts of different snow depth products on SIT. Meanwhile, we have also presented a methodology to characterise the SIT level of uncertainty due to the snow depth deviations between products. The approach constructed an ensemble of SIT solutions using the different snow depth products over the 2013-2019 period. Thus, providing an estimation of the SIT level of uncertainty due to snow depth discrepancies and its space and time variability. Note, that equivalent methodologies are frequently used in multi-model approaches. Taking only into account the satellite products AMSR2B, ASD, DuST and the W99m climatology over the 2013-2019 period, we found a spatially averaged mean standard deviation of 20 cm (14% of the pan-Arctic mean SIT) and mean maximum deviation of 49 cm (35% of the pan-Arctic mean SIT) between the different SIT estimations. Deviations between SIT estimations reached up to 77 cm (55% of pan-Arctic mean SIT) in marginal and coastal zones.

We have highlighted in this study that the DuST and ASD products, albeit computed from data acquired by the same sensors at the same time, present some significant differences due to the methodology applied. The main difference is, that for ASD we compare LRM data (PLRM for CS-2) while DuST compares LRM data with SAR data. The advantage of comparing LRM data together is that they have comparable footprint sizes, which by difference reduces the effects of surface roughness to highlight the effects of volume backscatter and penetration. DuST, on the other hand, relies on the calibration of the measurements on OIB without taking into account effects of the surface roughness. In fact, differences in surface roughness and the effect it has on radar penetration and retrieval of surface elevation is currently one of the other main contributors to the SIT uncertainty budget. (e.g., Hendricks et al., 2010; Landy et al., 2020), which still needs to be further investigated. We have only considered the roughness as mainly depending on the size of the footprint, but the backscattering coefficient variability is not as trivial and depends on several snow physical properties such as density, grain size, salinity and/or moisture (e.g., Lundberg et al., 2006; Adodo et al., 2018; Nandan et al., 2020). That is, specular surfaces have stronger nadir scattering of the radar pulse, affecting the shape of the radar echo and, in turn, the retrieval of the surface elevation. Empirical re-trackers like TMFRA have shown to be sensitive over rougher surfaces (e.g., Ricker et al., 2014) In particular, compared to physical retracking they have shown to over-estimate sea ice freeboards over rougher ice, while also underestimating sea ice freeboard over smooth, thin ice (e.g., Laforge et al., 2020; Landy et al., 2020). In turn, when assuming that snow depth can be retrieved from the difference between the Ku-band and Ka-band freeboards, the effect that surface roughness, and particularly the ku band volume retrodiffusion/penetration has on the retrieval of the surface elevation yields further investigation. An analysis of these dependencies is out the scope of this study but it is mandatory to further use the ASD data to improve our understanding of snow properties.

The use of two satellites, operating in different modes, with different orbits, also leads to important uncertainties and differences that cannot be neglected. The high-priority candidate mission, CRISTAL, which shall entail a dual-frequency Ka/Ku SAR altimeter could address some of these issues. The fact of having coincident and collocated SAR measurements obtained from the same platform, with similar ground footprints, for both frequencies, will enable direct comparisons and a better understanding of the relative impacts of the surface roughness, radar penetration and volume backscattering. In this context, ASD demonstrates the possibilities of dual-frequency snow depth estimations, which could be further improved using physical retracking (as opposed to empirical re-trackers which has their own limitations, e.g., Ricker et al., 2014). In addition, CRISTAL will also mitigate the issue of the spatial coverage limitation of 81.5°N imposed by SARAL in the Arctic. It is also important to note that the CRISTAL mission is the only altimetric mission planned to succeed CS-2 and avoid a gap of observations. Note that this COPERNICUS mission is not yet officially selected. If it is not confirmed, sea ice thickness measurement by radar altimetry above 81.5°N will end with the CS-2 mission. Prior to potential CRISTAL data, comparisons with the very recent ICESat-2/CS-2 snow depth estimations (Kwok et al., 2020) computed from the difference between laser and Ku-band measurements certainly provides important first results. Moreover, the recent change of CS-2 orbit to align with ICESat-2 (CRYO2ICE), providing collocated measurement with ≈3h temporal delay, also allows for investigation of Ku-band penetration and the discrepancies that different radar footprints can introduces.

*Data availability.* The ASD data used in this study are freely available at http://ctoh.legos.obs-mip.fr/data/sea-ice-products. In case of un-availability please contact the authors Florent Garnier (FG) or Sara Fleury (SF) to access the data.


*Author contributions.* FG and SF have conceived the study and wrote the paper. All named authors have participated to the present article within the named projects and have brought contributions to the elaboration of its final version.

*Competing interests.* The authors declare that they have no conflict of interest.

*Acknowledgements.* This research is supported by the ESA POLAR+ Snow on Sea Ice, the CryoSat+ Antarctic Ocean projects (CSAO+) and the CryoSat SciEnce-oriented data ANalysis over Sea-ICE areas (CryoSeaNICE). It has benefited from the support of the CNES TOSCA CASSIS project and the ESA living planet fellowship. The work has also been supported by the Programme Nationale de Télédétection Spatiale (PNTS, http://programmes.insu.cnrs.fr/pnts/), grant n° PNTS-2020-11. We also thank the Center for Topographic studies of the Oceans and Hydrosphere (CTOH) at LEGOS.

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

## Appendix A: Appendices

| Arctic | | ASD | MERCATOR | PIOMAS | NESOSIM | W99m | DuST |
|---|---|---|---|---|---|---|---|
| Interannual mean (cm) | 11 | 10.3 | 2.5 | 7.8 | 9.7 | 14.9 | 17.1 |
| | 12 | 10.6 | 4 | 10 | 11.9 | 15 | 17.5 |
| | 01 | 11.4 | 6.5 | 14 | 14.8 | 17.1 | 18.1 |
| | 02 | 12.4 | 8.8 | 17.5 | 18 | 18.8 | 19.1 |
| | 03 | 13.8 | 10.8 | 20.9 | 20.7 | 20.4 | 20.3 |
| | 04 | 15.3 | 11.6 | 23 | 22.38 | 19 | 20.9 |
| **Climatic mean** | | 12.3 | 7.4 | 15.5 | 16.2 | 17.5 | 18.8 |
| Interannual varibility (cm) | 11 | 4.8 | 1.4 | 3 | 2.4 | 2 | 4.3 |
| | 12 | 4.6 | 2 | 3.5 | 3 | 1.8 | 4 |
| | 01 | 4.5 | 2.6 | 4 | 3.1 | 2.3 | 4.3 |
| | 02 | 4.4 | 3 | 4.4 | 3.5 | 2.6 | 4.3 |
| | 03 | 4.7 | 3.9 | 5.5 | 4.3 | 2.6 | 4.5 |
| | 04 | 5.2 | 4.3 | 5.8 | 4.7 | 3 | 4.9 |
| **mean ≡ MIV** | | 4.7 | 2.9 | 4.4 | 3.5 | 2.4 | 4.4 |
| Annual variability (cm) | 2013-2014 | 4.2 | 3.3 | 5.5 | 5 | 4.3 | 4.3 |
| | 2014-2015 | 4 | 3.1 | 5.8 | 5 | 4.1 | 3.9 |
| | 2015-2016 | 4.1 | 2.7 | 4.7 | 4.1 | 3.8 | 4.2 |
| | 2016-2017 | 4.2 | 3.9 | 6.1 | 4.9 | 4.4 | 4.2 |
| | 2017-2018 | 4.2 | 4 | 6.9 | x | 3.6 | 3.8 |
| | 2018-2019 | 4.2 | 3.9 | 6.5 | – | 3.2 | – |
| **mean ≡ MAV** | | 4.18 | 3.5 | 5.9 | 4.8 | 3.9 | 4.1 |

**Table A1.** Statistical comparison of the different snow depth estimations in the Arctic over the six winters of the 2013-2019 period. AMSR2's data are not indicated since only the months of March and April were available.

1010

| Antarctic | | ASD | MERCATOR | GIOMAS | AMSR2 |
|---|---|---|---|---|---|
| (cm) | 05 | 14.9 | 10.2 | 26.1 | 17.3 |
| | 06 | 15.2 | 12.8 | 29 | 18 |
| | 07 | 16 | 16.9 | 33.5 | 18.7 |
| Interannual mean | 08 | 16.7 | 21.3 | 37.3 | 19.2 |
| | 09 | 17.8 | 25.2 | 40.5 | 17.4 |
| | 10 | 19.7 | 26.5 | 41.2 | 15.2 |
| **climatic mean** | | 16.7 | 18.8 | 34.6 | 17.6 |
| (cm) | 05 | 6 | 3.5 | 8.4 | 5.6 |
| | 06 | 5.9 | 4.5 | 7.8 | 5.5 |
| | 07 | 5.9 | 5.2 | 7.1 | 5.5 |
| Interannual varibility | 08 | 5.6 | 5.6 | 6.6 | 5.3 |
| | 09 | 6.2 | 6.1 | 6.5 | 4.9 |
| | 10 | 7.4 | 6.7 | 7 | 5.3 |
| **mean ≡ VIM** | | 6.2 | 5.3 | 7.2 | 5.3 |
| (cm) | 2013 | 5.5 | 6.7 | 7.5 | 4.1 |
| | 2014 | 5.2 | 6.6 | 6.6 | 4.3 |
| | 2015 | 5.1 | 6.8 | 6.4 | 3.7 |
| Annual variability | 2016 | 5.5 | 5.4 | 9.2 | 4.2 |
| | 2017 | 5.6 | 6.4 | – | 3.7 |
| | 2018 | 5.5 | 6.7 | – | 3.8 |
| | 2019 | 5.4 | 6.0 | – | 4.1 |
| **mean ≡ VAM** | | 5.4 | 6.4 | 7.4 | 4 |

**Table A2.** Statistical comparison of the different snow depth estimations in the Antarctic over the seven winters of the 2013-2019 period.

| all snow products | | m11 | m12 | m01 | m02 | m03 | m04 | mean |
|---|---|---|---|---|---|---|---|---|
| **Mean** | | 0.94 | 1.08 | 1.25 | 1.42 | 1.59 | 1.65 | 1.32 |
| **STD** | mean | 0.27 | 0.26 | 0.25 | 0.26 | 0.27 | 0.28 | 0.27 |
| | min | 0.20 | 0.19 | 0.18 | 0.18 | 0.18 | 0.18 | 0.18 |
| | max | 0.35 | 0.34 | 0.34 | 0.34 | 0.37 | 0.41 | 0.36 |
| **Max dev** | mean | 0.74 | 0.74 | 0.71 | 0.72 | 0.78 | 0.84 | 0.76 |
| | min | 0.55 | 0.54 | 0.50 | 0.49 | 0.51 | 0.53 | 0.52 |
| | max | 0.98 | 0.97 | 0.95 | 0.97 | 1.09 | 1.23 | 1.03 |
| obs snow products | | m11 | m12 | m01 | m02 | m03 | m04 | mean |
| **Mean** | | 1.09 | 1.21 | 1.34 | 1.48 | 1.62 | 1.65 | 1.40 |
| **STD** | mean | 0.21 | 0.21 | 0.21 | 0.20 | 0.19 | 0.19 | 0.20 |
| | min | 0.10 | 0.11 | 0.10 | 0.10 | 0.09 | 0.08 | 0.09 |
| | max | 0.32 | 0.32 | 0.31 | 0.30 | 0.31 | 0.30 | 0.31 |
| **Max dev** | mean | 0.51 | 0.52 | 0.50 | 0.48 | 0.47 | 0.47 | 0.49 |
| | min | 0.24 | 0.24 | 0.24 | 0.23 | 0.22 | 0.20 | 0.23 |
| | max | 0.78 | 0.80 | 0.78 | 0.75 | 0.76 | 0.77 | 0.77 |

**Table A3.** Statistical impact of the various snow depths on SIT in the Arctic over the 2013-2019 time period. The upper part consider all the snow depth products while the upper part consider only the products based on observations (ASD, DuST, AMSR2B and W99m). We remind that AMSR2B SIT estimations on MYI are only taken into account for the months of March and April.

| Long Name | Acronym | Reference | Data coverage |
|---|---|---|---|
| Alti Snow Depth | ASD | Present paper | 03/2013-04/2019 |
| Dual-altimeter Snow Thickness | DuST | Lawrence et al. (2018) | 03/2013-04/2018. |
| AMSR2 data of the university of Bremen | AMSR2B | Rostosky et al. (2018) | 03/2013-04/2018 |
| AMSR2 data from the National Snow and Ice Data Center | AMSR2 NSIDC | Meier et al. (2018) | 03-2013-continuous |
| The Pan-Arctic (Global) Ice-Ocean Modeling and Assimilation System (v2.1) | P(G)IOMAS | Schweiger et al. (2011) | 03/2013-04/2019 |
| LIM-2 model in the PSY4V3 MERCATOR configuration | MERCATOR | Lellouche et al. (2018) | 2013-2019 |
| The NASA Eulerian Snow On Sea Ice Model (v1.0) | NESOSIM | Petty et al. (2018) | 2013-2017 |

**Table A4.** Time period of snow depth products used in this article. Note that the current data availability may have changed since this article was written.

| Acronym | Long name |
|---|---|
| ALS | Airborne Laser Scanner |
| ASD | Alti Snow Depth |
| AMSR | Advanced Microwave Scanning radiometer |
| ARISE | Antarctic Remote Sensing Experiment |
| ASIRAS | ESA airborne Ku-band interferometric radar |
| ASPeCt | Antarctic Sea Ice Process and Climate |
| ASR | Arctic System Reanalysis |
| AVISO+ | Archiving, Validation and Interpretation of Satellite Oceanographic data |
| AWI | Alfred Wegener Institute |
| BROMEX | Bromine Ozone Mercury Experiment |
| CMEMS | Copernicus Marine Environment Monitoring Service |
| CNES | Centre national d'études Spatiales |
| CPOM | Centre for Polar Observation and Modelling |
| CRISTAL | Copernicus Polar Ice and Snow Topography ALtimeter |
| CRREL | Cold Regions Research and Engineering Laboratory |
| CryoSeaNICE | CryoSat SciEnce-oriented data ANalysis over Sea-ICE areas |
| CryoVEx | CRYOsat Validation Experiment |
| CS-2 | CryoSat-2 |
| CSAO+ | CryoSat+ Antarctic Ocean |
| DTU | Technical University of Denmark |
| DuST | Dual-altimeter Snow Thickness |
| EASE | Equal-Area Scalable Earth |
| ECMWF | European Centre for Medium-Range Weather Forecasts |
| ERA-Interim | Re-Analysis-Interim |
| ESA | European Space Agency |
| EUMETSAT | European Organisation for the Exploitation of Meteorological Satellites |
| FMCW | Frequency Modulated Continuous Wave |
| FYI | First Year Ice |
| GCOM-W | Global Change Observation Mission-Water |
| GOP | Geophysical Ocean Products |
| HPCM | High Priority Copernicus Candidate Mission |
| ICESat-2 | Ice,Cloud, and Land Elevation Satellite-2 |
| JRA | Japanese Meteorological Agency |
| JAXA | Japan Aerospace Exploration Agency |
| LEGOS | Laboratoire d'Etudes en Géophysique et Océanographie Spatiales |
| LIM | Louvain-la-Neuve sea Ice Model |
| LRM | Low Resolution Mode |
| MERRA | Modern-Era Retrospective analysis for Research and Applications |
| MYI | Multi Year Ice |
| N-ICE | Nansen basin during the Norwegian young sea ICE |
| NASA | National Aeronautics and Space Administration |

| MOSAIC | Multidisciplinary drifting Observatory for the Study of Arctic Climate |
|--------|------------------------------------------------------------------------|
| NCAR | National Center for Atmospheric Research |
| NCEP | National Center for Environmental Prediction |
| NEMO | Nucleus for European Modelling of the Ocean |
| NSIDC | National Snow and Ice Data Center |
| SMMR | Scanning Multichannel Microwave Radiometer |
| SMOS | Soil Moisture Ocean Satellite |
| SLA | Sea Level Anomalies |
| SSM/I | Special Sensor Microwave/Imager |
| SPICES | Space-borne Observation for Detecting and Forecasting Sea Ice Cover Extremes project |
| IFS | Integrated Forecast System |
| IMB | Ice Mass Balance buoys |
| ODATIS | Online Data Extraction Service |
| OIB | Operation Ice Bridge |
| OSISAF | Ocean and Sea Ice Satellite Application Facility |
| PLRM | Pseudo Low Resolution Mode |
| PP | Pulse Peakiness |
| SAR | Synthetic Aperture Radar |
| sd | snow depth |
| sgdr | Sensor Geophysical Data Records |
| SIT | Sea Ice Thickness |
| SHEBA | Surface HEat Budget of the Arctic Ocean |
| TFMRA | Threshold First Maximum Retracker Algorithm |
| UCL | University College of London |
| W99m | Warren-99 modified climatology |

**Table A5.** List of acronyms.