# Peer review of "Advances in altimetric snow depth estimates using bi-frequency SARAL/CryoSat-2 Ka/Ku measurements."

_The Cryosphere, 2021_

## Referee Comment (RC1)

Dear F. Garnier,

Thank you for submitting the manuscript tc-2021-79 with the title: "Advances in altimetric snow depth estimates using bi-frequency SARAL/CryoSat-2 Ka/Ku measurements." I find the paper very interesting, and with a lot of information – a very comprehensive piece of work. I in particular find it nice to see some results for the Antarctica. In general, I find the paper well organized and the figures and tables are clear. The methods and results are also well described. I do find the discussions part a bit short, please, see further questions raised in the general comments.

I have a few general comments and a longer list of technical comments, which I urge you to address:

**General Comments:**
**Validation data:**
You are absolutely right, there are not many have observations of snow depths in Antarctica.

Did you look at the AWI snow depth buoys yourself, or from where did you draw the conclusion that these data are not reliable ? Do you have a reference for this choice ? I do understand to exclude the AWI snow depth buoys in the Arctic, as most of the data was obtained north of 81.5N. Please, find the location mapped in Figure 1.

You state (Line 130) that there are no ASPeCT data for the period 2013-2019. This is not completely true. You can find the data including snow depths up to 2019 from Hamburg ([https://icdc.cen.uni-hamburg.de/en/seaiceparameter-shipobs.html](https://icdc.cen.uni-hamburg.de/en/seaiceparameter-shipobs.html))

Also for future validation data sets the AWI IceBird campaigns have since 2019 carried a snow radar ([https://www.awi.de/en/science/climate-sciences/sea-ice-physics/projects/ice-bird.html](https://www.awi.de/en/science/climate-sciences/sea-ice-physics/projects/ice-bird.html)).

In line 129-130: You also state that neither the OIB nor CryoVEx data are available yet in Antarctica. This is not entirely true, the level-1 processed data is available ([https://earth.esa.int/eogateway/campaigns/cryovex-karen-antarctica-2017-2018](https://earth.esa.int/eogateway/campaigns/cryovex-karen-antarctica-2017-2018)), but higher level freeboard and snow depth data is not yet publicly available. For OIB there are freeboard but no snow for 2009-2010 missions.

[Figure]

Figure 1: AWI snow depth buoys Antarctica (left) and Arctic (right)

**CryoSat-2 Baselines:**

I am a bit confused about the different baselines, which you use, and why you do not use the most recent one.

In line 150 you refer to the GOP CryoSat-2 Baseline C Product User Handbook, but later in the same section (line 154) you write that you are using Baseline-B. Is Baseline B latest baseline for the GOP data set ? And If Baseline B is not the latest, which changes would you expect in the final ASD product if you used a later baseline ?

Further down the document Line 425) you use Baseline C for SIT estimation ? The most recent Baseline is Baseline D. Why did you not use this, and what would you expect for changes if you used Baseline D ?

**Periods:**
I find it a bit confusing with the different time-periods, sometimes you include 2013-2018 and at other times 2013-2019. e.g. data based on input to Fig. 3 (2013-2018) and Fig. 4 (2013-2019)?. Then also the values in Table A1 (2013-2019) for ASD is not similar for those 2 as stated in e.g. line 307.

I cannot find anything consistent when and why you are using different time-intervals.

I urge you to go for the 2013-2019 period.

Also why did you discard the OIB data from 2019, and 2013 for the climatology?

**Discussion:**
I find the discussion to be rather short. Some points which I find interesting to discuss further is:

You come up with suggestions of how to produce a snow depth climatology prior the SARAL/CS-2 overlap period, which I find great, however, I miss suggestions for post SARAL/CS-2 period. You mention CRISTAL, but there will most likely be a gap. How should this be covered ?

Do you expect all months to be equally representative considering the different temperatures, and effect of melt versus reduced penetration depths of the Ku-band radar signal ?

Do you expect any degradation of the ASD after SARAL went into drifting orbit ?

Even though you discuss a bit about the limitations of the individual validation data sets, you still draw the conclusion that ASD is the "best" option, when you compare with the validation data. What would happen if you, e.g. used another processing of OIB, would DuST then be the "best" option ? How can we secure the most optimal observations in the future to validate satellite derived snow depth products ?

**Technical comments:**

For consistency, please, consider which of the following options you would like to include in the manuscript with respect to:

- Re-analysis or Reanalysis
- Ka band (Ku band)  -> Ka-band (Ku-band)

**1 Introduction:**

Line 10: It is **further** validated in the Arctic …

Line 42: in **the** Arctic

Line 49: in **the** Arctic

Line 54: CRYOsat Validation Experiment -> CryoSat Validation Experiment

Line 54-56: I am not quite sure why you have picked these references, i.e. Haas et al. 2006 and Helm et al. 2006. In case you want to add these you should in principle include all reports from all CryoVEx campaigns since 2003. Overviews along the way are provided in Hvidegaard et al. 2006 and Skourup et al. (2012).  For later campaigns, please, refer to relevant CryoVEx technical reports to be found at https://earth.esa.int/eogateway/search?category=Campaigns&filter=cryosat

Skourup, H., S. M. Hvidegaard, R. Forsberg, I. Einarsson, A. V. Olesen, L. S. Sørensen, L. Stenseng, S. Hendricks, V. Helm and M. Davidson: CryoVEx 2011-12 airborne campaigns for CryoSat validation. Proceedings paper (ESA-SP-710), 20 years of progress in radar altimetry, 24-29 September, Venice, Italy, 2012

Hvidegaard, S. M., R. Forsberg, and H. Skourup: Sea ice thickness from Airborne Laser Scanner. Arctic Sea Ice Thickness: Past, Present and Future, edited by P. Wadhams and G. Amanatidis. Climate Change and Natural Hazards Series, Brussels, 2006

Line 56: Their measurements include bi-frequency altimetry snow depth estimations as of 2017 (see Sect. 3.3). -> Their measurements include bi-frequency altimetry snow depth estimations as of 2017 (Skourup et al., 2019) (see Sect. 3.3).

Skourup, H., A. V. Olesen, L. Sandberg Sørensen, S. Simonsen, S. M. Hvidegaard, N. Hansen, A. F. Olesen, A. Coccia, K. Macedo, V. Helm, R. S. Ladkin, R. Forsberg, A. E. Hogg, I. Otosaka, A. Shepherd, C. Haas and J. Wilkinson. ESA CryoVEx/KAREN and EU ICE-ARC 2017 - Arctic field campaign with combined airborne Ku/Ka-band radar and laser altimeters, together with extensive in situ measurements over sea- and land ice. Technical Report, National Space Institute, Danish Technical University (DTU Space), ISBN-978-87-91694-45-5, August 2019

Line 56: cover.Their -> cover. Their (that is insert space after dot)

Line 61: snow depth measurements **in the Arctic** (Kurtz …)

Line 70: several CryoVEx and OIB missions have been conducted in Antarctica -> several OIB missions and the CryoVEx 2017/18 campaign (Hvidegaard et al., 2020) have been conducted in Antarctica

Hvidegaard, S. M., R. Forsberg, H. Skourup, M. L. Kristensen, A. V. Olesen, A. F. Olesen, A. Coccia, K. Macedo, V. Helm, R. S. Ladkin, R. Tilling, A. E. Hogg, Adriano Lemos and A. Shepherd. ESA CryoVEx/KAREN Antarctica 2017-18 - Antarctic field campaign with combined airborne Ku/Ka-band radar and laser altimeters, together with extensive in situ measurements over sea- and land ice. Technical Report, National Space Institute, Danish Technical University (DTU Space), ISBN 978-87-91694-50-9, October 2020

Line 104: its -> It is

Line 104: its -> It is

Line 113: Mode(LRM) -> Mode (LRM)

Line 48: I suggest you add the more local/regional climatologies such as Forsström et al. (2011), which provides snow depths for e.g. the Fram Strait, which represent an area outside the central Arctic where W99 climatology is not working properly due to extrapolation.

Forsström, S., Gerland, S., & Pedersen, C. (2011). Thickness and density of snow-covered sea ice and hydrostatic equilibrium assumption from in situ measurements in Fram Strait, the Barents Sea and the Svalbard coast. Annals of Glaciology, 52(57), 261-270. doi:10.3189/172756411795931598

Line 130: AspeCt data including snow depths are available up to 2019 from Hamburg (https://icdc.cen.uni-hamburg.de/en/seaiceparameter-shipobs.html)

**2 Data processing of ASD**

Line 145: which is assumed to be reflected near the top of the snow pack, **i.e.** the

air/snow interface

Line 150 Why do you refer to CryoSat-Baseline-C Product user handbook when you are using Baseline-B as stated in Line 153-154 ? See also general comments.

Line 159-162: You explicitly mention DTU15 MSS, but not the other models why is this the case ?

Line 167-68: Does these limitations on PP apply to both Ku- and Ka-band ?

Line 170: Threshold First Maximum Retracker Algorithm (TMFRA; -> Threshold First Maximum Retracker Algorithm (T**FM**RA;

Line 177: we need to take into account the decreasing of the Ku radar echo velocity when it penetrates into the snow pack -> we need to take into account the **decrease** of the Ku radar echo velocity **as** it penetrates into the snow pack

Line 186: radarKu -> $Ku_r$ (as you also made this for the snow $sd_r$)

Line 192-193: In the Arctic, it also shows that these data are different from the Warren 99 modified (W99m) climatology, where nearly everywhere the W99m climatology exhibits thicker snow layers over sea ice. -> In the Arctic, it also shows that these data are different from the W99m climatology, where the W99m climatology tend to exhibit thicker snow layers over sea ice.

**3 External data sets**

Line 197-199: For both hemispheres, the time period of model and satellite products that have been used is explained in Fig. 2 (ranges from 2014–2019 for the Arctic, and 2013–2019 for the Antarctic with limitations for some data products for both hemispheres). -> For both hemispheres, the time period of model and satellite products ranges from 2014–2019 for the Arctic, and 2013–2019 for the Antarctic (with limitations for some data products for both hemispheres), as explained in Fig. 2.

Line 208: located near the Canadian Archipelago ( and the Beaufort Sea, and only -> located near the Canadian Archipelago and the Beaufort Sea, and only

Line 219: cover over FYI Because of this limitation, -> cover over FYI. Because of this limitation,

Line 222: Other months are only available on FYI -> Other months are only available over FYI

Line 232: description can be found in Schweiger et al. (2011) -> description can be found in Schweiger et al. (2011).

Line 238: Louvain-la-Neuve sea Ice Model (LIM2) (Fichefet and Maqueda, 1997; Vancoppenolle et al., 2012) -> Louvain-la-Neuve sea Ice Model (LIM2) (Fichefet and Maqueda, 1997; Vancoppenolle et al., 2012).

Line 230: OSISAF -> OSI SAF

Line 252: CRYOsat Validation Experiment -> CryoSat Validation Experiment

Line 256: (Haas et al., 2017), -> (Skourup et al. (2019), Haas et al. (2017)), Please, also use DOI for the CryoVEx 2017 campaign data: https://doi.org/10.5270/esa-enocas0 - CryoVex-KAREN 2017 Campaign: "ESA CryoVEx/KAREN and EU ICE-ARC 2017 - Arctic field campaign with combined airborne Ku/Ka-band radar and laser altimeters, together with extensive in situ measurements over sea- and land ice" obtained from the ESA campaign site: https://earth.esa.int/eogateway/campaigns/cryovex-karen-2017?text=CryoVEx+2017

Line 260: OIB is one of the largest airborne mission in polar -> OIB is one of the largest airborne mission**s** in polar

Line 264: (Kurtz et al., 2012 (Updated 2015; King et al., 2015) I am not sure I understand this ?

Line 267: Why did you not include OIB from 2019 ?!?

Line 267-268: comparison with the ASD climatology, we compare with all OIB spring campaigns during 2009–2012. Did you use quiklooks here or the IDCS4 ? It is not clear from section 3.3.2

Line 272: et al., 2006; Perovich and Richter-Menge, 2015) Snow depth -> et al., 2006; Perovich and Richter-Menge, 2015). Snow depth

**4 Comparison between snow depth data**

Line 277: 500×500 EASE2 grid … is the grid in km ? i.e. 500x500km ?

Line 281: of all snow depth monthly maps -> of all monthly snow depth maps

Line 294-296: Note, that when comparing the various snow depth data with the OIB snow radar data, at achieve a corresponding spatial scale, we have applied a 25 km window rolling mean for smoothing the OIB data. -> Note, that when comparing the various snow depth data with the OIB snow radar data, we have applied a 25 km window rolling mean for smoothing the OIB data, to achieve similar spatial scales.

Line 305: zones -> regions

Line 305: and higher snow depths over region of thicker sea ice -> and higher snow depths over regions of thicker sea ice

4.2.1 I do not like the phrasing "Annual means" for means calculated only based on March and April data, maybe rephrase this to seasonal or winter means

Line 319: comparable both in terms -> similar both in terms

Line 330: Although investigating which causes these discrepancies -> Although investigating **what** causes these discrepancies

Line 334: ASD is the only publicly available altimetric snow depth product in Antarctica. -> ASD is **currently** the only publicly available altimetric snow depth product in Antarctica.

Line 337: I would remove: "(equivalent to PIOMAS but for Antarctica)", as you have already mentioned this.

Line 342: east side, -> eastern part

Line 343: One relevant difference between these two data is -> One relevant difference between these two data **sets** is

Line 353: between -> from

Figure text 7: considered -> included

Figure text 6: during -> acquired on

Figure text 6: Remove: "2015 spring campaign)

Line 353: Fig. 6 shows an example the 29th March 2015 track -> Fig. 6 shows an along-track example from the OIB flight on 29th March 2015

Line 354: comparing with the OIB data over the entire 2014-2018 time period -> comparing **the various snow depth products** with the OIB data over the entire 2014-2018 time period

Line 354: Why do you not include OIB from 2019 ?

Line 357: It is almost impossible to see the shading, you might want to upgrade the color in order to be visible

Line 357: data are nearly always included within the ASD envelope of uncertainties (in shaded red). -> data are almost within the ASD envelope of uncertainties (in shaded red) at all times.

Line 369: Remove: "always"

Line 361: overestimate OIB with a level comparable with DuST -> overestimate at a level comparable with DuST, when compared to OIB.

Line 365: MYI Kwok et al. (2017). -> MYI (Kwok et al., 2017).

Figure 6: Out of curiosity., what happens after trackpoint 20000 ? Here there are a huge difference between OIB/ASD/DuST, could be interesting to look further into this.

Line 369: a less optimal solution -> a less optimal solution in this case.

Line 375: Fig. 8 and Table 2 show how the satellite mean values are closer to the in-situ observations than the model -> Fig. 8 and Table 2 show how the satellite **derived** mean values are closer to the in-situ observations than the model**s**

Line 376: seasonal changes in the IMBs -> seasonal changes captured in the IMBs

Line 376: or -> nor

Line 377: localized -> point

Figure 8: Why did you not include the ASD uncertainties as shading in this plot similar to Figure 6 and 9 ?

Line 387: Ka-ASR estimation exhibits very thin snow thickness (< 10cm).

Line 389: Ka band radar -> Ku-band radar

Line 392: (in pink shading). -> similar as for Figure 7 it is difficult to see, and also in figure 7 you called I red shading, please, be consistent.

Line 394: It does not really make sense to use W99/W99m in these areas, but ok since I presume this is what is used in the SIT productss

Line 390: Why is there a gap in the ALS-ASR data ?

Line 390: Except around 71.3N, where a specific event may have occurred, the ASD snow product tracks the magnitude of the CryoVEx airborne data

Line 390-392: I am not really convinced about this. But you are right, that

Line 391: KAREN-ASR

Line 395: The -> the

**5 Towards an ASD snow depth climatology**

Line 397: the temporal coverage -> the limited temporal coverage

Line 397: (only after 2013) -> (only available post-2013)

Line 399: Remove: "to this limitation"

Figure 10: the ASD and the Warren W99m climatologies considering all the tracks of 2009,2010,2011 and -> the ASD and the W99m climatologies considering all the tracks of 2009, 2010, 2011 and

Line 407: as obtained in Fig. 7 with the ASD data. -> as obtained by direct correlation between ADS and OIB measurements (Fig. 7).

Line 409: product for the years of Envisat -> product representing the Envisat-era.

Line 409. Remove: "Warren"

**6 Impact of snow depth on SIT estimation**

Line 425: Baseline C ?!? See general comment.

Figure 11: The red and blue colors for the different representations are difficult to separate, i.e. ASD, DuST, SIT min-max. I encourage you to find another solution for this.

Line 431: taking into account the radar speed velocity decreasing which depends on the snow depth data -> taking into account the reduction of radar velocity within the snow layer, which depends on the snow depth data

Line 436: lower -> low

Line 439: comprised -> distributed

Line 446: $4.10^3 km^3$ -> $4 \times 10^3 km^3$

Line 449: 3.5.1015 -> 3.5 x 1015

Line 450: $2.10^8$ -> $2 \times 10^8$

Line 459: maxdevy,m(SIT) -> (maxdevy,m(SIT))

Line 459: Eqs.(9) -> Eqs. (9)

Line 459 and onwards, be consistent to refer to equations in a similar way, e.g. Eqs. (9) or Eqs 9

Line 466: Eqs. 11 to 13. Ny is the number of years. -> Eqs. (11) to (13), where Ny is the number of years.

Line 471: I do believe Eq. (13) should be on the rightside of the summation in Eq. (12) ?

Line 480: due only to snow depth -> caused by the snow depth

Line 481: We distinguish two cases: the case « obs snow products », which considers ASD, DuST, W99m and AMSR2B and the case « all snow products », which includes in addition the PIOMAS, MERCATOR and NESOSIM model solutions. -> We define two cases: 1) « obs snow products », which considers ASD, DuST, W99m and AMSR2B and 2) « all snow products », which includes in addition the PIOMAS, MERCATOR and NESOSIM model solutions.

Line 484: Remove: "For similar reasons, AMSR2-NSIDC is also not taken into account", as this has not been in the Arctic analysis

Line 485: Eqs.(11) to (17) -> Eqs. (11) to (17)

Line 485-486:

Kine 488; (spatial mean of $SIT_{y,m,p}$ maps are also included -> ($SIT_{y,m,p}$ ) are also included

Line 489: at the ocean/ice transitions -> marginal ice zones

Line 495: snow depth from -> snow depth **variability** from

Line 495: an other -> another

Line 499: 76*cm* -> 76cm

.Line 508: for constraining models -> to constrain models

Line 509: initiative -> initiatives

Line 512: Polar Week -> Polar Science Week

**7 Conclusions and Discussions**

Line 519: and a CryoVEx airborne campaign -> and CryoVEx airborne campaign data

Line 519: Ka-Ku snow depth -> Ka-Ku and ALS-Ku snow depths

Line 523: always exhibits higher snow depth patterns -> is always biased high.

Lone 530: always far lower the -> always biased low when compared to the other products.

Line 531: in -> on

Line 540: Weddell sea are not represented -> Weddell sea are not captured by the model

Line 554: would need to be done -> is needed

Line 557: versions, etc...), -> versions),

Line 560: is its temporal coverage, imposed by SARAL (only from 2013). -> is its **limited** temporal coverage, imposed by SARAL which is only available since 2013.

Line 563: Please add: which strongly overestimates snow depths**, and is useless outside the central Arctic due to extrapolation**.

Line 564: 2009-2013 … wasn't it 2009-2012 ?

Line 565-566: has a mean bias of about 14 cm with -> on average is biased ~14 cm high with

Line 568: could -> should

Line 570: pacts of the snow depth on SIT -> pacts of different snow depth products on SIT

Line 575: 14.2% -> 14%

Line 576-577:  Please, add some of the locations here.

Line 584: uncertainty -> uncertainty budget.

Line 585: which still to be further investigated -> which still **needs**  to be further investigated

Line 587: or -> and/or

*Data availability:*
It is nice with a link to the data set presented in this paper. In some journals you need to link or provide information about all the data sets used, but I am not sure whether this also is the standard for TC, or whether the links provided in the text is enough.

---

## Author Comment (AC1)

**Title: Advances in altimetric snow depth estimates using bi-frequency SARAL/CryoSat-2 Ka/Ku measurements.**

*Florent Garnier, Sara Fleury, Gilles Garric, Jérôme Bouffard, Michel tsamados, Antoine Laforge, Marion Bocquet, Rénée Mie Fredensborg Hansen and Frédérique Rémy*

**Referee#1 global comment**

Thank you for submitting the manuscript tc-2021-79 with the title: "Advances in altimetric snow depth estimates using bi-frequency SARAL/CryoSat-2 Ka/Ku measurements." I find the paper very interesting, and with a lot of information – a very comprehensive piece of work. I in particular find it nice to see some results for the Antarctica. In general, I find the paper well organized and the figures and tables are clear. The methods and results are also well described. I do find the discussions part a bit short, please, see further questions raised in the general comments. I do find lots of technical issues, which has been raised together with the general comments in the supplemented pdf-file.

**Answers to the anonymous referee#1 comments**

We would like to thank the reviewer for her/his careful reading of the manuscript and for the relevant and constructive remarks. In order to fit with your comments and those of the other referees, we have made a complete revision of the manuscript that should have considerably improved the quality of the writing and the readability of the document. Note that we have also broaden the discussion part. We hope that the modifications will meet your requirements. Please find below the details on how your general and specific comments have been taken into account. In this « author's response document », the referee's comments are in bold type, the answers are in italic type, and the corrections to the revised manuscript are in normal type.
* * *
**1) Answers to the anonymous referee#1 : general comments**

**1.1) Validation data**

**You are absolutely right, there are not many have observations of snow depths in Antarctica.**

**Did you look at the AWI snow depth buoys yourself, or from where did you draw the conclusion that these data are not reliable ? Do you have a reference for this choice ? I do understand to exclude the AWI snow depth buoys in the Arctic, as most of the data was obtained north of 81.5N. Please, find the location mapped in Figure 1.**

*Indeed, we have looked at the AWI snow buoys. The main issue is that they provide snow accumulation, which is not exactly the snow depth. In the Arctic, both might be quite comparable but, as you mentioned, the Saral orbit inclination drastically limits the comparison. In addition, we have several other datasets. In Antarctica, we might haven't look deeply enough but, probably mainly because of the flooding, our comparisons did not seem to be consistent. We haven't been able to identify relevant patterns that could be directly linked with the snow depth. A deeper analysis should be necessary but, in any cases, this comparison*

*wouldn't be sufficient for a proper validation in Antarctica. Then, we have decided not to present these comparisons in this article. In order to clarify this point we have modified the sentence in the introduction as follows:*

35  L81-83: To our knowledge, only the Meereis data portal (https://www.meereisportal.de/) provides snow buoys data of snow accumulation (Grosfeld et al., 2016) (for both hemispheres), but the comparison with snow depth is not direct. It is, for instance, limited by the flooding of ice floes due to the heavy snow loading occurring in the Antarctic.
* * *
**You state (Line 130) that there are no ASPeCT data for the period 2013-2019. This is not completely true. You can**
40  **find the data including snow depths up to 2019 from Hamburg (https://icdc.cen.uni- hamburg.de/en/seaiceparameter-shipobs.html)**

**Also for future validation data sets the AWI IceBird campaigns have since 2019 carried a snow radar (https://www.awi. de/en/science/climate-sciences/sea-ice-physics/projects/ice-bird.html).**

45  *We do apologize for the inaccuracy of our statement concerning ASPeCt. Of course, for future snow depth validation and comparison in the Arctic, we look forward to the IceBird snow radar datasets. We have modified as the introduction of the revised manuscript as follows:*

L84-88: Currently, the main expedition providing snow depth data in the Southern hemisphere, is the ASPeCt program (Worby et al., 2008), established in 1996 to model the role of the Antarctic sea ice in the coupled atmosphere-ice-ocean system. ASPeCt
50  recent standardized data (Kern, 2020) cover the period 2002-2019 but with a too sparse amount of information for a reliable assessment. Future validations in the Antarctic will also benefit from the recent AWI IceBird campaigns carrying a snow radar since 2019 (https://www.awi.de/en/science/climate-sciences/sea-ice-physics/projects/ice-bird.html).
* * *
**In line 129-130: You also state that neither the OIB nor CryoVEx data are available yet in Antarctica. This is not entirely**
55  **true, the level-1 processed data is available (https://earth.esa.int/eogateway/campaigns/cryovex-karen-antarctica-2017-2018), but higher level freeboard and snow depth data is not yet publicly available. For OIB there are freeboard but no snow for 2009-2010 missions.**

*You are completely right that our phrasing is subject to misunderstanding. We also had this comment when presenting these results for the 10th years of CryoSat-2 conference. L1b of OIB and CryoVEx are available but they need a treatment that would*
60  *take a lot of time and which is not entirely in our field of expertise. we have modified as follows*

L80: Actually, several CryoVEx and OIB missions have been conducted in the Antarctic, but the data are still only available in raw L1b level and difficult to interpret..
* * *
**1.2) CryoSat-2 Baselines**

65  **I am a bit confused about the different baselines, which you use, and why you do not use the most recent one. In line 150 you refer to the GOP CryoSat-2 Baseline C Product User Handbook, but later in the same section (line 154) you write that you are using Baseline-B. Is Baseline B latest Baseline for the GOP data set ? And If Baseline B is not the latest, which changes would you expect in the final ASD product if you used a later baseline ?**

*We use the GOP CryoSat-2 Baseline B until 2017 and Baseline C from 2017 to 2019. We did not used the Baseline C for all the*
70  *time period simply because the Baseline C was not available at the time we processed the data. It is now available. Because*

*the waveforms are exactly the same in the two Baselines (and then the computed snow depths), we have decided to keep this version for this article. Note that we are currently re-processing the data so that the ASD product will be computed over the 2013-2020 period in both hemisphere from the baseline C only . The only difference is then that it will also cover the SARIN mode zones. You are right that it is important to let the reader (and probable future users) be aware of this. For that purpose, we have modified the revised manuscript as follows :*

L166-171 : Since the Baseline-C PLRM GOP product was not fully available at the time we have computed the ASD data, we have used the Baseline B (until 2017 and the Baseline C from 2017 to 2019) that does not include the SARin data. Then, the ASD data does not yet cover the SARin mode zones. The next version of the ASD product will include SARin mode zones. Apart from this spatial coverage, there will be no difference with the ASD version presented in this article since waveforms in L1b products are the identical in Baseline B and C.
* * *
**Further down the document (Line 425) you use Baseline C for SIT estimation ? The most recent Baseline is Baseline D. Why did you not use this, and what would you expect for changes if you used Baseline D ?**

*It is true that the Baseline D is the most recent Baseline (and soon Baseline E...). We did not use the Baseline D because we haven't reprocessed all CryoSat-2 freeboard with Baseline D. The waveforms are identical in the Baselines C and D so that does not change the results on freeboards. Differences between the 2 baselines are especially when you use L2 products. Since we never mention the baseline D in the document, we would prefer not to add this point in order to avoid confusion between the different products.*
* * *
**1.3) Periods**

**I find it a bit confusing with the different time-periods, sometimes you include 2013-2018 and at other times 2013-2019. e.g. data based on input to Fig. 3 (2013-2018) and Fig. 4 (2013-2019)?. Then also the values in Table A1 (2013-2019) for ASD is not similar for those 2 as stated in e.g. line 307.**

**I cannot find anything consistent when and why you are using different time-intervals.**

**I urge you to go for the 2013-2019 period.**

**Also why did you discard the OIB data from 2019, and 2013 for the climatology?**

*We agree that the various time period used for the diagnosis might be confusing. We have used (slightly) different time periods because of the availability of the different products. Our objective is to perform comparisons as reliable as possible, i.e with the exact same amount of data (same spatial and temporal coverage). For instance in Figure 3, we present the AMSR2B data that are only available until 2018. Then we discard 2019. This is also why we did not use the 2019 OIB campaign. We could obviously perform comparisons using,for each product, the maximum of data but we think that statistics are more relevant this way. In order to make the data availability clear for the reader, we added a summary table (Table A4 in appendices) Also the following sentences have been added in the revised document.*

L299-300: Note that to improve the consistency of the analyses we always use the larger common time period for all the compared products. Thus, the covered time periods may vary depending on the availability of the snow depth products.
L219: Table A4 specifies the time period of the different satellite and model data.
* * *
**1.4) Discussions**

**I find the discussion to be rather short. Some points which I find interesting to discuss further is: You come up with**
110 **suggestions of how to produce a snow depth climatology prior the SARAL/CS-2 overlap period, which I find great, however, I miss suggestions for post SARAL/CS-2 period. You mention CRISTAL, but there will most likely be a gap. How should this be covered ?**

*In spite of the uncertainty due to the variability of snow properties and their effects on the signal, we believe that all the 6 winter months should be equally representative. It would not be the case for other months, for instance in spring the melting*
115 *strongly impact freeboard calculations (at least distinction between leads and floes). Actually, this is one of the reason why we still only compute snow depth over the 6 winter months. Note that we have broaden the discussion part to better mention CRISTAL and highlight the importance of snow properties and the impacts of roughness and penetration. for instance :*

L596-604: Since the results of such comparisons with validation data can vary from one methodology to another (e.g., grid sizes, smoothing kernel, dataset versions), they do not aim to assess the best snow depth product. However, they demonstrate
120 how ASD provides a relevant snow depth solution, in good agreement with several validation data, and that ASD allows for characterisation the deviations between the different snow depth products. A more refined comparison with the CryoVEx data 600 (including various tracks) is mandatory to understand the relative impacts of roughness and penetration and their link with snow properties. For instance, Willatt et al. (2011) show that the Ku-band dominant scattering surface can significantly vary with snow temperature, with a reduced penetration when temperatures increase. This feature could enhance an underestimation
125 of ASD data with the warming of temperatures due to climate change. It also points out the difficulty to retrieve snow depth from altimetry beyond the 6 months of winter as variations in snow properties are most important in the summer period.
* * *
**Do you expect all months to be equally representative considering the different temperatures, and effect of melt versus reduced penetration depths of the Ku-band radar signal ?**

130 *In spite of the uncertainty due to the variability of snow properties and their effects on the signal, we believe that all the 6 winter months should be equally representative. It would not be the case for other months, for instance in spring the melting strongly impact freeboard calculations (at least distinction between leads and floes). Actually, this is one of the reason why we still only compute snow depth over the 6 winter months. Note that we have broaden the discussion part to better highlight the importance of snow properties and the impacts of roughness and penetration. for instance :*

135 L571-576: A more refined comparison with the CryoVEx data (including various tracks) is mandatory to understand the relative impacts of roughness and penetration and their link with snow properties. For instance, Willatt et al. (2011) show that the Ku-band dominant scattering surface can significantly vary with snow temperature, with a reduced penetration when temperatures increase. This feature could enhance an underestimation of ASD data with the warming of temperatures due to climate change. It also points out the difficulty to retrieve snow depth from altimetry beyond the 6 months of winter as variations in snow
140 properties are most important in the summer period.
* * *
**Do you expect any degradation of the ASD after SARAL went into drifting orbit ?**

*We do not expect any degradation of the ASD for orbits modifications because the snow depths are calculated from the difference of the monthly means.*

145
* * *
**Even though you discuss a bit about the limitations of the individual validation data sets, you still draw the conclusion that ASD is the "best" option, when you compare with the validation data. What would happen if you, e.g. used another**

processing of OIB, would DuST then be the "best" option ? How can we secure the most optimal observations in the future to validate satellite derived snow depth products ?

150 *Using another OIB product, in particular the one which is use for calibration, it is quite possible that the comparisons with OIB underline better statistics for DuST than for ASD. However, it won't be drastically different from the results we show in the article. It would rather be a question of magnitude, not annual or interannual variabilities. In addition, different methodologies for the comparisons could give different results. We think that it is not relevant to assess the best product since each has relevant patterns. Our objective is to demonstrate that ASD is a relevant solution to estimate snow depth. The main reason towards the*
155 *use of ASD compare to other products is that it is available in the two hemispheres which is crucial for data assimilation in global operational systems. We have better specified this point in the last part of the introduction of the revised manuscript.*
* * *
**2) Technical comments**

**For consistency, please, consider which of the following options you would like to include in the manuscript with respect**
160 **to: • Re-analysis or Reanalysis • Ka band (Ku band) -> Ka-band (Ku-band)**

*We have carefully modified the document in order to use the spellings Ka-band(Ku-band) and Reanalysis.*

**2.1) Introduction**

**Line 10: It is further validated in the Arctic ...**
**Line 42: in the Arctic**
165 **Line 49: in the Arctic**
**Line 54: CRYOsat Validation Experiment -> CryoSat Validation Experiment**

*The above corrections have been modified in the document*
* * *
**Line 54-56: I am not quite sure why you have picked these references, i.e. Haas et al. 2006 and Helm et al. 2006. In case**
170 **you want to add these you should in principle include all reports from all CryoVEx campaigns since 2003. Overviews along the way are provided in Hvidegaard et al. 2006 and Skourup et al. (2012). For later campaigns, please, refer to relevant CryoVEx technical reports to be found at https://earth.esa.int/eogateway/search?category=Campaigns&filter=cryosat Skourup, H., S. M. Hvidegaard, R. Forsberg, I. Einarsson, A. V. Olesen, L. S. Sørensen, L. Stenseng, S. Hendricks, V. Helm and M. Davidson: CryoVEx 2011-12 airborne campaigns for CryoSat validation. Proceedings paper (ESA-SP-**
175 **710), 20 years of progress in radar altimetry, 24-29 September, Venice, Italy, 2012**
**Hvidegaard, S. M., R. Forsberg, and H. Skourup: Sea ice thickness from Airborne Laser Scanner. Arctic Sea Ice Thickness: Past, Present and Future, edited by P. Wadhams and G. Amanatidis. Climate Change and Natural Hazards Series, Brussels, 2006**

**Line 56: Their measurements include bi-frequency altimetry snow depth estimations as of 2017 (see Sect. 3.3). -> Their**
180 **measurements include bi-frequency altimetry snow depth estimations as of 2017 (Skourup et al., 2019) (see Sect. 3.3). Skourup, H., A. V. Olesen, L. Sandberg Sørensen, S. Simonsen, S. M. Hvidegaard, N. Hansen, A. F. Olesen, A. Coccia, K. Macedo, V. Helm, R. S. Ladkin, R. Forsberg, A. E. Hogg, I. Otosaka, A. Shepherd, C. Haas and J. Wilkinson. ESA CryoVEx/KAREN and EU ICE-ARC 2017 - Arctic field campaign with combined airborne Ku/Ka-band radar and**

laser altimeters, together with extensive in situ measurements over sea- and land ice. Technical Report, National Space Institute, Danish Technical University (DTU Space), ISBN-978-87-91694-45-5, August 2019

*The citations have been modified following your recommendations.*

L61-64: Since 2003, the CryoSat Validation Experiment (CryoVEx) campaigns (e.g Haas et al. (2006); Helm et al. (2006); Skourup et al. (2013); Hvidegaard et al. (2006)) have provided data with the main goal of investigating radar penetrations into ice and snow cover. Their measurements include bi-frequency altimetry snow depth estimations as of 2017 (H. Skourup and Wilkinson, 2019)
* * *
**Line 56: cover.Their -> cover. Their (that is insert space after dot)**

**Line 61: snow depth measurements in the Arctic (Kurtz ...)**

**Line 70: several CryoVEx and OIB missions have been conducted in Antarctica -> several OIB missions and the Cry-oVEx 2017/18 campaign (Hvidegaard et al., 2020) have been conducted in Antarctica**

**Hvidegaard, S. M., R. Forsberg, H. Skourup, M. L. Kristensen, A. V. Olesen, A. F. Olesen, A. Coccia, K. Macedo, V. Helm, R. S. Ladkin, R. Tilling, A. E. Hogg, Adriano Lemos and A. Shepherd. ESA CryoVEx/KAREN Antarctica 2017-18 - Antarctic field campaign with combined airborne Ku/Ka-band radar and laser altimeters, together with extensive in situ measurements over sea- and land ice. Technical Report, National Space Institute, Danish Technical University (DTU Space), ISBN 978-87- 91694-50-9, October 2020**

*The modifications and the reference proposed have been added to the revised document*
* * *
**Line 104: its -> It is**

**Line 113: Mode(LRM) -> Mode (LRM)**

*these corrections have been added to the revised document*
* * *
**Line 48: I suggest you add the more local/regional climatologies such as Forsström et al. (2011), which provides snow depths for e.g. the Fram Strait, which represent an area outside the central Arctic where W99 climatology is not working properly due to extrapolation.**

**Forsström, S., Gerland, S., & Pedersen, C. (2011). Thickness and density of snow-covered sea ice and hydrostatic equilibrium assumption from in situ measurements in Fram Strait, the Barents Sea and the Svalbard coast. Annals of Glaciology, 52(57), 261-270. doi:10.3189/172756411795931598**

*We have added the reference of this climatology in the revised document.*

L54: The climatology of Forsström et al. (2011) provides snow depths in areas outside the central Arctic where the Warren- 99 climatology is not working properly.
* * *
**Line 130: AspeCt data including snow depths are available up to 2019 from Hamburg (https://icdc.cen.uni-hamburg.de/en/seaiceparameter-shipobs.html)**

220 *This point has already been considered in the revised document. Please see the corresponding answer in section : 1.1) Validation data*
* * *
**2.2) Data processing of ASD**

**Line 145: which is assumed to be reflected near the top of the snow pack, i.e. the air/snow interface**

225 *This modification has been added to the revised document*
* * *
**Line 150 Why do you refer to CryoSat-Baseline-C Product user handbook when you are using Baseline-B as stated in Line 153-154 ? See also general comments.**

*We refer to this handbook because we partly use the Baseline C (from 2017 to 2019). Please see answer to the general*
230 *comments section 1.2) CryoSat-2 Baselines. Furthermore, this handbook is valid for the Baseline B as there are only a few changes between the 2 baselines (and L1b waveforms are identical).*
* * *
**Line 159-162: You explicitly mention DTU15 MSS, but not the other models why is this the case ?**

*It is true that we specify the MSS but not the other corrections because : 1) it is the most important correction in the freeboard*
235 *calculations and 2) some of the other geophysical corrections are not exactly identical for SARAL and CryoSat-2. We believe that it would be confusing for the reader, without added value, to specify all the corrections and differences for both satellites as it has almost no impact on the freeboards (freeboards are calculated from a difference of heights).*
* * *
**Line 167-68: Does these limitations on PP apply to both Ku- and Ka-band ?**

240 *We apply exactly the same criteria for SARAL and CryoSat-2. To make sure this is clear for the reader we add the following sentence:*

L186-187: This criteria is the same for SARAL and CS-2.
* * *
**Line 170: Threshold First Maximum Retracker Algorithm (TMFRA; -> Threshold First Maximum Retracker Algo-**
245 **rithm (TFMRA;**

**Line 177: we need to take into account the decreasing of the Ku radar echo velocity when it penetrates into the snow pack -> we need to take into account the decrease of the Ku radar echo velocity as it penetrates into the snow pack**

Line 186: radarKu -> Kur (as you also made this for the snow sdr)

Line 192-193: In the Arctic, it also shows that these data are different from the Warren 99 modified (W99m) climatology, where nearly everywhere the W99m climatology exhibits thicker snow layers over sea ice. -> In the Arctic, it also shows that these data are different from the W99m climatology, where the W99m climatology tend to exhibit thicker snow layers over sea ice.

*These modifications have been added to the revised document.*
* * *
**External data sets**

Line 197-199: For both hemispheres, the time period of model and satellite products that have been used is explained in Fig. 2 (ranges from 2014–2019 for the Arctic, and 2013–2019 for the Antarctic with limitations for some data products for both hemispheres). -> For both hemispheres, the time period of model and satellite products ranges from 2014–2019 for the Arctic, and 2013–2019 for the Antarctic (with limitations for some data products for both hemispheres), as explained in Fig. 2.

Line 208: located near the Canadian Archipelago ( and the Beaufort Sea, and only -> located near the Canadian Archipelago and the Beaufort Sea, and only

Line 219: cover over FYI Because of this limitation, -> cover over FYI. Because of this limitation,

Line 222: Other months are only available on FYI -> Other months are only available over FYI

Line 232: description can be found in Schweiger et al. (2011) -> description can be found in Schweiger et al. (2011).

Line 238: Louvain-la-Neuve sea Ice Model (LIM2) (Fichefet and Maqueda, 1997; Vancoppenolle et al., 2012) -> Louvain-la-Neuve sea Ice Model (LIM2) (Fichefet and Maqueda, 1997; Vancoppenolle et al., 2012).

Line 230: OSISAF -> OSI SAF

Line 252: CRYOsat Validation Experiment -> CryoSat Validation Experiment

*The above modifications have been added to the revised document.*
* * *
Line 256: (Haas et al., 2017), -> (Skourup et al. (2019), Haas et al. (2017)), Please, also use DOI for the CryoVEx 2017 campaign data: https://doi.org/10.5270/esa-enocas0 - CryoVex-KAREN 2017 Campaign: "ESA CryoVEx/KAREN and EU ICE-ARC 2017 - Arctic field campaign with combined airborne Ku/Ka-band radar and laser altimeters, together with extensive in situ measurements over sea- and land ice" obtained from the ESA campaign site: https://earth.esa.int/eogateway/ca karen-2017?text=CryoVEx+2017

*These references have been added to the revised document. A reference to the technical report has also been added:*

L279: For further informations, please see the technical report at https://earth.esa.int/eogateway/documents/20142/1526226/CryoVEx2017- final-report.pdf.
* * *
**Line 260: OIB is one of the largest airborne mission in polar -> OIB is one of the largest airborne missions in polar**

*This modification has been added to the revised document.*
* * *
**Line 264: (Kurtz et al., 2012 (Updated 2015; King et al., 2015) I am not sure I understand this ?**

*An update of the Kurtz et al., 2012. has been made in 2015. For a better understanding we just kept Kurtz et al., 2012 in the revised manuscript.*
* * *
**Line 267: Why did you not include OIB from 2019 ?!?**

*We did not include OIB from 2019 in order to consistently compare with DuST and AMSR2B snow depths for which 2019 was not available. Please see section 1.3) Periods, for a complete answer to this comment.*
* * *
**Line 267-268: comparison with the ASD climatology, we compare with all OIB spring campaigns during 2009–2012. Did you use quicklook here or the IDCS4 ? It is not clear from section 3.3.2**

*We use the same quicklook OIB product version to compare with the ASD climatology. It is explicited by the sentence L273: OIB snow depth data presented in this paper are the NSIDC OIB Quicklook version.*
* * *
**Line 272: et al., 2006; Perovich and Richter-Menge, 2015) Snow depth -> et al., 2006; Perovich and Richter-Menge, 2015). Snow depth**

*This modification has been added to the revised document.*
* * *
**Comparison between snow depth data**

**Line 277: 500×500 EASE2 grid ... is the grid in km ? i.e. 500x500km ?**

*500x500 is the number of pixel of 12.5km (in the Arctic grid). To clarify this point, the sentence has been modified has it follows:*

L284: Model and satellite snow depth estimations are projected onto a 12.5km pixel size EASE2 grid similar of ASD using a linear two-dimensional multivariate interpolation.
* * *
**Line 281: of all snow depth monthly maps -> of all monthly snow depth maps**

**Line 294-296: Note, that when comparing the various snow depth data with the OIB snow radar data, at achieve a corresponding spatial scale, we have applied a 25 km window rolling mean for smoothing the OIB data. -> Note, that when comparing the various snow depth data with the OIB snow radar data, we have applied a 25 km window rolling mean for smoothing the OIB data, to achieve similar spatial scales.**

**Line 305: zones -> regions**

**Line 305: and higher snow depths over region of thicker sea ice -> and higher snow depths over regions of thicker sea ice**

*These modifications have been added to the revised document.*
* * *
**4.2.1 I do not like the phrasing "Annual means" for means calculated only based on March and April data, maybe rephrase this to seasonal or winter means**

*The phrasing Annual means has been replaced by « seasonal » means.*
* * *
**Line 319: comparable both in terms -> similar both in terms**

**Line 330: Although investigating which causes these discrepancies -> Although investigating what causes these discrepancies**

**Line 334: ASD is the only publicly available altimetric snow depth product in Antarctica. -> ASD is currently the only publicly available altimetric snow depth product in Antarctica.**

*These modifications have been added to the revised document.*
* * *
**Line 337: I would remove: "(equivalent to PIOMAS but for Antarctica)", as you have already mentioned this.**

*This sentence in brackets has been removed.*
* * *
**Line 342: east side, -> eastern part**

**Line 343: One relevant difference between these two data is -> One relevant difference between these two datasets is**

**Line 353: between -> from**

**Figure text 7: considered -> included**

**Figure text 6: during -> acquired on**

**Figure text 6: Remove: "2015 spring campaign)**

**Line 353: Fig. 6 shows an example the 29th March 2015 track -> Fig. 6 shows an along-track example from the OIB flight on 29th March 2015**

**Line 354: comparing with the OIB data over the entire 2014-2018 time period -> comparing the various snow depth products with the OIB data over the entire 2014-2018 time period**

*These modifications have been added to the revised document.*
* * *
**Line 354: Why do you not include OIB from 2019 ?**

*Please see the previous answers explaining why we have decided to discard these data.*
* * *
**Line 357: It is almost impossible to see the shading, you might want to upgrade the color in order to be visible**

*We have upgraded the red shading in figures 6,8 and 9 of the revised manuscript.*
* * *
**Line 357: data are nearly always included within the ASD envelope of uncertainties (in shaded red). - > data are almost within the ASD envelope of uncertainties (in shaded red) at all times.**

**Line 359: Remove: "always"**

**Line 361: overestimate OIB with a level comparable with DuST -> overestimate at a level comparable with DuST, when compared to OIB.**

**Line 365: MYI Kwok et al. (2017). -> MYI (Kwok et al., 2017).**

*These modifications have been added to the revised document.*
* * *
**Figure 6: Out of curiosity., what happens after C ? Here there are a huge difference between OIB/ASD/DuST, could be interesting to look further into this.**

*Trackpoint 20000 is a coastal region. The differences mainly come from the larger freeboard uncertainties in these zones.*
* * *
385 **Line 369: a less optimal solution -> a less optimal solution in this case.**

**Line 375: Fig. 8 and Table 2 show how the satellite mean values are closer to the in-situ observations than the model -> Fig. 8 and Table 2 show how the satellite derived mean values are closer to the in- situ observations than the models**

**Line 376: seasonal changes in the IMBs -> seasonal changes captured in the IMBs**

**Line 376: or -> nor**

390 **Line 377: localized -> point**

*These modifications have been added to the revised document.*
* * *
**Figure 8: Why did you not include the ASD uncertainties as shading in this plot similar to Figure 6 and 9?**

*Similar to Figure 6 and 9 we have added the ASD uncertainties as shading in Figure 8.*

395
* * *
**Line 387: Ka-ASR estimation exhibits very thin snow thickness (< 10cm).**

*This modification has been integrated to the revised document.*
* * *
**Line 389: Ka band radar -> Ku-band radar**

400 *This sentence does not refers to the Ku-band but to the snow depth computed from Ka-band. To avoid misunderstanding we have modified this sentence as it follows :*

L431: The main difference between the snow depths calculated from the laser and the Ka-band radar snow depth estimations seems to be a bias.
* * *
405     **Line 392: (in pink shading). -> similar as for Figure 7 it is difficult to see, and also in figure 7 you called I red shading, please, be consistent.**

*According to your suggestion, we always refer to "red shading" and the contrast of figure 7 has been increased for a better readability.*
* * *
410 **Line 394: It does not really make sense to use W99/W99m in these areas, but ok since I presume this is what is used in the SIT products**

*We agree that it does not make sense to use the W99 climatology in these areas. As it is used in SIT products, we still want to keep it in figure so that it emphasises the need not to use the W99 climatology.*

**Line 390: Why is there a gap in the ALS-ASR data ?**

*There is a gap because the laser is more sensitive to meteorological conditions, for instance the cloud cover, than the Ka-band altimeter. Some data are then missing. Note that this is also the case with ICESAT-2.*
* * *
**Line 390: Except around 71.3N, where a specific event may have occurred, the ASD snow product tracks the magnitude of the CryoVEx airborne data**

*This modification has been integrated to the revised document.*
* * *
**Line 390-392: I am not really convinced about this. But you are right, that**

*We did not fully understand this comment.*
* * *
**Line 391: KAREN-ASR**

**Line 395: The -> the**

*These modifications have been integrated to the revised document.*
* * *
**Towards an ASD snow depth climatology**

**Line 397: the temporal coverage -> the limited temporal coverage**

**Line 397: (only after 2013) -> (only available post-2013)**

**Line 399: Remove: "to this limitation"**

**Figure 10: the ASD and the Warren W99m climatologies considering all the tracks of 2009,2010,2011 and -> the ASD and the W99m climatologies considering all the tracks of 2009, 2010, 2011 and**

**Line 407: as obtained in Fig. 7 with the ASD data. -> as obtained by direct correlation between ADS and OIB measurements (Fig. 7).**

**Line 409: product for the years of Envisat -> product representing the Envisat-era.**

**Line 409. Remove: "Warren"**

*These modifications have been integrated to the revised document.*
* * *
**Impact of snow depth on SIT estimation**

**Line 425: Baseline C ?!? See general comment.**

*Please see our comment on section 1.2) CryoSat-2 Baselines.*
* * *
**Figure 11: The red and blue colours for the different representations are difficult to separate, i.e. ASD, DuST, SIT min-max. I encourage you to find another solution for this.**

*We have modified some colours and line styles in Figure 11 in order to increase it readability.*
* * *
**Line 431: taking into account the radar speed velocity decreasing which depends on the snow depth data -> taking into account the reduction of radar velocity within the snow layer, which depends on the snow depth data**
**Line 436: lower -> low**
**Line 439: comprised -> distributed Line 446: 4.103km3 -> 4 x 103km3 Line 449: 3.5.1015 -> 3.5 x 1015 Line 450: 2.108 -> 2 x 108**
**Line 459: maxdevy,m(SIT) -> (maxdevy,m(SIT))**
**Line 459: Eqs.(9) -> Eqs. (9)**
**Line 459 and onwards, be consistent to refer to equations in a similar way, e.g. Eqs. (9) or Eqs 9**
**Line 466: Eqs. 11 to 13. Ny is the number of years. -> Eqs. (11) to (13), where Ny is the number of years.**

*These modifications have been integrated to the revised document.*
* * *
**Line 471: I do believe Eq. (13) should be on the right side of the summation in Eq. (12) ?**

*It is exact. We have corrected this point.*
* * *
**Line 480: due only to snow depth -> caused by the snow depth**

**Line 481: We distinguish two cases: the case « obs snow products », which considers ASD, DuST, W99m and AMSR2B and the case « all snow products », which includes in addition the PIOMAS, MERCATOR and NESOSIM model solutions. -> We define two cases: 1) « obs snow products », which considers ASD, DuST, W99m and AMSR2B and 2) « all snow products », which includes in addition the PIOMAS, MERCATOR and NESOSIM model solutions.**

**Line 484: Remove: "For similar reasons, AMSR2-NSIDC is also not taken into account", as this has not been in the Arctic analysis**

**Line 485: Eqs.(11) to (17) -> Eqs. (11) to (17)**

**Line 488: (spatial mean of SITy,m,p maps are also included -> (SITy,m,p ) are also included**

**Line 489: at the ocean/ice transitions -> marginal ice zones**

**Line 495: snow depth from -> snow depth variability from**

**Line 495: an other -> another**

**Line 499: 76cm -> 76cm**

**Line 508: for constraining models -> to constrain models**

**Line 509: initiative -> initiatives**

**Line 512: Polar Week -> Polar Science Week**

*These modifications have been integrated to the revised document.*
* * *
**Conclusions and Discussions**

**Line 519: and a CryoVEx airborne campaign -> and CryoVEx airborne campaign data Line 519: Ka-Ku snow depth -> Ka-Ku and ALS-Ku snow depths**
**Line 523: always exhibits higher snow depth patterns -> is always biased high.**
**Lone 530: always far lower the -> always biased low when compared to the other products. Line 531: in -> on**
**Line 540: Weddell sea are not represented -> Weddell sea are not captured by the model Line 554: would need to be done -> is needed**
**Line 557: versions, etc...), -> versions),**
**Line 560: is its temporal coverage, imposed by SARAL (only from 2013). -> is its limited temporal coverage, imposed by SARAL which is only available since 2013.**

*These modifications have been integrated to the revised document.*
* * *
**Line 563: Please add: which strongly overestimates snow depths, and is useless outside the central Arctic due to extrapolation.**

*We have added this sentence.*
* * *
**Line 564: 2009-2013 ... wasn't it 2009-2012 ?**

*Indeed it is 2009-2012. It has been corrected*
* * *
**Line 565-566: has a mean bias of about 14 cm with -> on average is biased  14 cm high with Line 568: could -> should Line 570: pacts of the snow depth on SIT -> pacts of different snow depth products on SIT Line 575: 14.2% -> 14**

*These 2 modifications have been added to the revised document.*
* * *
525 **Line 576-577: Please, add some of the locations here.**

*Higher impacts are mainly in coastal and marginal zones. We modified the sentence as it follows:*

L638-639: Deviations between SIT estimations reached up to 77 cm (55% of global mean SIT) in marginal and coastal zones.
* * *
**Line 584: uncertainty -> uncertainty budget.**
530 **Line 585: which still to be further investigated -> which still needs to be further investigated Line 587: or -> and/or**

*These 2 modifications have been added to the revised document.*
* * *
**Data availability: It is nice with a link to the data set presented in this paper. In some journals you need to link or provide information about all the data sets used, but I am not sure whether this also is the standard for TC, or whether**
535 **the links provided in the text is enough.**

*There is already a link in the paper to access the data. In addition, the data link has also been added L146 of the revised document.*

**References**

Grosfeld, K., Treffeisen, R., Asseng, J., Bartsch, A., Bräuer, B., Fritzsch, B., Gerdes, R., Hendricks, S., Hiller, W., Heygster, G., et al.: Online
    sea-ice knowledge and data platform< www. meereisportal. de, Polarforschung, 85, 143–155, 2016.

H. Skourup, A. V. Olesen, L. S. S. S. S. S. M. H. N. H. A. F. O. A. C. K. M. V. H. R. S. L. R. F. A. E. H. I. O. A. S. C. H. and Wilkinson, J.:
    Arctic field campaign with combined airborne Ku/Ka-band radar and laser altimeters, together with extensive in situ measurements over
    sea- and land ice, https://doi.org/10.5270/esa-enocas0-CryoVex-KAREN2017Campaign, 2019.

Haas, C., Haapala, J., Hanson, S., Rabenstein, L., Rinne, E., and Wilkinson, J.: CryoVEx 2006: field report, 2006.

Helm, V., Hendricks, S., Göbell, S., Rack, W., Haas, C., Nixdorf, U., and Boebel, T.: CryoVex 2004 and 2005 (BoB) data acquisition and
    final report, Alfred Wegener Institute, Bremerhaven, Germany., 2006.

Hvidegaard, S. M., Forsberg, R., and Skourup, H.: Sea ice thickness estimates from airborne laser scanning, Sea Ice Thickness: Past, Present
    and Future, pp. 193–206, 2006.

Kern, S.: ESA-CCI_Phase2_Standardized_Manual_Visual_Ship-Based_SeaIceObservations_v02, https://doi.org/10.26050/WDCC/ESACCIPSMVSBSIC
    https://doi.org/10.26050/WDCC/ESACCIPSMVSBSIOV2, 2020.

Skourup, H., Hvidegaard, S. M., Forsberg, R., Einarsson, I., Olesen, A. V., Sornsen, L. S., Stenseng, L., Hendricks, S., Helm, V., and
    Davidson, M.: CryoVEx 2011-12 Airborne Campaigns for CryoSat Validation, 20 Years of Progress in Radar Altimatry, 710, 98, 2013.

Willatt, R., Laxon, S., Giles, K., Cullen, R., Haas, C., and Helm, V.: Ku-band radar penetration into snow cover on Arctic sea ice using
    airborne data, Annals of Glaciology, 52, 197–205, 2011.

Worby, A. P., Geiger, C. A., Paget, M. J., Van Woert, M. L., Ackley, S. F., and DeLiberty, T. L.: Thickness distribution of Antarctic sea ice,
    Journal of Geophysical Research: Oceans, 113, 2008.

---

## Author Comment (AC2)

5

**Title: Advances in altimetric snow depth estimates using bi-frequency SARAL/CryoSat-2 Ka/Ku measurements.**

Florent Garnier, Sara Fleury, Gilles Garric, Jérôme Bouffard, Michel tsamados, Antoine Laforge, Marion Bocquet, Rénée Mie Fredensborg Hansen and Frédérique Rémy

**Referee#2 global comment**

The manuscript "Advances in altimetric snow depth estimates using bi-frequency SARAL/CryoSat-2 Ka/Ku measurements." by Garnier et al. shows interesting results on satellite derived snow depth on sea ice in both the Arctic and the Antarctic. This study is timely and relevant within the context of changing Arctic and Antarctic sea ice regime and how snow depth in particular

10 is a big challenge to retrieve from satellites. However, I found the manuscript not ready to be immediately accepted to TC due to the following brief reasons (detailed comments to follow), and therefore recommend Major Revisions. I am willing to review the revised manuscript. Please note that authors have the right to accept/follow or reject my comments, as the comments are from my scientific perspective and welcome to be challenged.

**Answers to the anonymous referee#2 comments**

15 We would like to thank the reviewer for her/his careful reading of the manuscript and for the relevant and constructive remarks. In order to fit with your global comment, we have made an complete revision of the manuscript that should have considerably improved the quality of the writing and the readability of the document. We hope that the modifications we have made will meet your requirements. Please find below the details on how your general and specific comments have been taken into account. In this « author's response document », the referee's comments are in bold type, the answers are in italic type, and the corrections 20 to the revised manuscript are in normal type.

Answers to the ananymous referee#2 : general comments

a) Since the study exclusively focuses on snow depth retrievals, I expected the authors to have a dedicated discussion on how snow geophysical properties in general has an effect on the retrieval biases. This is crucial since snow on FYI and MYI exhibit variations in their geophysical characteristics. Between Arctic and the Antarctic, there is almost a contrasting difference in snow covers (for e.g. substantial flooding on Antarctic sea ice and resultant slush/snow-ice formation that can hamper snow depth retrievals). Unfortunately, I did not see any of these critical discussions affecting the retrievals. In general, I found the authors neglecting snow properties impacting snow depth retrievals. This is clear in both introduction, results and discussions.

30 The authors simply washed out the 6.5 cm bias between ASD and DuST, due to recalibration issues. But I do not think that is the only reason. Although I appreciate authors briefly discussing the effect of snow properties (lines 580 to 590), I think they need to be 'tied' with what the biases show both in the Arctic and the Antarctic.

Moreover, it is right now, difficult to differentiate between the results from the Arctic and the Antarctic in the discussion section. I would strongly suggest to have sub-sections for the discussion and NOT to merge with the conclusions. The new

**35 discussion section could have discussing results separated by hemispheres, discuss results based on bias from different products, and also discuss the impact of snow properties on these SD retrievals.**

After discussions with co-authors, we have decided to keep a single "Discussion and conclusion" section. The main reason is that adding a full section on snow properties would confuse the main purpose of this article. The objective is to present our new snow depth product from altimetry (ASD) in order to enhance future studies that would, among others, analyse the impacts

40 of snow properties. In order to be sure that this is clear from the beginning of the manuscript, we have modified the last part of the introduction (please see answer to comment b) )

We think that the ASD product could definitely help to better characterize snow properties but the analysis is very complex and would need a dedicated article. Anyway, we agree that snow properties are a key point that was not enough mentioned. For that purpose, we have added several references on snow properties in the revised document (of course also in the last section). Note that we have also better separated the results in the Arctic from those in the Antarctic in the last section.

Since both DuST and ASD products are based on the difference of the same Saral Ka-band and CryoSat-2 Ku-band data, the deviations between the two datasets should mainly come from the difference of the methodology used to compute snow depth. For instance, DuST is constructed from SAR mode data while ASD from pLRM data. The bias between LRM and SAR due to roughness is corrected from a calibration with OIB. Nevertheless, we agree that a thorough examination could reveal correlations with snow properties. These points have been added several times in the revised document. For instance :

L227-230: However, unlike the ASD product (which uses the CS-2 PLRM data, see Sect. ??), the DuST product uses SAR mode freeboard estimates of CS-2. The CS-2 and SARAL data are then calibrated against data from National Aeronautics and Space Administration's (NASA's) OIB airborne measurements to align the freeboard observations from SARAL/AltiKa with the snow/air interface, and the freeboard observations from CS-2 with the snow/ice interface.

55

60

75

45

50

b) Overall, I found the paper a bit difficult to follow, especially with a lot of grammatical errors, use of unsual sea ice/remote sensing terminologies (e.g. emerged fraction of sea ice, sea ice sinking etc...), and difficulties to follow the figures. Also, it was a bit difficult to follow the paper objective (lines 125 to 135). Although, I understood what the objective is, it needs to be stated clearly in these lines. I totally understand there are a lot of datasets, models and climatologies to compare and report to, it may be easier for a reader to follow the objectives if the datasets are not included in the objectives description, but keep them separate. Just a suggestion.

We apologize for the editorial oversights in the manuscript. We have proofread the manuscript thoroughly to correct as many grammatical errors as possible. Also, In order to increase the clarify and better define the objectives of the paper we have modified the end of the introduction as follows:

65 L 132: A recent study of Zhou et al. (2020b) presented an inter-comparison of available snow depth products from re-analyses, passive radiometry and altimetry (DuST). Similar, this paper reviews the state-of-the-art by comparing current main snow depth estimations. Yet, the main objective is to present and assess the upgraded version of the ASD product (see Sect. 2), covering the 2013-2019 period in both hemispheres. The article fits in with the upcoming HPCM CRISTAL mission (Kern et al., 2020) and aims to demonstrate the potential of such snow depth data to further specific studies on, for instance, improved sea ice volume estimations, freshwater budgets, snow properties or data assimilation. Except from an analyse of the impact of the snow depth

uncertainty on SIT retrieval, it doesn't explicitly address these questions.

The paper is organized as follows:

- First, we detail the methodology to process the ASD product and present all the datasets used in this study.
- We compare ASD with the main other existing snow depth satellite and model data in both hemispheres.
  - The datasets are then assessed against OIB, CryoVEx and IMB validation data in the Arctic.

- To circumvent the temporal limitation induced by SARAL, we propose a preliminary snow depth climatology based on ASD.
- The last section aims to quantify the SIT level of uncertainty due to the snow depth from an ensemble of SIT estimations calculated from the satellite and model snow depth datasets presented in the previous sections.
- We finally discuss and conclude, emphasizing current needs for snow depth data in sea ice studies.

c) There are a lot of acronyms in this paper, which makes it difficult to follow. Suggest to make a list of acronyms table so that the readers can follow?

85 As suggested we have added a list of acronyms table (table A5). We now only explicit in the text the most important acronyms, which reduces multiple line-long passages put into parenthesis.

**Answers to the Referee#2: Detailed Comments**

Abstract

80

90 Line 5-6: Maybe its just me, but when I talk about dual-frequency penetration of Ka- and Ku-band frequencies from air/snow and snow/sea ice interfaces, I write Ka-band SARAL/AltiKa and Ku-band CryoSat-2, than the other way around with CryoSat-2 written first. Also, note frequency followed by the satellite, than vice-versa.

These corrections have been made in the whole revised document.

**95 Line 11: Anyway, the authors mention all model and satellite data. So why not also mention in situ IMB and airborne OIB in the validation sentence.**

We agree that it is better to also mention all the validation data. We have modified the sentence as follows:

L11-12: The ASD product is further validated in the Arctic against the Ice Mass Balance (IMB) buoys, the CRYOsat Validation EXperiment (CryoVEx) and Operation Ice Bridge's (OIB) airborne measurements.

100

**Line 12: "space and time patterns" sound unusual in the context. Suggest to use 'a consistent spatiotemporal snow depth solution'**

We have modified the sentence as follows:

L13: These comparisons demonstrate that ASD is a relevant snow depth solution, with spatiotemporal patterns consistent with those of the alternative Ka/Ku DuST product, but with a mean bias of about 6.5 cm.

**Introduction**

The authors provide a good overview of snow depth products that are available from satellite data, in situ and airborne data and models/reanalysis. But there are lit review about the uncertainty caused by snow properties affecting snow

110 depth retrievals (although I noticed few lines about it in the discussion section). This is important to address as the Ka- and Ku-band penetration and differences has a strong sensitivity to snow property and their spatial and temporal variability.

As suggested in the general comments we have added several mentions to snow properties in the revised manuscript.

**115 Line 24: no need to capitalize 'Sea Ice Thickness'**

This comment has been taken into account in all the revised document.

Line 25: What do you mean by emerged fraction of sea ice? I am sure you meant fraction of sea ice above sea ice? Keep it simple.

120 In order to reflect this comment we have modified the sentence as follows:

L26: The principle is to measure the fraction of the sea ice above the sea level...

**Line 26: definition of leads is implied in sea ice. Suggest to delete phrases in brackets.**

We agree that the definition of leads is implied in sea ice but we would like to keep the introduction easily understandable for potential reader that are not familiar with it.

Line 30: remove 'the' before snow depth. Use SD right after snow depth. Add 'adding to the overall SIT uncertainty'

We have modified the sentence L30 according to your comment.

**130 Lines 31-33: Not just snow loading and speed effects, but also affecting snow properties (both geophysical and thermodynamic) from freeze-up to melt-onset.**

We have underline that snow properties impacts snow depth measurements:

L32-37 : For example, it is necessary to account for the snow loading (Laxon et al., 2013) and for the decrease in altimetric radar speed as it penetrates into the snow pack (Kwok and Cunningham, 2015; Mallett et al., 2020). In fact, variabilities of snow properties affect radar signals and then the snow depth measurements. More generally, snow cover has strong impacts on the sea ice (e.g., Massom et al., 2001; Powell et al., 2005; Bin et al., 2008; Sturm and Massom, 2009; Ricker et al., 2014) that affects the entire climate system (e.g., Ingram et al., 1989; Ledley, 1991; Eicken et al., 1995; Singarayer et al., 2006).

Lines 33-40: I think its better to move this section towards the beginning of intro, as it fits well with the general importance of snow on sea ice. If authors plan to do so, please also add how snow affects volume budget calculations, light 140 penetration affecting primary productivity and impacting sea ice stability affecting migration and hunting.

Following your comment, we tried to put this part at the very beginning of the introduction but we found that it breaks the logical progression of the introduction. Also, this part is still at the beginning of the introduction. However, we have taken into account your comment and add the following sentences :

145 L40-46: Such processes of sea ice formation and melting govern sea ice physical and chemical properties that impact the biological processes in sea ice (Van Leeuwe et al., 2018). The vertical distribution of light under sea ice that control biological processes and biogeochemical fluxes is also strongly linked with the snow depth (e.g., Perovich, 2007; Arndt and Nicolaus, 2014; Arndt et al., 2017). In addition snow accumulated over sea ice from precipitations represent a tank of freshwater likely to be carried into the ocean. Recent increasing of seasonal ice has promoted the amount of snow water discharge in the ocean which impacts the freshwater budget (Andersen et al., 2019; Overland et al., 2019).

150

**Line 39: air/ice drag and not the other way. Also what surface roughness are you mentioning? snow surface or ice surface or both?**

We have modified the revised document as you suggested. We are mentioning the snow surface roughness.

155 L46: Snow cover also modifies surface roughness that impacts the air/ice drag coefficient and transfer coefficients of latent and sensible heat fluxes (Andreas et al., 2005).

**Line 41: remove 'the' before snow cover on sea ice. Check throughout the manuscript. Its not the snow cover that is unknown. Its the snow depth. Correct?**

We have removed "the" before "snow cover" in all the revised document. We have replaced snow cover by snow depth 160

**Line 42: Move Warren et al. 1999 from Line 44 to after W99m in Line 42.**

This modification has been done.

**Line 45: I wouldn't call in outdated. That's a bold statement considering that W99 is still a baseline snow depth data. I 165 would call W99 data to be old. That's all.**

To demonstrate the weakness of the W99 snow depth climatology is an important feature of the article. We would rather keep the word outdated since W99 is constructed from data prior to the first impacts of the climate change. Because of the fast and recent strong evolutions of the Arctic sea ice, W99 does not fit with the current period.

**Line 48: remove apostrophe from all years.**

This correction has been taken into account in the revised document.

Lines 50-68: This section could be cleaned, separated by hemispheres. RIght now, both Arctic and Antarctic lit review are mixed up. Reviewer 1 also mentioned about references that needs to be changed and so I do not repeat the comment. In addition to IMB as an in situ data, there should be mention about snow depths from magnaprobes. There are many campaigns from NP drifting stations, SHEBA, CryoVex, even recently concluded MOSAiC campaigns, where magnaprobe-derived snow depths have been integral in validation.

In the revised manuscript, this paragraph only refer to the Arctic. The following paragraph focus on the Antarctic in which we have carefully considered the comments of referee#1. In addition, we have added a mention to the MOSAIC expedition and the ICEBIRD campaign as follows:

L75-77: From September 2019 to October 2020, the international MOSAIC expedition monitored the central Arctic (Shupe et al., 2020) providing, among others, data on snow precipitation, snow water equivalent (SWE) and magnaprobe snow depth measurements (e.g., Munoz-Martin et al., 2020).

185 L87: Future validations in the Antarctic will also benefit from the recent AWI IceBird campaigns carrying a snow radar since 2019 (https://www.awi.de/en/science/climate-sciences/sea-ice-physics/projects/ice-bird.html).

Lines 70-124: This section could also be cleaned, separated by hemispheres. Right now, both Arctic and Antarctic lit review are mixed up. What I miss from lines 50-124 are the overall limitations of these datasets/methods/retrievals that has led to motivating your study. It would be good to clarify that, before moving ahead to objectives.

The logical progression of the introduction is by type of observation. At level 1, we separate the validation data, the model and the satellite data. Within these "pseudo-sections", we then separate (as far as possible) the Arctic and the Antarctic. We agree that another progression could have been chosen. Our goal was to consider ASD as "a snow depth products for both hemispheres", even if we do understand that it is, right now, only validated for the Arctic. Since a separation by hemisphere almost need a full re-writing of the introduction we propose to keep this way. However, in order to consider your comment and

195 almost need a full re-writing of the introduction we propose to keep this way. However, in order to consider your comment and to avoid confusions between the two hemispheres, we have modified some sentences and the order of some paragraphs. The main limitations are the space and time coverage (due to SARAL) and the hypothesis of the ku-band and ka-band dominant scattering surfaces (related) to snow properties. We clearly present these hypothesis in the revised manuscript

L159-161: Note that the validity of this hypothesis is strongly linked with the space and time variability of snow properties such as snow density, grain sizes, the liquid water content or the surface roughness.

L163-164: Note that SARAL orbit is limited to 81.5° in latitude, which reduces the coverage of snow depth data in the Arctic.

Lines 125-136: As mentioned earlier, the objectives are a bit too specific and confusing. It would be nice to keep an overarching objective, followed by sub-objectives based on datasets and methods used.

205 We have re-written the objectives and the plan of the manuscript in the revised version. Please see our response to general comments.

**Line 126: datasets is one word.**

We have written "dataset" (in one word) in all the revised document

210

220

**Line 127: The main part.... Remove this line, reads redundant to lines 125 and 126**

we have removed this sentence.

**Line 128: The ASD product... Aren't you developing an updated version of ASD product? So before comparison with other data, you should showcase the upgraded version correct? Moreover all the AMSR-2, DUST, OIB, CryoVex and 215 IMB are all 'derived' datasets. Please indicate that.**

Indeed the ASD product presented here is an updated version of the product presented in Guerreiro et al. (2016). We do agree that first compare the two version and explain the differences would have been a consistent approach. However, the assumption on the difference of penetration between Ka-band and Ku-band is nearly the only common point between these 2 products. For instance, in the previous version of ASD, snow depth is only calculated at crossing points of SARAL and CryoSat-2 trajectories.

Monthly maps are then an extrapolation of a few snow depth data. Also the pLRM CS-2 data were a prototype version computed by CNES.

Of course we have first investigated the deviations between the 2 products. It appears that the results are quite different, with significantly higher gradients in the previous version but it is difficult to draw some conclusion from the differences. Is it

mainly due to extrapolations? is it due to monthly mean freeboard smoothings? along track better consistencies? Note that, 225 in addition, the previous ASD version hasn't been computed as a product for users but more as an experimental dataset for the work of a PhD. The data have been only computed for two winters (2013-2015) and only cover the central Arctic. Since the article is already quite long and compare a larger amount of data we have chosen to avoid this comparison that,

for our point of view, does not bring much information. Then apart from your imperative requirement for publication, we would prefer not to put more information concerning this previous product in this article.

230

**Lines 129-131: I guess reviewer 1 already makes a strong point about validation of Antarctic snow depth data.**

We have carefully taken in to account the reviewer#1 comments to better specify the validation data in the Antarctic. For instance, instead of stating that OIB data are not available, which is not entirely true, we mention that only L1b data are available in the Antarctic. Please see answer to referee#1 for more informations. 235

Line 134: First discuss and then conclude, not the other way around. Also suggest a separate discussion section.

Please see our response to general comments

**240 Data Processing of ASD**

**Line 140: provide the data link here**

The data link has been added to the proposed location in the document.

Lines 144-148: Introduce AltiKa here. Moreover, I found this a bit awkward. Ka-band assumed to be "reflected near the top of the snow pack" and Ku-band "near the snow/ice interface" are NOT EQUIVALENT to "air/snow" or "snow/ice" interfaces. This needs to be addressed as "near" can be either below or at the air/snow interface or above or below the snow/ice interface. Maybe its just a phrasing issue, but its a mistake in terms of radar scattering assumption.

We do agree that this phrasing might be subject to misunderstanding. The sentence as then be modified as follows:

L155-157: Snow depth calculation is based on the difference of penetration between the Ka-band range altimeter of SARAL
(which is assumed to be reflected at the air/snow interface) and the Ku-band range altimeter of CS-2 (which is assumed to be reflected at the snow/ice interface).

**Lines 148-155: You talk about Baseline C product handbook although Baseline B products are used. Clarify**

We use the GOP CryoSat-2 Baseline B until 2017 and Baseline C from 2017 to 2019. The reason why we did not used the baseline C for all the time period is simply because the Baseline C was not available at the time we had processed the data. It is now available but it would need some times to reprocess all the data. Because the waveforms are exactly the same in the two Baselines (and then the computed snow depths), we have decided to keep this version for this article. Note that we are currently re-processing the data so that the ASD product will be computed over the 2013-2020 period in both hemisphere from the baseline C only. The only difference is then that it will also cover the SARIN mode zones. You are right that it is important to let the reader (and probable future users be aware of this). For that purpose the sentences L151-155 of the article has been

modified as follows :

L167-170: Since the Baseline-C PLRM GOP product was only available from 2017 at the time we have computed the ASD data, we have used the Baseline B for the period 2013-2016. It does not impact ASD since we use only the L1b product levels which have identical waveforms on both baselines. However, the baseline B does not include the SARin data. Then, the ASD data does not yet cover the SARin mode zones. The next version of the ASD product will include SARin mode zones.

265

**Lines 156: Use Ku-band and Ka-band**

We have carefully modified the document in order to always use the spellings Ka-band (and Ku-band).

**270 Line 169-185: Its TFMRA, not TMFRA.**

We need to take into account of 'decrease' in velocity of Ku-band waves, as it penetrates through the snow pack.

These mistakes have been corrected in the revised version.

275 Also, epsilon can be mistaken as dielectric permittivity, therefore suggest to change the uncertainty symbol to something else.

We now use the symbol  $\delta$  instead of  $\epsilon$  in the revised version of the manuscript.

The uncertainty calculation seems to be estimated dependent only on the snow density aspect, and not taking into account of other snow properties such as temperature, salinity or microstructure.

Especially in the Antarctic, where the ASD product uncertainty is dramatically low (only up to like 5 cm?). That's a bit odd since, as authors would know, that many of the Antarctic sea ice sectors are flooded, especially in the Weddell or the Bellingshausen sectors. The slush layers that form itself induces significant retrieval errors in snow depth. Considering that, the low uncertainty values needs to be rechecked, since the calculation is just based on snow density estimates. Makes sense?

**Ma**

285

We totally agree that the methodology to compute uncertainties might be subject to comments. However, the methodology we used is quite common in altimetry. Ricker et al. (2014); Tilling et al. (2015); Di Bella et al. (2018) use very similar methodology to calculate CryoSat-2 and SARAL freeboard uncertainties (uncertainty on snow depth is derived from uncertainty

- 290 on freeboards). The methodology is not to explicitly consider the various source of uncertainties such as snow/ice properties. Note that a lot of other source of uncertainties are not considered such as the use of a fixed threshold for the retracker, the lead/floe classification criteria, the SLA interpolations under floes.... The various sources of uncertainty are considered all together and calculated from the statistical dispersion of the the sea level anomaly (SLA) within a radius of 25km. We assume that within this radius SLA should be identical. Dispersion of SLA values is then the measurement of uncertainties.
- 295 We clearly agree that this methodology should underestimate the uncertainties and need to be refine as flooding, slush layers and snow properties should clearly impact the Ku-band and Ka-band signals However the quantification of their impact on snow depth retrieval is a work in itself and is clearly out of the scope of this study. Note that we are currently working within the ESA FDR4ALT project on methodology to improve uncertainty estimates. Hopefully, refined methodology to refine uncertainties should come in the next years.

300

Also interesting are no data points within the CAA, except for few points around the Hudson Bay region? Please clarify.

This is because we did not include the SARIN mode zones of CryoSat-2. Please see our previous answers at the beginning of this section for more details.

305 Lines 190-195: Very vague description of results from Figure 1. Keeping in mind that other products are to compared against ASD, the explanation of Figure 1 is weak. Also to note that this is the first time ASD is produced in the Antarctic. Although, there are a multitude of uncertainties, I would expect the authors to expand on the ASD results in the Antarctic and discuss the biases.

The description of the ASD features is rather of the next sections. However, in order to take your comment into account we have added the following sentences:

L 208: In the Antarctic, thicker snow is located in the Weddell sea and, in a less extend, in the coastal zones of the bellingshausen and Amundsen seas. It is also relevant to identify the very low snow values associated with the Ross ice shelf. In the Arctic, the

snow distribution follows well the dynamics of sea ice, the most characteristic of which is the export of MYI in the Beaufort gyre. This figure also shows that the ASD data are very different from the W99m climatology, where the W99m climatology tend to exhibit thicker snow layers over sea ice.

Satellite Data

**DuST: Similar 'to'. I see SARAL everywhere although, AltiKa is the sensor. I would suggest to replace SARAL with AltiKa, everywhere. Move 'The DuST data are....'' to the beginning of the section.**

As proposed, we have moved the sentence to the beginning of the section. Also Since we use the name of the satellite for CS-2, it is better to also do it for SARAL (for homogeneity). However, in order to clarify the name of the sensors we have modified the following sentence : L155-158: Snow depth calculation is based on the difference of penetration between AltiKa, the Ka-band

range altimeter of SARAL (which is assumed to be reflected at the air/snow interface) and SIRAL, the Ku-band range altimeter of CS-2 (which is assumed to be reflected at the snow/ice interface).

**325 AMSR-2: Since both DuST and AMSR-2 products are derived from different spatial resolutions, there needs to be an explanation about how these product resampling is carried out.**

The gridded model and snow depth monthly data are projected onto the EASE2 grid of ASD using a simple two-dimensional multivariate interpolation (griddata function in python). We have added this specification in the revised manuscript:

L297: Model and satellite snow depth estimations are projected onto a 12.5km pixel size EASE2 grid (similar of ASD) using a linear two-dimensional multivariate interpolation.

**Model Data**

One of the critical comments I have in this section is why authors choose three models to compare. I couldn't find any rationale for this. Is it because one model product is better than the other or does one have a better coverage? Please clarify.

**I guess MERCATOR also gives snow depth in the Antarctic? Clarify.**

We are not sure to fully understand the comment.

We have chosen 3 of the main sea ice models to perform the comparisons. The recent SnowModel-LG is not studied but is part of the study of Zhou et al. (2020a)). There might be others but we thought that it is sufficient to perform relevant comparisons.

Indeed, the MERCATOR model provides snow depths in the Antarctic. It is part of the manuscript. Maybe you meant NE-SOSIM? We have discussed with Alek Petty and, at the time we made the study, NESOSIM was not computed in the Antarctic.

**345 Validation Data**

For all validation datasets, right now, it is a bit confusing from where in the Arctic and in the Antarctic, the data are collected from. Although, authors do mention when data was collected, they don't mention from where. I would strongly suggest to have a hemispherical map showing the flight tracks and locations, or even a table with location and

**coordinates would be very useful for this paper and also a good source of citation for data tracks. This is applicable to both Cryovex, OIB and IMB datasets.**

The maps showing the location of flight tracks are provided with the figures. We have added one sentence to specify this point :

L269 : The validation data presented in this section are only compared with ASD in the Arctic. As mentioned in the introduction CryoVEx and OIB data in the Antarctic are still only L1b. IMB are only in the Arctic. The geographical location of the validation data are shown in the section 4.3.

**355 4.1 Methodology**

**Line 277: 500 by 500 km?**

It is 500x500 pixels of 12.5km It has been clarified in the revised manuscript.

Line 281: climate annual mean : average of all snow depth monthly maps from all years or average of every month for all years ?

the climatic annual mean is the average of all snow depth monthly maps from all years. We have added this specification to the revised manuscript:

L303: 1) The climatic annual mean, which is simply the average of all monthly snow depth maps from all years.

**365 Line 282: .. average of snow depth annual standard deviation? Also, previously snow depth was referred as SD, here it is shown as sd. Please correct.**

We now always refer snow depth as sd in the revised version of the document. For each year, we compute at each grid point the standard deviation map of the 6 monthly mean maps. The MAV is the mean of these standard deviation maps. We have modify as follows:

370 L 304: The Mean Annual Variability (MAV), is the average, on the years y, of the annual snow depth standard deviation  $(std_u(sd))$  maps computed from the 6 monthly mean maps.

Line 297: I guess reviewer 1 already mentioned about snow depth data in the Antarctic. So would be useful to incorporate that in the revised version and analysis (if data is publicly available).

We have carefully integrated all the references proposed by reviewer#1 in the revised manuscript.

Line 287: I didn't understand what Ny means. The authors say, number of year in the considered time period. It will be 1 always correct? A bit confused.

375 It is exactly the number of years in the time period considered for the calculations of the mean annual variability and the mean interannual variability. For instance to calculate the mean annual variability over the 2014-2018 time period, we have 5 winters  $\rightarrow Ny = 5$ .

**4.2.1 Results in the Arctic**

In the following sections, my comments will be more detailed and less on grammatical issues (will wait for the next round of review to work on them).

385

390

Lines 300-314: Ok, now here is my issue with Figure 3 ASD and DuST mean and standard deviation towards the East Siberian Sea and east of Baffin Bay. The authors mention about thin ice in these regions (btw its not Queen Elizabeth Islands, they are far north of where the snow depths are shown here), and thinner snow pack in these regions. But please keep in mind that the temporal window you have chosen is March and April when the ice thickness is the maximum. Food for thought and an analysis to revisit. Why does the AMSR-2 data shows thicker snow in these regions? A repeated question on why there are no CAA data shown (even for AMSR-2 data?). Please clarify.

We think there is a misunderstanding about geographical locations. Looking at the map in figure 1, we see that the Queen Elizabeth Islands are between the north Groenland and the Banks island. We mentioned thick ice (and thick snow) in these regions, not thin snow.

**395**

415

We just specify that we have not "chosen" March and April. Figure 3 are March and April bi-monthly maps because the AMSR2B data are only available for these 2 months (over the entire Arctic zone). In addition DuST use OIB data for calibration which only occurs in March and April.

400 As mentioned in a previous comment we do not have CAA data because we do not yet included snow depths from SARIN mode. The next ASD version will. These areas are yet covered by AMSR-2 but data are masked in order to provide statistics with the exact same data for the different products.

We are not sure that we can possibly assert why AMSR-2 has higher value. We assume that it might be linked with the difficulty to retrieve thick snow (MYI has a spectral signature comparable to snow depth) and the calibration with OIB. Over MYI, or thick ice, even the upgraded algorithm to derive snow from brightness temperature is not fully adapted and lead to overestimations. This assumption is reinforced by the results in Antarctica which show some very thick snow patterns like in the weddell sea that looks quite unrealistic (maybe these zones have MYI).

The thinner snow pattern running along the Alaska coastline from cap Bathurst is represented in all products but looks 410 more pronounced in the ASD data. This looks better in agreement with the Beaufort gyre circulation and the McKenzie shelf conditions.

Lines 320-330: An issue here with the analysis is that authors blindly talk about ice types here, but there is no data or discrimination of ice types shown here. I understand an additional plot would add to analysis complexity, but it would be ideal to have an ice type product like OSI-SAF to relate the snow depths to (as a function of ice types). Then statements such as 'patterns of deep snow over MYI would make better sense.

It is true that we do not highlight MYI and FYI in maps. However MYI and FYI distinction is a very common feature in sea ice studies. Scientific publications very rarely present such maps. It is possible to add a figure presenting OSI SAF sea ice types but we think that the article is already very long and include a lot of supplementaries. For now we prefer not to add such a plot.

**420 4.2.1 Results in Antarctica**

The first line of this paragraph is the impactful aspect to this paper. Yes, it is the only snow depth product. But I am a bit disappointed with how the results are discussed in this section. Both AMSR and GIOMAS shows strong regional-scale snow depth variability, as compared to almost very low in the ASD product. This strong bias between radar altimetry, models and passive microwave radiometry opens up a big avenue of discussion towards the altimetry retrieval chal-

Figure 1. Map of the Arctic to clarify geographical locations

- 425 lenges in snow depth on Antarctic sea ice. I think authors have shown the capability of ASD in the Antarctic (and Congrats for that), but needs strong foundational explanation of why this strong bias is occurring. Like I mentioned earlier, one of the big differences is the thick snow in certain sectors that cause reduced penetration of Ku-band microwaves, causing these biases. Think about it and maybe explain potential reasons causing these biases. It would be unfortunate and a waste of analysis if these are unaccounted for.
- 430 In order to fit with your comment we have added the following sentences in the section :

L373-380: The presence of thicker snow in East Antarctic for ASD is consistent with Worby et al. (2008) which comparisons with ARISE in-situ data have shown radiometric measurements snow depth underestimations. Wet-snow conditions due to flooding might be responsible for radiometric brightness temperature contrasts. The low variabilities in the Weddell sea for ASD were not expected since winter snow properties are extremely variable (Massom et al., 1997). In addition the large bias with AMSR2-NSIDC raises questions. Since Ku-band is supposed to better penetrate in cold and dry snow (Willatt et al., 2011), one hypothesis is that the presence of saline and warm moist basal snow layers, even in the absence of flooding, (Massom et al., 2001; Perovich and Richter-Menge, 1994) lead to ASD underestimations. For AMSR2-NSIDC, the snow depth retrieval algorithm is probably not well adapted for rougher snow that can be compared to MYI in the Arctic.

L386-389: Because of highly dynamics weather conditions, with persistent strong winds in the Antarctic, snow thickness distribution is not directly related to snowfalls (Massom et al., 2001). Conversion from precipitation to snow depth is then very different to the Arctic and Antarctic snow pack is not an uniform slab. These features should partly explain model difficulties and differences between the two hemispheres

**4.3 Comparisons with in-situ and airborne data**

445 Lines 358-365 and Table 1: Although, the reason for the over and underestimations with OIB vs ASD and DuST are attributed to OIB product quality, it still do not 'completely' answer the estimation errors. I think the authors could 'speculate' these biases linked to spatiotemporal changes in snow properties.

We have added the following sentence :

L405: The variabilities of snow properties from OIB daily basis data to the monthly means of the other datasets certainly explain some of the deviations.

Figure 7: I think if table 1 is used, then figure 7 is almost redundant, since it does not provide any additional information. If figure 7 is used, then remove 'fitting line' legend. You may indicate that in the figure caption

We would prefer to keep the Table and the Figure 7 as we think the 2 informations are important. We have removed the "fitting line" legends and indicated in the figure caption.

Lines 380-395: There are a lot of 'assumptions' without any evidence reported here. 'Ka-ASR estimation exhibits very thin snow thickness. Although it might be expected in this area, this solution still contains unrealistic negative values due to the fact that ASIRAS and KAREN freeboards are nearly equivalent over FYI (without negative values).' So my first query is how did you 'expect' thin snow cover in these areas?

460 We expect thin snow mainly because most of the Baffin Bay is covered with first year ice and free in summer. We have added a reference Landy et al. (2017) studying sea ice in the Baffin Bay.

AMSR2B and DuST nearly 'always' overestimate the snow depth. Why does it 'always' overestimate? I read through the paragraph and found the analysis very vague. Suggest to rewrite the section.

It is an observation from Figure 9. We have removed the word always which is confusing since only one track is analysed.

**465 Towards an ASD climatology**

475

Line 396: That's a bold statement showing that ASD data shows good results? But do authors conclude from analysis with airborne and ins situ data ASD is better? From the analysis, the ASD snow depth are a first-time estimate in the Antarctic, but not good. Also till lines 403, the ASD-clim is presently valid only for Arctic correct?

We don't really attempt to demonstrate that ASD is the "best" but the relevancy of these data. In this case, the use of the word
"good" may not be the most appropriate. Indeed, the statement L396 come from the previous comparisons with validation data. At present we only have been able to present the ASD-clim valid in the Arctic. We agree that it is not fully clear in the sentence lines 403. We have modified in the revised version as follows:

L441-446: for that purpose, we have constructed a preliminary altimetric snow depth climatology in the Arctic by averaging all the ASD snow depth maps of each month during the 2013-2019 period (designated as the ASD-clim). An equivalent climatology could also be constructed for the Antarctic but the lack of validation prevents its validity to be demonstrated at

14

present. To demonstrate the relevance of this climatology in the Arctic for the years prior to 2013, the ASD-clim data are projected on all the tracks of the four OIB missions occurring between 2009 and 2012, and presented in Fig. 10.

**Also Figure 10: are the blue points from all tracks? It would be good to show them with different color for different years. The legend seems to have a problem. Please fix it.**

480 We agree that using different colors could be interesting. However, the objective is precisely to consider those data all together, on a statistical point of view. For latter analysis we will keep this relevant proposition. We have modified the legend as follows:

Fig.10. Scatterplots comparing the ASD and the W99m climatologies with OIB by considering all the tracks of 2009, 2010, 2011 and 2012 OIB missions.

**Impacts of snow depth on SIT**

**485 Lines 429-430. Need references for these values.**

**We have added references for these values :**

L472-474: As in Ricker et al. (2014), we assume the sea water density ( $\rho_w$ ) is set to 1024  $kg/m^3$ . Consistently with the approach of Laxon et al. (2013), sea ice density ( $\rho_i$ ) is set to 882  $kg/m^3$  for MYI and 917  $kg/m^3$  for FYI (Alexandrov et al., 2010). Furthermore, the snow density ( $\rho_s$ ) is set to 300  $kg/m^3$ .

**490 Lines 444-452: I do not have a clue why freshwater budget came into the analysis, out of the blue. Suggest to delete the entire lines if there it is irrelevant to the paper.**

ASD snow depths would allow to provide relevant sea ice volume estimate that would allows to re-evaluate the freshwater budget. The aim of this paragraph is to highlight the fact that these data are also important for other studies. Since the 2 other referees did not mention this point we would prefer to keep this paragraph.

**495 Figure 12: Either the caption is wrong or the panel titles are wrong. The first column says min, while caption says mean. Please correct. Also suggest to rescale levels of values in the color bar for both top and bottom plots.**

**We have corrected the mistake in Fig.12 caption.**

Fig.12. Inter-annual minimum  $(min(std_m), \text{ first column})$ , mean  $(\overline{std_m}, \text{ second column})$  and maximum  $(max(std_m), \text{ third column})$  maps of the standard deviation (first row) and the maximum deviation (second row) for the month of April (m = 04).

500 The snow products used to compute the maps are ASD, DuST, AMSR2B and W99m. These maps correspond to the case « obs snow products ».

**The authors also mention 'global' means. I would stick to using 'pan-Arctic' means (please correct everywhere in this section), as 'global' referes to all sea ice occuring everywhere in the planet.**

We do agree that pan-Arctic is much more adapted than global. We have modified everywhere in the revised document.

**505 Conclusions and Discussions**

I have already made my suggestions to split conclusions and discussions. Also, in the discussion section, please have sub-sections for both Arctic and the Antarctic, also discuss the study limitations in terms of validation data and also issues in biases due to snow properties.

We have done our best to re-organize and add supplementary informations to this section. Please refer to our previous answer to your general comments

**References**

Alexandrov, V., Sandven, S., Wahlin, J., and Johannessen, O.: The relation between sea ice thickness and freeboard in the Arctic, The Cryosphere, 4, 373–380, 2010.

- Andersen, O. B., Nilsen, K., Sørensen, L. S., Skourup, H., Andersen, N. H., Nagler, T., Wuite, J., Kouraev, A., Zakharova, E., and Fernandez, D.: Arctic freshwater fluxes from earth observation data, in: Fiducial Reference Measurements for Altimetry, pp. 97–103, Springer, 2019.
- 515 D.: Arctic freshwater fluxes from earth observation data, in: Fiducial Reference Measurements for Altimetry, pp. 97–103, Springer, 2019. Andreas, E. L., Jordan, R. E., and Makshtas, A. P.: Parameterizing turbulent exchange over sea ice: The Ice Station Weddell results, Boundary-Layer Meteorology, 114, 439–460, 2005.

Arndt, S. and Nicolaus, M.: Seasonal cycle and long-term trend of solar energy fluxes through Arctic sea ice, The Cryosphere, 8, 2219–2233, 2014.

- 520 Arndt, S., Meiners, K. M., Ricker, R., Krumpen, T., Katlein, C., and Nicolaus, M.: Influence of snow depth and surface flooding on light transmission through A ntarctic pack ice, Journal of Geophysical Research: Oceans, 122, 2108–2119, 2017.
  - Bin, C., Vihma, T., Zhanhai, Z., Zhijun, L., and Huiding, W.: Snow and sea ice thermodynamics in the Arctic: Model validation and sensitivity study against SHEBA data, Advances in Polar Science, 19, 108–122, 2008.
- Di Bella, A., Skourup, H., Bouffard, J., and Parrinello, T.: Uncertainty reduction of Arctic sea ice freeboard from CryoSat-2 interferometric mode, Advances in Space Research, 62, 1251–1264, 2018.
- Eicken, H., Fischer, H., and Lemke, P.: Effects of the snow cover on Antarctic sea ice and potential modulation of its response to climate change, Annals of Glaciology, 21, 369–376, 1995.
  - Guerreiro, K., Fleury, S., Zakharova, E., Rémy, F., and Kouraev, A.: Potential for estimation of snow depth on Arctic sea ice from CryoSat-2 and SARAL/AltiKa missions, Remote Sensing of Environment, 186, 339–349, 2016.
- 530 Ingram, W., Wilson, C., and Mitchell, J.: Modeling climate change: An assessment of sea ice and surface albedo feedbacks, Journal of Geophysical Research: Atmospheres, 94, 8609–8622, 1989.
  - Kern, M., Cullen, R., Berruti, B., Bouffard, J., Casal, T., Drinkwater, M. R., Gabriele, A., Lecuyot, A., Ludwig, M., Midthassel, R., et al.: The Copernicus Polar Ice and Snow Topography Altimeter (CRISTAL) high-priority candidate mission, The Cryosphere, 14, 2235–2251, 2020.
- 535 Kwok, R. and Cunningham, G.: Variability of Arctic sea ice thickness and volume from CryoSat-2, Phil. Trans. R. Soc. A, 373, 20140157, 2015.
  - Landy, J. C., Ehn, J. K., Babb, D. G., Thériault, N., and Barber, D. G.: Sea ice thickness in the Eastern Canadian Arctic: Hudson Bay Complex & Baffin Bay, Remote Sensing of Environment, 200, 281–294, 2017.
- Laxon, S. W., Giles, K. A., Ridout, A. L., Wingham, D. J., Willatt, R., Cullen, R., Kwok, R., Schweiger, A., Zhang, J., Haas, C., et al.:
  CryoSat-2 estimates of Arctic sea ice thickness and volume, Geophysical Research Letters, 40, 732–737, 2013.
- Ledley, T. S.: Snow on sea ice: Competing effects in shaping climate, Journal of Geophysical Research: Atmospheres, 96, 17195–17208, 1991.
- Mallett, R. D. C., Lawrence, I. R., Stroeve, J. C., Landy, J. C., and Tsamados, M.: Brief communication: Conventional assumptions involving the speed of radar waves in snow introduce systematic underestimates to sea ice thickness and seasonal growth rate estimates, The Cryosphere, 14, 251–260, https://doi.org/10.5194/tc-14-251-2020, https://www.the-cryosphere.net/14/251/2020/, 2020.
- Massom, R. A., Drinkwater, M. R., and Haas, C.: Winter snow cover on sea ice in the Weddell Sea, Journal of Geophysical Research: Oceans, 102, 1101–1117, https://doi.org/10.1029/96JC02992, https://agupubs.onlinelibrary.wiley.com/doi/abs/10.1029/96JC02992, 1997.
  - Massom, R. A., Eicken, H., Hass, C., Jeffries, M. O., Drinkwater, M. R., Sturm, M., Worby, A. P., Wu, X., Lytle, V. I., Ushio, S., et al.: Snow on Antarctic sea ice, Reviews of Geophysics, 39, 413–445, 2001.
- 550 Munoz-Martin, J. F., Perez, A., Camps, A., Ribó, S., Cardellach, E., Stroeve, J., Nandan, V., Itkin, P., Tonboe, R., Hendricks, S., et al.: Snow and Ice Thickness Retrievals Using GNSS-R: Preliminary Results of the MOSAiC Experiment, Remote Sensing, 12, 4038, 2020.
  - Overland, J., Dunlea, E., Box, J. E., Corell, R., Forsius, M., Kattsov, V., Olsen, M. S., Pawlak, J., Reiersen, L.-O., and Wang, M.: The urgency of Arctic change, Polar Science, 21, 6–13, 2019.

Perovich, D. K.: Light reflection and transmission by a temperate snow cover, Journal of Glaciology, 53, 201–210, 2007.

- 555 Perovich, D. K. and Richter-Menge, J. A.: Surface characteristics of lead ice, Journal of Geophysical Research: Oceans, 99, 16341–16350, 1994.
  - Powell, D. C., Markus, T., and Stössel, A.: Effects of snow depth forcing on Southern Ocean sea ice simulations, Journal of Geophysical Research: Oceans, 110, 2005.
- Ricker, R., Hendricks, S., Helm, V., Skourup, H., and Davidson, M.: Sensitivity of CryoSat-2 Arctic sea-ice freeboard and thickness on radar-waveform interpretation, Cryosphere, 8, 1607–1622, 2014.
- Shupe, M., Rex, M., Dethloff, K., Damm, E., Fong, A., Gradinger, R., Heuze, C., Loose, B., Makarov, A., Maslowski, W., et al.: The MOSAiC Expedition: A Year Drifting with the Arctic Sea Ice, Arctic report card, 2020.

Singarayer, J. S., Bamber, J. L., and Valdes, P. J.: Twenty-first-century climate impacts from a declining Arctic sea ice cover, Journal of Climate, 19, 1109–1125, 2006.

- 565 Sturm, M. and Massom, R. A.: Snow and sea ice, Sea ice, 2, 153–204, 2009. Tilling, R. L., Ridout, A., Shepherd, A., and Wingham, D. J.: Increased Arctic sea ice volume after anomalously low melting in 2013, Nature Geoscience, 8, 643–646, 2015.
  - Van Leeuwe, M. A., Tedesco, L., Arrigo, K. R., Assmy, P., Campbell, K., Meiners, K. M., Rintala, J.-M., Selz, V., Thomas, D. N., and Stefels, J.: Microalgal community structure and primary production in Arctic and Antarctic sea ice: A synthesis, Elementa: Science of the Anthropocene, 6, 2018.
- Willatt, R., Laxon, S., Giles, K., Cullen, R., Haas, C., and Helm, V.: Ku-band radar penetration into snow cover on Arctic sea ice using airborne data, Annals of Glaciology, 52, 197–205, 2011.

570

Worby, A. P., Markus, T., Steer, A. D., Lytle, V. I., and Massom, R. A.: Evaluation of AMSR-E snow depth product over East Antarctic sea ice using in situ measurements and aerial photography, Journal of Geophysical Research: Oceans, 113, 2008.

575 Zhou, L., Stroeve, J., Xu, S., Petty, A., Tilling, R., Winstrup, M., Rostosky, P., Isobel R, L., Liston, Glen E, R. A., Tsamados, M., and Nandan, V.: Intercomparison of snow depth over sea ice from multiple methods, The Cryosphere, 2020a.

Zhou, L., Stroeve, J., Xu, S., Petty, A., Tilling, R., Winstrup, M., Rostosky, P., Lawrence, I. R., Liston, G. E., Ridout, A., et al.: Intercomparison of snow depth over sea ice from multiple methods, The Cryosphere Discussions, pp. 1–35, 2020b.

---

## Author Comment (AC3)

MS Record: tc-2021-791:

**Title: Advances in altimetric snow depth estimates using bi-frequency SARAL/CryoSat-2 Ka/Ku measurements.**

*Florent Garnier, Sara Fleury, Gilles Garric, Jérôme Bouffard, Michel tsamados, Antoine Laforge, Marion Bocquet, Rénée Mie Fredensborg Hansen and Frédérique Rémy*

**Referee#3 global comment**

The presented manuscript by Garnier et al. shows promising results on the timely topic of snow thickness both in the Arctic and the Antarctic for use especially in sea-ice altimetry to derive sea-ice thickness and subsequent data products. It is therefore relevant and clearly in the scope of TC. However, I found the manuscript is overall lacking some clarity in writing and presenting and I therefore recommend major revisions. My comments/suggestions are detailed below and are also meant to be challenged.

**Answers to the anonymous referee#3 comments**

We would like to thank the reviewer for her/his careful reading of the manuscript and for the relevant and constructive remarks. In order to fit with your global comment, we have made an complete revision of the manuscript that should have considerably improved the quality of the writing and the readability of the document. We hope that the modifications we have made will meet your requirements. Please find below the details on how your general and specific comments have been taken into account. In this « author's response document », the referee's comments are in bold type, the answers are in italic type, and the corrections to the revised manuscript are in normal type.

**General comments**

**1. In general, I find the manuscript lacks clarity and readability due to a lack of proper use of articles (e.g., the Arctic) and other frequent grammatical errors that need to be corrected.**

*We apologize for the editorial oversights in the manuscript. We have proofread the manuscript thoroughly to correct as many grammatical errors as possible.*

**2. Starting with the abstract, parts of the manuscript are hard to grasp for the reader as they are not well structured or flooded with acronyms and multiple line-long passages put into parenthesis.**

*In our revision of the manuscript, we have tried our best to increase the clarity of the document. For instance we have added 2 tables:*
*- Table A4 summarizes the time period of the different snow depth product used in the document.*
*- Table A5 list of acronyms table.*

*Table A5 allows to only explicit most important acronyms, which reduces multiple line-long passages put into parenthesis*

*Also, In order to increase the clarify and better define the objectives of the paper we have modified the end of the introduction as follows:*

L132: A recent study of Zhou et al. (2020) presented an inter-comparison of available snow depth products from re-analyses, passive radiometry and altimetry (DuST). Similar, this paper reviews the state-of-the-art by comparing current main snow depth estimations. Yet, the main objective is to present and assess the upgraded version of the ASD product (see Sect. 2), covering the 2013-2019 period in both hemispheres. The article fits in with the upcoming HPCM CRISTAL mission (Kern et al., 2020) and aims to demonstrate the potential of such snow depth data to further specific studies on, for instance, improved sea ice volume representations, freshwater budgets, snow properties or data assimilation. Except from an analyse of the impact of the snow depth uncertainty on SIT retrieval, it doesn't explicitly address these questions.

The paper is organized as follows:

– First, we detail the methodology to process the ASD product and present all the datasets used in this study.

– Section 4 compare ASD with the main other existing snow depth satellite and model data in both hemispheres.

– The datasets are then assessed against OIB, CryoVEx and IMB validation data in the Arctic in section 5

– To circumvent the temporal limitation induced by SARAL, we propose in section 6 a preliminary snow depth climatology based on ASD.

– The last section 7 aims to quantify the SIT level of uncertainty due to the snow depth from an ensemble of SIT estimations calculated from the satellite and model snow depth datasets presented in the previous sections.

– We finally discuss and conclude, emphasizing current needs for snow depth data in sea ice studies.

**3. Furthermore, while this might sound a bit picky, I found it quite irritating to read Antarctica throughout the manuscript as this would reference the continent, not the region (which would be the Antarctic). This goes in line with the usage of unfamiliar specific terms such as sea ice sinking in L32, which I think should be avoided and replaced by proper, i.e., commonly used terms.**

*All the grammatical issues raised in your specific comments have been corrected. We have carefully replaced "Antarctica" by "the Antarctic" and systematically used "the" in front of Arctic. Many phrasings have also been modified to improve the readability of the document.*

**4. Aside from these rather editorial remarks, I found the inclusion of the model data rather redundant and fail to have a clear take-home message from this aside from that model data appears to be very bad in general for snow depth. Could the authors elaborate a bit more on why they choose to include this in the presented way for the different datasets?**

*The synergy with models is crucial for instance to fill the observation gap. Of course we understand that we did not analyse in deep the sources of differences since it an entire article could have been dedicated to that topic. There are mainly two reasons why we wanted to include model data in our article*

– *we wanted to show, in the form of the state of the art, that there are still very few representations of snow depth on sea ice and that they are still affected by significant uncertainties and inconsistencies.*

– *Synergy between observations and models will become crucial, either to improve the models or to complete the still incomplete observations.The first step towards this synergy is to observe the coherence and deviations of these different representations.*

70 **5. The usage of different time periods for different comparison exercises for the ASD even with the same "reference" such as OIB (e.g., L14-16) comes along quite unintuitive and should be better justified/clarified by the authors!**

*We agree that the different time periods may affect the readability of the article. In our point of view, this is yet necessary to make consistent comparisons. To prevent the reader from misunderstandings we have specified this point in section 4.1 Methodology as indicated below. In addition Table A4 clarify the time periods of the snow depth different products.*

75 L319-322: Since snow depth products and validation data are available over different time periods (see table x), the comparisons are also performed on different time periods. Our approach is to take, for each comparison, the time period common to all the compared data. The aim is to provide reliable statistics. For each comparison, a label specify the considered time period.
* * *
**Specific comments**

**L1: Do not capitalize sea-ice thickness (also consider hyphenation); potentially add "retrieval" after sea-ice thickness for clarification!**

*This comment has been taken into account in all the revised document.*

**L2: Please include "the" in front of the term "Arctic" most likely throughout the whole manuscript – but I'll try to point most of them out.**

*We have included "the" in front of the term Arctic in the whole revised manuscript.*

**L2: Maybe it is just me, but shouldn't it be the "modified Warren climatology" and not the "Warren modified climatology"?**

*We now use the phrasing "modified Warren-99 climatology" in the whole revised manuscript*

**L3: Please consider using "the Antarctic" instead of "Antarctica" as this geographically references the continent where we would not expect much sea ice anyways.**

*We now use "the Antarctic" instead of "Antarctica" in the whole revised manuscript.*

**L9/11: add "the"; see L2**

*We have included "the" in front of the term Arctic in the whole revised manuscript.*

**L10: The official website from JAXA uses the acronym "AMSR2" without a hyphen.**

*We have removed the hyphen in the whole revised manuscript.*

**L11: "It's" refers to DuST?**

*"It's" refer to the ASD product. We have corrected the sentence as follows:*

L11-12: The ASD product is further validated in the Arctic against the Ice Mass Balance (IMB) buoys, the CRYOsat Validation EXperiment (CryoVEx) and Operation Ice Bridge's (OIB) airborne measurements.

**L14/16: Why are these time periods different?**

*The ASD product is available from 2013. Then, we compare with OIB campaign occuring after 2013, with respect to the other snow depth product availability. To assess the ASD climatology it is more consistent to compare with the OIB data prior to 2013. Since OIB campaign have strated in 2009 we then use the 2009-2012 time period.*

**L29: For the sake of completeness, a reference to the ESA CCI sea-ice thickness product covering both the Arctic and Antarctic should be included when detailing available sea-ice thickness products. This would be in line with the authors detailed introduction to available snow on sea ice products later on:**
**Paul, S., Hendricks, S., Ricker, R., Kern, S., and Rinne, E.: Empirical parametrization of Envisat freeboard retrieval of Arctic and Antarctic sea ice based on CryoSat-2: progress in the ESA Climate Change Initiative, The Cryosphere, 12, 2437–2460, https://doi.org/10.5194/tc-12-2437-2018, 2018.**
**Or the referenced data publications by Hendricks et al. (2018).**

*The articles Paul et al. (2018); Hendricks et al. (2018) have been added to the revised version of the document.*

L28 : By integrating such sea ice freeboard estimations in the hydrostatic equilibrium equation, several SIT products have been computed (e.g., Laxon et al., 2013; Kwok and Cunningham, 2015; Guerreiro et al., 2017; Paul et al., 2018; Landy et al., 2019;

Laforge et al., 2020).

115    L462-463 : Note, that experimental SIT estimations (Hendricks et al., 2018) have already been done in the Antarctic in the framework of the Sea Ice Climate Change Initiative (SI-CCI).

**L32: sea ice sinking a very unfamiliar terminology**

*We have modified the sentence as follows:*

L32-33: For example, it is necessary to account for the snow loading (Laxon et al., 2013)

120   **L37: "transfer of solar heat energy of the ice-ocean interface" reads kind of clunky and should be rephrased as energy from the ocean directed upwards probably does not fall under the term of "solar heat energy".**

*We have modified the sentence as follows:*

L37-38: Because of its high albedo and a low thermal conductivity, the snow regulates the transfer of solar heat energy penetration accross the ice-ocean interface (e.g., Grenfell and Maykut, 1977; Sturm et al., 1997)

125   **L42: "the" again**

*We have included "the" in front of the term Arctic in the whole revised manuscript.*

**L56: Insert a space before "Their"**

*We have inserted a space before their.*

**L68: If I am not mistaken, this should be "have" instead of "has" as we talk of the campaigns**

130   *We agree. the sentence have been corrected.*

**L69/70: "the Antarctic" again**

*We now use "the Antarctic" instead of "Antarctica" in the whole revised manuscript.*

**L71: There is also an English URL seaiceportal.de**

*We now provide the link (https://seaiceportal.de/en/) in the revised version of the manuscript.*

135   **L86: no " the" needed before radiometric data**

*We have removed "the" before radiometric data*

**L89 and 58: two different Names for the AMSR-E/2 missions and acronyms... please correct!**

*In the whole revised manuscript AMSR-E refers to the Advanced Microwave Scanning Radiometer for the Earth Observing System and AMSR2 to the Advanced Microwave Scanning Radiometer-2. To ensure clarity and consistency, the use of identical*
140   *spellings have been verified.*

**L122: remove "over"**

*We have removed over.*

**L128 following: This part would profit from a numbered list format or bullet points**

**L134: In general discussion comes before the conclusion**

145   **L139/140: In Line 109 the authors referenced CryoSeaNICE as source of the development?**

*A preliminary version of the ASD product has been developed as part of the CryoSeaNICE project but the version presented in this article has been developed whithin the CSAO+ and Polar+ Snow on Ice projects. We agree that the phrasing in the introduction (L109-) might be confusing. For clarity we have modified these sentences as follows:*

L123-127: Recently, Armitage and Ridout (2015) have demonstrated this possibility by considering the difference of pene-
150   tration between the CS-2 Ku-band frequency radar (13.5 GHz), assuming it is reflected near the snow/ice interface, and the

SARAL/AltiKa Ka-band frequency radar (35.7 GHz), assuming it is reflected near the top of the snow pack, i.e the air/snow interface. Thereafter, a preliminary Altimetric Snow Depth (ASD) version covering the 2013-2016 winter period has been developed at the « Laboratoire d'Etudes en Géophysique et Océanographie Spatiales » (LEGOS) during the European Space Agency (ESA) CryoSat SciEnce-oriented data ANalysis over Sea-ICE areas (CryoSeaNICE) project (Guerreiro et al., 2016).

**L144: 500x500 of what? Pixels?**

*500x500 is the number of pixel of 12.5km in the Arctic grid. To clarify this point, the sentence has been modified as follows:*

L297: Model and satellite snow depth estimations are projected onto the same EASE2 grid with a pixel size of 12.5 km.

**L147: Technically, these are still elevations that are compared from what we read before in the text and not yet free-boards.**

*We indicated freeboard because our methodology is based on the difference of freeboards. As you requested we have modified the sentence as follows:*

L158: The main assumption is that the difference between these two surface elevations is only due to the penetration of the Ku radar in the snow pack, and that the Ku radar penetrates fully to the snow/ice interface.

**L150 and 154: The authors reference the Baseline C handbook but use Baseline B data? Is that an error/typo and in case it is not, could the authors elaborate a bit more on their reasoning why they use a quite outdated baseline for their data? And what differences if any they would expect to more recent Baseline-based products?**

*We use the GOP CryoSat-2 Baseline B until 2017 and Baseline C from 2017 to 2019. The reason why we did not used the baseline C for all the time period is simply because the Baseline C was not available at the time we had processed the data. It is now available but it would need some times to reprocess all the data. Because the waveforms are exactly the same in the two Baselines (and then the computed snow depths), we have decided to keep this version for this article. Note that we are currently re-processing the data so that the ASD product will be computed over the 2013-2020 period in both hemisphere from the baseline C only . The only difference is then that it will also cover the SARIN mode zones. You are right that it is important to let the reader (and probable futur users be aware of this). For that purpose the sentences L151-155 of the article has been modified as follows:*

L167-170: Since the Baseline-C PLRM GOP product was only available from 2017 at the time we have computed the ASD data, we have used the Baseline B for the period 2013-2016. It does not impact ASD since we use only the L1b product levels which have identical waveforms on both baselines. However, the baseline B does not include the SARin data. Then, the ASD data does not yet cover the SARin mode zones. The next version of the ASD product will include SARin mode zones.

**L170: The abbreviation should be "TFMRA"**

*This has been corrected in the revised version.*

**L170/174: Could the authors elaborate a bit on their reasoning for a two-time smoothing? Is the data otherwise that noisy?**

*The 25km radius smoothing on leads is because we consider that the sea level anomaly should not significantly vary within this window. Variability should then be altimetric noise. Note that uncertainties are also computed using this assumption. For the 50km median smoothing applied to the retrieved freeboards, it is subject to discussions. Actually, we also compute freeboards with 25km smoothing. Other groups like AWI use a 25km smoothing on freeboards. This is more a matter of choice that only*

*very weakly affect the results. In our opinion the 50km smoothing reduce noises in marginal and coastal zones. It should be more adapted to reduce SARIN mode noises that will be included in the next versions .*

**L175/176: Would one expect comparable results between the two approaches of crossover calculations and monthly maps? Did the authors look into the differences and can provide a bit of insight?**

*The ASD product presented here is indeed an updated version of the product presented in Guerreiro et al. (2016). We do agree that first compare the two version and explain the differences would have been a consistent approach.*

*Of course we have first investigated the deviations between the 2 products. It appears that the results are quite different, with significantly higher gradients in the previous version but it is difficult to draw some conclusion from the differences. Is it mainly due to extrapolations (of croosover snow depths points) ? is it due to monthly mean freeboard smoothings ? along track better consistencies ? Note that, in addition, the previous ASD version hasn't been computed as a product for users but more as an experimental dataset for the work of a PhD. The data have not been computed after 2015 and only cover the central Arctic.*

*Since the article is already quite long and compare a larger amount of data we have chosen to avoid this comparison that, for our point of view, does not bring much information. Then apart from your imperative requirement for publication, we would prefer not to put more information concerning this previous product in this article.*

**Figure1: I would highly suggest to use "named" subpanels such as a), b), c), etc. as they can be more easily references in the text and the figure caption. This could improve the overall readability of the manuscript.**

*We do agree that it is a relevant advise. Unfortunately we haven't got the time before re-submission to modify this point. In case of a second review, we would do it.*

**L219: There is a "." missing before "Because".**

*This has been corrected in the revised version.*

**L221: Starting in which year ending in 2019?**

*To clarify this point we have modify the revised manuscript as follows:*

L239-240: This product is available on a daily basis from November 2012 to April 2018 on a polar stereographic grid with a 25 km x 25 km resolution.

**L247: The sentence reads incomplete after the URL, please change.**

*We have modified the sentence as follows in the revised version of the document.*

L262-264: Data used in this paper are snow depth monthly mean maps provided on a 100 km $\times$ 100 km stereographic polar grid issued from the NESOSIM 1.0 configuration (Petty et al., 2018). It is freely available at https://earth.gsfc.nasa.gov/cryo/data/nasa-eulerian-snow-sea-ice-model-nesosim.

**Figure2: "winter months" should be clarified in the figure caption**

*The caption has been modified as follows :*

Times series of annual mean snow depth of the different products in both hemispheres. Annual means are calculated as the average of the spatial means of all monthly grid maps. Only the 6 winter months are considered in the calculation (November-April in the Arctic and May-October in the Antarctic). Note that the AMSR2 data are not represented in the Arctic since the

AMSR2-NSIDC product is only available over FYI and the AMSR2B product is only available at full spatial coverage (FYI and MYI) in March and April.

**L277: how are they projected? Or rather resampled – using a NN approach? How are multiple assignments dealt with (mean,max,min)?**

225 *The gridded model and snow depth monthly data are projected onto the EASE2 grid of ASD using a simple two-dimensional multivariate interpolation (griddata function in python). We have added this specification in the revised manuscript:*

L296: Model and satellite snow depth estimations are projected onto the 12.5km pixel size EASE2 grid similar of ASD using a linear two-dimensional multivariate interpolation.

**L293: What do the authors mean by in-situ tracks? The OIB data?**

230 *We mean the gridded data (model and satellite products) are projected using a bilinear interpolation (linear is reductive) on the tracks of the in-situ and airborne data. It means that we compare the differents data in the space of the validation data instead of gridding the validation data in EASE2 grid as it is more commonly done. We think that this methodology is more appropriate (in data assimilation calculations are always performed in the observational space). We have modified the revised manuscript as follows :*

235 L315: snow depth model and satellite gridded maps are projected along the aircraft trajectories using a bilinear interpolation.

**L294: "monthly maps corresponding to that day" – does that mean, e.g. the the 12th of January gets monthly map of January? Please clarify.**

*It is exact. A snow depth airborne data of the 12th of January would be compared with, for instance, the ASD snow depth monthly mean data of January. To clarify this point we have modified the manuscript as follows :*

240 L315-316: Airborne and in-situ are generally daily data, the comparisons are performed with mean maps of the month to which that day belongs.

**L303: "lower than" – this means the snow is 3 cm thinner compared to AMSR2B? Please clarify!**

*Indeed, "lower than" means that the snow is, on average, 3 cm thinner compared to AMSR2B.*

**L307: "the" Appendix**

245 *This has been corrected in the revised version.*

**L309: This reads like we are comparing complete winter seasons but from what I read before I assumed DuST data is only available and used for the months of March and April? This should be clarified by the authors.**

*It not exactly true. The algorithm to compute DuST data use a re-calibration function based on the OIB data, which only exist in March and April. However the function is used to calculate snow depth data for the 6 months of winter. Note that considering*

250 *that snow depth is thicker in March and April, it is consistent to observe that positive bias for the DuST. To clarify this point we have modified the section 3.1.1 as follows*

L221-222: The DuST (Lawrence et al., 2018) data are provided in the Arctic on a 1.5° longitude x 0.5° latitude grid for by the Centre for Polar Observation and Modelling, University College London (www.cpom.ucl.ac.uk/DuST). They are available for the 6 winter month (November to April) until 2018.

255 **L314: only by the spatial availability or also by the temporal?**

*We have added this point in the revised version of the document:*

L339: ....since DuST calibration is limited by the spatial and temporal availability of OIB.

**Table A1 in general is referenced frequently in this section. Maybe the authors should consider to actually include it here and not the Appendix.**

260 *We agree that the table is frequently referenced in this section. Meanwhile this table is not needed to understand the section. Table A1 is only necessary for in-deep analysis. Then, for clarity, we would prefer to keep it in the Appendix.*

**Figures 3/4: see comment on Figure 2 [(a),b),c) + winter month)]**

*winter month has been explicated in the methodology section.*

**L343: "weaker" is quite judging... maybe rephrase to "lower"**

265 *This has been corrected in the revised version.*

**L359: I am not familiar with the DuST product in any detail but form what the authors provide as information it is specifically tuned/recalibrated using OIB data yet falls short in every comparison to OIB data? Could the authors elaborate on their opinion why that is the case and what they do specifically better?**

*It is not calibrated with the quicklook OIB version. We do not want to say that we do better but that the use of the pLRM data*
270 *for CS-2 avoid a recalibration with OIB which is dependent to 1) the OIB version 2) the space and time resolution of OIB 3) not possible in the Antarctic at the moment.*

**L359: The authors should also stay consistent with their use of Ka/Ku or band/-band throughout the manuscript!**

*This point has been taken into account in the whole manuscript.*

**Figure 6: It is quite hard to separate in other figures as well but I found it especially hard in this one between the three**
275 **different red colors!**

*The readability of the figure has been improved in the revised version of the article.*

**L404: "a more optimal" – maybe just say "better/improved"?**

*We have corrected this point using the word "better".*

L446: Fig. 10 shows that the ASD climatology would be a better solution than W99m,

280 **L420: Again, this lacks a proper reference to the ESA CCI product, see comment above.**

*We have added the reference to the work of Hendricks et al. (2018) :*

L461-462: Note, that experimental SIT estimations (Hendricks et al., 2018) have already been done in the Antarctic in the framework of the Sea Ice Climate Change Initiative (SI-CCI).

**L497: The double "mean" in this term is very confusing and should be clarified.**

285 *To simplify, we have modified the revision version as follows :*

L536-537: Considering « obs snow products » and all winter months, we obtained a SIT mean standard deviation of ≈ 20 cm (14%).

**Figure 12: The order and titles of the figures do not match the figure caption (min, mean, max). The last caption sentence**

The snow products used to compute the maps are ASD, DuST, AMSR2B and W99m. These maps correspond to the case « obs snow products ».

295 **L537: "based"**

*We have corrected this mistake in the revised document*

**References**

Armitage, T. W. K. and Ridout, A. L.: Arctic sea ice freeboard from AltiKa and comparison with CryoSat2 and Operation IceBridge, Geophysical Research Letters, 42, 6724–6731, https://doi.org/10.1002/2015GL064823, https://agupubs.onlinelibrary.wiley.com/doi/abs/10.1002/2015GL064823, 2015.

Grenfell, T. C. and Maykut, G. A.: The optical properties of ice and snow in the Arctic Basin, Journal of Glaciology, 18, 445–463, 1977.

Guerreiro, K., Fleury, S., Zakharova, E., Rémy, F., and Kouraev, A.: Potential for estimation of snow depth on Arctic sea ice from CryoSat-2 and SARAL/AltiKa missions, Remote Sensing of Environment, 186, 339–349, 2016.

Guerreiro, K., Fleury, S., Zakharova, E., Kouraev, A., Rémy, F., and Maisongrande, P.: Comparison of CryoSat-2 and ENVISAT radar freeboard over Arctic sea ice: toward an improved Envisat freeboard retrieval, The Cryosphere, 11, 2059–2073, https://doi.org/10.5194/tc-11-2059-2017, https://www.the-cryosphere.net/11/2059/2017/, 2017.

Hendricks, S., Paul, S., and Rinne, E.: Southern hemisphere sea ice thickness from the CryoSat-2 satellite on a monthly grid (L3C), v2.0e thickness and volume., https://doi.org/10.5285/48fc3d1e8ada405c8486ada522dae9e8, 2018.

Kern, M., Cullen, R., Berruti, B., Bouffard, J., Casal, T., Drinkwater, M. R., Gabriele, A., Lecuyot, A., Ludwig, M., Midthassel, R., et al.: The Copernicus Polar Ice and Snow Topography Altimeter (CRISTAL) high-priority candidate mission, The Cryosphere, 14, 2235–2251, 2020.

Kwok, R. and Cunningham, G.: Variability of Arctic sea ice thickness and volume from CryoSat-2, Phil. Trans. R. Soc. A, 373, 20140 157, 2015.

Laforge, A., Fleury, S., Dinardo, S., Garnier, F., Remy, F., Benveniste, J., Bouffard, J., and Verley, J.: Toward improved sea ice freeboard observation with SAR altimetry using the physical retracker SAMOSA+, Advances in Space Research, 2020.

Landy, J. C., Tsamados, M., and Scharien, R. K.: A facet-based numerical model for simulating SAR altimeter echoes from heterogeneous sea ice surfaces, IEEE Transactions on Geoscience and Remote Sensing, 57, 4164–4180, 2019.

Lawrence, I. R., Tsamados, M. C., Stroeve, J. C., Armitage, T. W., and Ridout, A. L.: Estimating snow depth over Arctic sea ice from calibrated dual-frequency radar freeboards, The Cryosphere, 12, 3551–3564, 2018.

Laxon, S. W., Giles, K. A., Ridout, A. L., Wingham, D. J., Willatt, R., Cullen, R., Kwok, R., Schweiger, A., Zhang, J., Haas, C., et al.: CryoSat-2 estimates of Arctic sea ice thickness and volume, Geophysical Research Letters, 40, 732–737, 2013.

Paul, S., Hendricks, S., Ricker, R., Kern, S., and Rinne, E.: Empirical parametrization of Envisat freeboard retrieval of Arctic and Antarctic sea ice based on CryoSat-2: progress in the ESA Climate Change Initiative, The Cryosphere, 12, 2437–2460, 2018.

Petty, A. A., Webster, M., Boisvert, L., and Markus, T.: The NASA Eulerian Snow on Sea Ice Model (NESOSIM) v1.0: initial model development and analysis, Geoscientific Model Development, 11, 4577–4602, https://doi.org/10.5194/gmd-11-4577-2018, https://www.geosci-model-dev.net/11/4577/2018/, 2018.

Sturm, M., Holmgren, J., König, M., and Morris, K.: The thermal conductivity of seasonal snow, Journal of Glaciology, 43, 26–41, 1997.

Zhou, L., Stroeve, J., Xu, S., Petty, A., Tilling, R., Winstrup, M., Rostosky, P., Lawrence, I. R., Liston, G. E., Ridout, A., et al.: Inter-comparison of snow depth over sea ice from multiple methods, The Cryosphere Discussions, pp. 1–35, 2020.

---

## Author Response (AR2)

MS Record: tc-2021-791:

**Title: Advances in altimetric snow depth estimates using bi-frequency SARAL/CryoSat-2 Ka/Ku measurements.**

*Florent Garnier, Sara Fleury, Gilles Garric, Jérôme Bouffard, Michel tsamados, Antoine Laforge, Marion Bocquet, Rénée Mie Fredensborg Hansen and Frédérique Rémy*

**Report#2 global comment to the "authors response to the referee#2 comments"**

Dear authors,
thank you very much for you detail responses to my raised issues. I am in general quite satisfied with your made changes and have just some small comments and remarks regarding some of your answers. As I stated earlier, these are of course meant to be challenged as I also would therefore challenge some of your responses.

**Answers to the report#2**

We would like to thank once again the reviewers for their careful reading of the manuscript and for the relevant and constructive remarks. We really think that the quality of the document has been considerabely improved by your comments. We have done our best to take into account the last comments you have made on the manuscript. We hope that it will meet your requirements. Please find below the details on how your general and specific comments have been taken into account. The referee's comments are in bold type, the answers are in italic type, and the corrections to the revised manuscript are in normal type. Note that we have also removed the grey coloured cells as requested by the editor.

**Answers to the report#2 : technical comments**

**Line 137: I think this should be "analysis" and not "analyse"**

*This correction has been made in the final document.*

**Figure 1: I would still encourage to follow my advise but of course this is not mandatory for publication. Nonetheless, I find it a bit weird that this is expected to be such a time costly step... from my own experience, adding these minor changes to a Figure would be a work for an hour tops.**

*As requested, we have added the a),b),c)...labels in the figure 1 and modified the caption in consequence.*

**L165 (question about the Baseline): While I can live with the additional information also put in to the manuscript by the authors, given that there is just now the new release of Baseline E data, using Baseline B/C (up to three "steps"**

**older data than what is now would be called state-of-the-art) data for a new publication seems really out-fashioned to me. Especially, given that by now pretty much everything is reprocessed by ESA and available (at least Baseline D) and to my understanding there is especially quite some change between Baseline C and D regarding the waveforms.**

*Maybe our previous answer was not enough clear. We think that you are talking about the CryoSat-2 SAR mode data !*

*Indeed, the Baseline-E has just been released for SAR mode data (the authors have participated to the analysis and validation). However, to compute the snow depth, we use the pseudo low resolution mode data (Baseline Ocean) for which the Baseline C is currently the latest Baseline. At the time we wrote the article, the Baseline C was only available from 2017. Since, The reprocessing is now achieved, we have already re-computed the ASD data over the whole period using the latest Baseline C. It means that the latest ASD version now includes SARIN modes for both hemisphere over the entire 2013-2020 period.. Note that a note relating the availability of ASD and this update is planned in Earth Observation information discovery platform https://earth.esa.int/eogateway.*

*The description and explaination of the CS-2 Baselines used to compute the ASD product is described from L161 to L170 in the final version of the article. In order to be clearer, we have modified the final version as follows:*

L167-170: Since the latest Baseline-C PLRM GOP product was only available from 2017 at the time we have computed the ASD data, we have used the Baseline B for the period 2013-2016. The next version of the ASD product will be produced with only the Baseline-C PLRM GOP product to include all SARin mode zones.
* * *
**L185 (question about the smoothing): To my understanding to what I can read out of the AWI manual (https://epic.awi.de/id/eprint/ there is no smoothing applied but rather gridding to the 25km grid. I guess this could also be called "smoothing" to an extent as several measurements in a 25x25km grid have to be "merged" somehow into one value, but is in general something different than an additional filter like the authors apply here. I think this has to be justified and is not subjective or a "matter of choice" as stated by the authors. The information by the authors that they also produce data with a 25km smoothing is very interesting. Does this impact their results in a comparison? This also again stresses my differentiation made above between an additional filter/smoothing and the simple gridding. Sadly, these information did not make it into the manuscript.**

**L190 (question about the comparison of Crossovers and Maps): This sadly also applies to this reply. Personally, I agree with the authors that this article is already quite long but I think there is quite some important information in your reply. While I do not want to (or can) force the authors to put all this to the public I just generally want to argue for a more transparent work ethic in science in general... Finding these apparently substantial differences could be important to other researchers/users same as the reasoning why to do certain processing steps like the smoothing. If nothing else, I would like to encourage the authors to try to improve in this regard in future publications and their work.**

*In our opinion, the most important point when comparing data is the consistency between the space and time resolutions. AWI use 25km grids when we use 12.5 km grids with a additional 25 km radius smoothing. The two approaches should provides relatively equivalent results but, of course, the non-gaussianities have some impacts that we are not able to properly quantify. However, we are quite convinced that it rather be quite small. Note that we still believe that the 25 km is quite a matter of choice, in the sense that it is an expected effective resolution of the satellite data. (In my personnal opinion, i would say that it is very spatially dependent and should be more 50km or even 100 km at some locations.)*

*Compare to satellites, in-situ and airborne data have far higher space and time resolutions. Then, it is mandatory to smooth the external data to provide consistent comparisons. Of course, because data in 25 km sections are rarely gaussians, the smoothong has a mean effect which tends, in this case, to reduce the mean values. This mean effect is important and must be considered to correct potential bias with satellite due to the difference in resolutions.*

*To conclude, we completely agree with you on the duty of scientists to be transparent. It is even probably more important nowadays, when scientific studies are easily diverted for headline announcements. We apologize if you felt like we wanted to*

*hide some parts of the work, be sure that it was not intentional and that we will make our best to improve this in our future works.*

75 *In order to consider these comments, we have added the following sentences in the final version of the manuscript*

L189: Finally, a 25 km radius median smoothing is applied to the retrieved freeboards. The reason is that we assume that the ASD product should not be able to provide relevant information at smaller scales. However, additional analyses and comparisons with validation data would be necessary to properly characterize this point. Note that the ability to consistently observe small scales would certainly be significantly improved in the future dual-frequency snow depth products from the CRISTAL 80 mission.

L321-326: In order to achieve similar spatial scales in the comparisons, we have applied a 25 km window rolling mean to smoooth the external data. Due to the non-gaussianities within 25 km sections, we have noticed that the smoothing has a mean effect which slightly tends to reduce the mean value of data. It is important to consider that this allows to correct potential bias that would be induced by a difference in resolutions. Note that the space and time consistency between model and satellite data 85 is also ensured by projection onto similar grids.

L189: Finally, a 25 km radius median smoothing is applied to the retrieved freeboards. The reason is that we assume that the ASD product should not be able to provide relevant information at smaller scales. However, additional analyses and comparisons with validation data would be necessary to properly characterize this point. Note that the ability to consistently observe small scales would certainly be significantly improved in the future dual-frequency snow depth products from the CRISTAL 90 mission.